

# Oceanic CO₂ outgassing and biological production hotspots induced by pre-industrial river loads of nutrients and carbon in a global modelling approach

Lacroix Fabrice[1,2], Ilyina Tatiana[1], and Hartmann Jens[3]

[1]Ocean in the Earth System, Max Planck Institute for Meteorology, Hamburg, Germany
[2]Department of Geoscience, Université Libre de Bruxelles, Brussels, Belgium
[3]Institute for Geology, Center for Earth System Research and Sustainability, University of Hamburg, Hamburg, Germany

*Correspondence to:* Fabrice Lacroix (fabrice.lacroix@mpimet.mpg.de)

**Abstract.** Rivers are a major source of nutrients, carbon and alkalinity for the global ocean, where the delivered compounds strongly impact biogeochemical processes. In this study, we firstly estimate pre-industrial riverine fluxes of nutrients, carbon and alkalinity based on a hierarchy of weathering and land-ocean export models, while identifying regional hotspots of the land-ocean exports. Secondly, we implement the riverine loads into a global biogeochemical ocean model and describe their implications for oceanic nutrient concentrations, the net primary production (NPP) and CO₂ fluxes globally, as well as in a regional shelf analysis. Thirdly, we quantify the terrestrial origins and the long-term oceanic fate of riverine carbon in the framework, while assessing the potential implementation of riverine carbon fluxes in a fully coupled land-atmosphere-ocean model. Our approach leads to annual pre-industrial riverine exports of 3.7 Tg P, 27 Tg N, 158 Tg Si and 603 Tg C, which were derived from weathering and non-weathering sources and were fractionated into organic and inorganic compounds. We thereby identify the tropical Atlantic catchments (20% of global C), Arctic rivers (9% of total C) and Southeast Asian rivers (15% of total C) as dominant providers of carbon to the ocean. The riverine exports lead to a global oceanic source of CO₂ to the atmosphere (231 Tg C yr$^{-1}$), which is largely a result of a source from inorganic riverine carbon loads (183 Tg C yr$^{-1}$), and from organic riverine carbon inputs (128 Tg C yr$^{-1}$). Additionally, a sink of 80 Tg C yr$^{-1}$ is caused by the enhancement of the biological carbon uptake by dissolved inorganic nutrient inputs, resulting alkalinity production and a slight model drift. While large outgassing fluxes are mostly found in proximity to major river mouths, substantial outgassing fluxes can also be observed further offshore, most prominently in the tropical Atlantic. Furthermore, we find evidence for the interhemispheric transfer of carbon in the model; we detect a stronger relative outgassing flux (49% of global river induced outgassing) in the southern hemisphere in comparison to the hemisphere's relative riverine inputs (33% of global river inputs), as well as an outgassing flux of 17 Tg C yr$^{-1}$ in the Southern Ocean. Riverine exports lead to a strong increase in NPP in the tropical West Atlantic, Bay of Bengal and the East China Sea (166%, 377% and 71% respectively). While the NPP is not strongly sensitive to riverine loads on the light limited Arctic shelves, the CO₂ flux is strongly altered due to substantial dissolved carbon supplies to the region. While our study confirms that the ocean circulation is the main driver for open ocean biogeochemical distributions, it reveals the necessity to consider riverine exports for the representation of heterogeneous features of the coastal ocean, to





represent riverine-induced carbon outgassing, as well as to consider the long-term volcanic $CO_2$ flux to close the atmospheric carbon budget in a coupled land-ocean-atmosphere setting.

# 1 Introduction

Rivers deliver substantial amounts of land-derived biogeochemical compounds to the ocean. For the present-day, they provide

4-11 Tg P yr$^{-1}$ of phosphorus (P), 37-66 Tg N yr$^{-1}$ of nitrogen (N), 158-200 Tg Si yr$^{-1}$ of dissolved silica (Si) and iron (Fe, no estimate available) as dissolved and particulate inorganic and organic compounds to the ocean (Seitzinger et al., 2005, 2010; Dürr et al., 2011; Beusen et al., 2009; Tréguer and De La Rocha, 2013; Beusen et al., 2016). They also supply carbon (C) as dissolved inorganic carbon (260-550 Tg C yr$^{-1}$), dissolved organic carbon (130-380 Tg C yr$^{-1}$), as well as particulate organic carbon (100-197 Tg yr$^{-1}$) to the ocean (Meybeck, 1982; Amiotte Suchet and Probst, 1995; Ludwig et al., 1996, 1998;

Mackenzie et al., 1998; Meybeck and Vörösmarty, 1999; Seitzinger et al., 2005; Hartmann et al., 2009; Seitzinger et al., 2010; Regnier et al., 2013; Aarnos et al., 2018). These loads, which originate from natural and anthropogenic sources, strongly affect the biogeochemistry in the coastal ocean, where nutrients and carbon are transformed during biogeochemical processes, are exported to the sediment, or are exported offshore (Stepanauskas et al., 2002; Froelich, 1988; Dagg et al., 2004; Krumins et al., 2013; Sharples et al., 2017). In global ocean models however, biogeochemical riverine exports and their contributions

to the cycling of carbon have been strongly simplified or ignored. In this study, we attempt to reduce these gaps of knowledge by estimating the magnitudes of biogeochemical riverine exports as a function of Earth System Model variables for the pre-industrial time period by using a hierarchy of weathering and land-export models. We then describe their long-term implications for oceanic biogeochemical cycles, with the NPP and $CO_2$ fluxes as focal points.

Natural riverine carbon and nutrients originate from the terrestrial biosphere and the weathering of the lithosphere, which

both consume atmospheric $CO_2$ while supplying freshwaters with carbon (Ludwig et al., 1998).

Weathering directly releases nutrients (P, Si and Fe) that can be taken up by the terrestrial ecosystems, or exported directly to aquatic systems (Hartmann et al., 2014). In these ecosystems, they are reported to enhance the carbon uptake due to their limitation of the biological primary production (Elser et al., 2007; Fernández-Martínez et al., 2014). Furthermore, alkalinity and carbon are released in the weathering process, while $CO_2$ is drawn down from the atmosphere (Amiotte Suchet and

Probst, 1995; Meybeck and Vörösmarty, 1999; Hartmann et al., 2009). Spatially explicit approaches that quantify the weathering release have a strong potential for providing fluxes of nutrients for Earth System Models (e.g. for P in Hartmann et al. (2014)). They also have been used in the past to provide valuable information regarding the drawdown of atmospheric $CO_2$ and its transformation to surface water alkalinity (Ludwig et al., 1998; Roelandt et al., 2010). It is acknowledged that chemical weathering rates are a first-order function of hydrology, lithology, rates of physical erosion, soil properties and temperature

(Amiotte Suchet and Probst, 1995; Hartmann et al., 2009, 2014). Approaches to estimate chemical weathering yields therefore depend on quantifying these controls (Hartmann et al., 2014).

The terrestrial biosphere is also a source of carbon and nutrients to freshwater systems (Meybeck and Vörösmarty, 1999; Seitzinger et al., 2010; Regnier et al., 2013). The formation of organic matter through biological primary production consumes





atmospheric $CO_2$ (Ludwig et al., 1998). Leaching and physical erosion can then mobilize dissolved and particulate organic matter from soils and peatlands, and export it to rivers. While the natural P and Fe within the organic matter originates from weathering, C and N mostly originate from atmospheric fixation (Meybeck and Vörösmarty, 1999; Green et al., 2004). Within rivers, autochthonous production can take place, which further transforms inorganic nutrients to organic matter while taking up

5 $CO_2$. In soils and in rivers, organic matter remineralization also occurs, which transforms the organic matter back to dissolved inorganic nutrients and carbon.

 Rivers therefore export P, N, Si, Fe and C as dissolved and particulate inorganic and organic compounds, as well as alkalinity to the ocean. It is known that dissolved inorganic nutrients enhance the primary production in the ocean, and thus cause an uptake of atmospheric $CO_2$ (Tyrrell, 1999). The riverine inputs of dissolved inorganic carbon on the other hand causes carbon

10 outgassing in the ocean (e.g. Sarmiento and Sundquist (1992)). The fate and contributions of terrestrial organic matter to the oceanic carbon cycle have been open questions for over two decades (Ittekkot, 1988; Hedges et al., 1997; Cai, 2011; Lalonde et al., 2014). While the nutrients contained in reactive organic matter can control phytoplankton and bacterial growth in given regions (Seitzinger and Sanders, 1997; Stepanauskas et al., 2002; Björkman and Karl, 2003), the reactivity of organic matter has been strongly debated (Ittekkot, 1988; Hedges et al., 1997; Lalonde et al., 2014), since it is thought to already have undergone

15 substantial degradation in rivers (Ittekkot, 1988; Vodacek et al., 2003). In the case of terrestrial dissolved organic matter (tDOM), the previous riverine degradation leads to high carbon to nutrients ratios found in tDOM (i.e. C:P weight ratios of over 500, Meybeck (1982); Seitzinger et al. (2010)). The strong previous degradation of tDOM and its high carbon to nutrients ratios imply low biological reactivity in the ocean, yet tDOM is not a major constituent of organic mixtures in open ocean sea-water or sediment pore water (Ittekkot, 1988; Hedges et al., 1997; Benner et al., 2005), Recent studies have suggested that

20 photodegradation might be responsible for partly or completely degrading tDOM to dissolved $CO_2$, thus possibly closing this gap of knowledge (Vodacek et al., 2003; Fichot and Benner, 2014; Lalonde et al., 2014; Aarnos et al., 2018). The degradation of tDOM could cause substantial regional outgassing due to its large transfer of carbon (Cai, 2011; Müller et al., 2016; Aarnos et al., 2018). In the case of terrestrial particulate organic matter (POM), even stronger gaps in knowledge exist (Cai, 2011). POM has however been reported to affect coastal ocean biogeochemistry regionally by firstly controlling the availability of

25 nutrients and secondly by altering the optical properties of aquatic systems (Froelich, 1988; Dagg et al., 2004; Stramski et al., 2004). Furthermore, a substantial proportion of weathered P is exported to the ocean bound to iron (Fe-P, Compton et al. (2000)). Within estuaries and the coastal ocean, a fraction of P in Fe-P is thought to be desorbed, and thus converted to a bioavailable compound (dissolved inorganic phosphorus).

 The cycles of P and N and their land-ocean exports have been strongly perturbed over the 20th century, leading to a doubling

30 of P and N riverine loads at the least (Seitzinger et al., 2010; Beusen et al., 2016). In the case of C and Si, the perturbations have been reported to be far less substantial at the global scale, although regional changes could have large implications for the coastal ocean (Seitzinger et al., 2010; Regnier et al., 2013; Maavara et al., 2014, 2017). It has furthermore been suggested that riverine P and N exports were already substantially perturbed prior to 1850-1900 due to increased soil erosion from land-use changes, fertilizer use in agriculture and sewage sources (Mackenzie et al., 2002; Filippelli, 2008; Beusen et al., 2016). Since

35 global modelling studies that tackle anthropogenic perturbations of the climate are usually initialized for 1850 (Giorgetta et al.,



2013) or 1900 (Bourgeois et al., 2016), these exports due to non-natural sources should also be taken into account in the initial pre-industrial model states.

Until now, riverine point sources of biogeochemical tracers have been omitted or poorly represented in global ocean biogeo-chemical models, despite being suggested to strongly impact the biogeochemistry of coastal regions (Stepanauskas et al., 2002; Froelich, 1988; Dagg et al., 2004; Krumins et al., 2013; Sharples et al., 2017) and to cause a natural source of atmospheric $CO_2$ in the ocean (Sarmiento and Sundquist, 1992; Aumont et al., 2001; Gruber et al., 2009; Resplandy et al., 2018). This $CO_2$ outgassing flux of 0.2 to 0.8 Gt C $yr^{-1}$ is significant in the context of the present-day oceanic carbon uptake of around 2.3 Gt C $yr^{-1}$ (IPCC, 2013). In a modelling study tackling the interhemispheric transfer of carbon, Aumont et al. (2001) derived carbon loads from an erosion model and analyzed the oceanic outgassing caused by the riverine carbon. The impacts of nutrients and alkalinity were however not considered and carbon was only added to the ocean as dissolved inorganic carbon. Da Cunha et al. (2007) analyzed the impacts of present-day river loads on the oceanic primary production in a global biogeochemical model for an analysis period of 10 years. Bernard et al. (2011) added river loads to an ocean biogeochemistry model to focus on their implications for global opal export distributions. In a global coastal ocean study, Bourgeois et al. (2016) quantified the coastal anthropogenic $CO_2$ uptake, but did not analyze the impacts of riverine loads. To our knowledge, a study has yet to give a com-prehensive overview of pre-industrial land-ocean river exports and their long-term global impacts on oceanic biogeochemical cycling in a 3-dimensional framework. In previously published literature, riverine loads were added according to present-day estimates, despite severe perturbation of the land-ocean N and P exports having taken place during the 20th century (Beusen et al., 2016). An initial ocean state with pre-industrial riverine supplies would however be necessary to truly assess the tempo-ral dynamical impacts associated with these perturbations. Studies until now have also not considered differing characteristics of terrestrial organic matter to those of oceanic organic matter. Furthermore, it is often unclear under which criteria the al-kalinity supplies to the ocean were constrained in global models. Regional sensitivities of coastal regions to biogeochemical riverine loads have not been assessed at the global scale, largely due to the incapability of global models to represent plausible continental shelf sizes in the past (Bernard et al., 2011).

In this study, we **1.** implement a representation of pre-industrial riverine loads into a global ocean biogeochemical model, considering both weathering and non-weathering sources of nutrients, carbon and alkalinity. We compare our estimates with a wide range of published literature values, while also determining regions of strong riverine exports and their contributions to global exports. **2.** The implications of riverine fluxes for the oceanic NPP and $CO_2$ flux are assessed globally, as well as regionally in an analysis of shallow shelves. **3.** We evaluate the origins and fate of riverine carbon quantitatively, while assessing the balance between the land carbon uptake and the oceanic carbon outgassing. This balance of the land carbon uptake and the oceanic outgassing is then used to assess the potential implementation of riverine fluxes in a fully coupled land-atmosphere-ocean setting.





## 2 Methods

To address the objectives of this study, we derived the most relevant pre-industrial land-ocean fluxes of biogeochemical compounds dependently on pre-industrial output of Earth System Model simulations. We used a hierarchy of models to derive biogeochemical compound sources from weathering and non-weathering to catchments, and estimated their terrestrial trans-

formations to organic matter. This resulted in a spatially explicit quantification of global riverine loads. The derived riverine loads were then implemented into a global ocean biogeochemical model in order to assess their global and regional impacts on the ocean biogeochemistry. In order to quantify the effects of the riverine supplies on coastal regions, we also defined 10 shelves with depths of less than 250m.

### 2.1 Deriving pre-industrial riverine loads

We first briefly describe the framework used to derive pre-industrial riverine exports here, while the individual models and assumptions are explained in detail in the next subsections. We focused on the exports of phosphorus (P), nitrogen (N), silica (Si), iron (Fe), carbon (C) and alkalinity (Alk) for global catchments. The catchments were defined by using the largest 2000 catchments from the Hydrological Discharge (HD) model (Hagemann and Dümenil, 1997; Hagemann and Gates, 2003), a component of the Max Planck Institute Earth System Model (MPI-ESM). The catchments were derived from the runoff flow

directions of the model at a horizontal resolution of 0.5 degrees. The exorheic river catchments (catchments, which discharge into the ocean) were considered to be catchments with river mouths that have a distance of less than 500 km to the coastline in the HD model. Catchments with larger distances were considered to be endorheic catchments (catchments, which do not discharge into the ocean). The biogeochemical tracers released in these in the endorheic catchments in our framework were assumed to be retained permanently, whereas the riverine exports from exorheic catchments were added to the ocean.

We considered both weathering sources as well as non-weathering sources of P, N, Si, Fe, C and Alk to river catchments (Figure 1). These were derived, if possible, from spatially explicit models. Within the catchments, we accounted for transformations of P, N and Fe to organic matter through biological productivity on land and in rivers. These biological transformations were however not modelled dynamically in our approach. Instead, the net amounts of catchment P, N and Fe to have been transformed to organic matter on land or in rivers were derived from fixed ratios to C depending on the tDOM and POM composition

(see Figure 1). The composition of tDOM and POM and their its ratios of C, P, N and Fe were assumed to be identical for every catchment out of simplicity. The amounts of P, N and Fe that were transformed to organic matter were subtracted from their inorganic catchment pools, in order to avoid double counting. The organic carbon in tDOM and POM, which was derived from river mouth load estimations of the NEWS2 study (Seitzinger et al., 2010), was assumed to ultimately originate from terrestrial biological $CO_2$ uptake (Ludwig et al., 1998) and was therefore not subtracted from the catchment DIC pools. Additionally,

a fraction P was also assumed to have been adsorbed to Fe-P. The rivers in our approach therefore deliver terrestrial organic matter (tDOM and POM), inorganic compounds (DIP, Fe-P, DIN, DSi, DFe and DIC) and alkalinity (Alk) to the ocean (Figure 1).



**a**

| | Source | Model/Study | | Source | Model/Study |
|---|---|---|---|---|---|
| | *Weathering* | Hartmann et al. (2009) | **Si** | *Weathering* | Beusen et al. (2009) |
| **C** | *Non-Weathering* | | | | |
| | DOC (Atmospheric) | NEWS2 | | *Non-Weathering* | |
| | POC (Atmospheric) | NEWS2 | **N** | Atmospheric | Fixed ratio to P |
| | | | | Anthropogenic | |
| | *Weathering* | Hartmann et al. (2014) | **Fe** | *Weathering* | Fixed ratio to P |
| | *Non-Weathering* | | | | |
| | Fertilizer (1900) | Hart et al. (2004) | **Alk** | *Weathering* | Hartmann et al. (2009) |
| **P** | Sewage (1900) | Morée et al. (2013) | | | Goll et al. (2014) |
| | Allochtonous (1900) | Beusen et al. (2016) | | | |

**b**

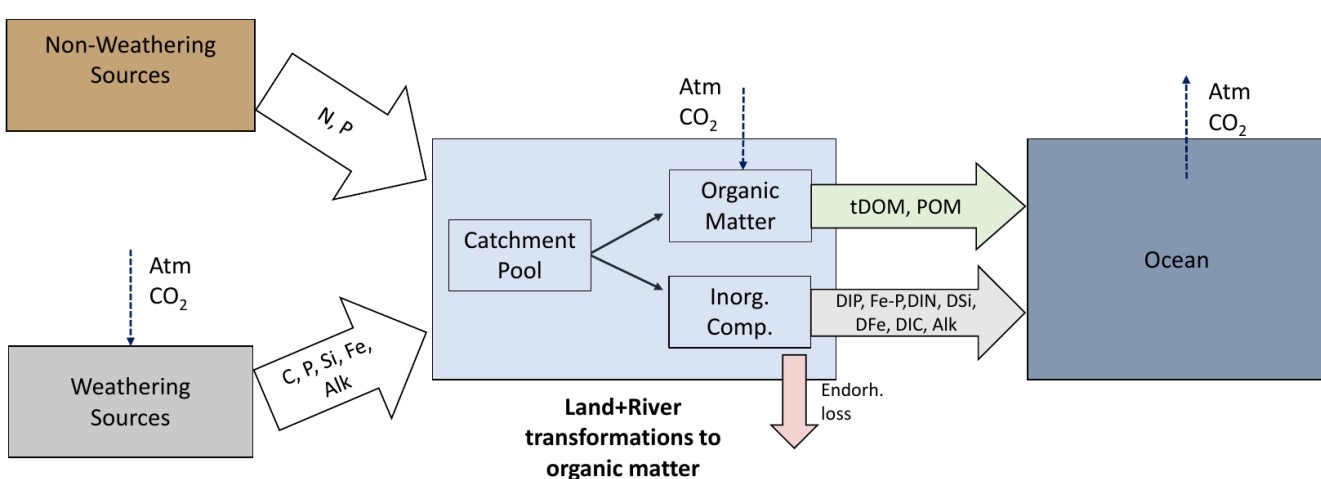

**Figure 1.** (a) Table of sources of nutrient, carbon and alkalinity inputs to the catchments and (b) scheme of origins and transformations of the catchment compounds The abbreviations are: Inorg. Comp.: Inorganic Compounds, tDOM: terrestrial dissolved organic matter, POM: particulate organic matter , DIP: dissolved inorganic phosphorus, Fe-P: Iron-bound phosphorus, DIN: dissolved inorganic nitrogen, DSi: dissolved silica, DFe: dissolved iron, DIC: dissolved inorganic carbon, Alk: alkalinity.



The surface runoff, surface temperature and precipitation data used to drive the framework submodels were obtained from output of a fully coupled MPI-ESM CMIP5 pre-industrial simulation (Giorgetta et al., 2013). We thereby used the annual 100 year means of model runoff, temperature and precipitation data computed by the MPI-ESM on a 1.875 degree Gaussian grid. We scaled the runoff data to account for the runoff model bias with regards to global estimations (Fekete et al., 2002), which

is discussed in Section 3.1.

### 2.1.1   Terrestrial dissolved and particulate organic matter characteristics

We assumed that the pre-industrial loads of tDOM and POM did not strongly differ to their present-day loads at the global scale. In the NEWS2 study (Mayorga et al., 2010; Seitzinger et al., 2010), in which anthropogenic perturbations to riverine loads were analyzed, only small changes in POC and DOC loads to the ocean were found over the 1970 to 2000 time period.

Regnier et al. (2013) suggest an anthropogenic perturbation of the total carbon flux to the ocean of around 10%, for which we did not account in this study.

The riverine loads of tDOM and POM were therefore derived from the DOC and POC river loads for the reference year 1970 (NEWS2), which were in turn determined from the models of Harrison et al. (2005) and Beusen et al. (2005). The Harrison et al. (2005) model quantifies the DOC catchment yields as a function of runoff, wetland area and consumptive water use. The

Beusen et al. (2005) describes the POC catchment yields as a function of catchment suspended solids yields, which depend on grassland and wetland areas, precipitation, slope and lithology.

The riverine organic matter exported to the ocean consisted of globally constant fractions of C, P, N and Fe. The tDOM C:P consistence was based on a C:P weight ratio of 1000:1 derived from Meybeck (1982) and Compton et al. (2000). The global total N:P mole ratio was chosen to be 16:1, which is in accordance to natural and pre-industrial estimations of previous

studies (Seitzinger et al., 2010; Beusen et al., 2016). For the P:Fe ratio, the same mole ratio as for the Fe biological uptake in the ocean was chosen ($1:3.0 \ 10^{-4}$). The total C:N:P:Fe mole ratio of tDOM was therefore $2584:16:1:3.0 \ 10^{-4}$. The C:P ratio of riverine POM is highly uncertain, but global C:P mole ratios from observational data (56-499) (Meybeck, 1982; Ramirez and Rose, 1992; Compton et al., 2000) suggest a much closer ratio to the oceanic production and export ratios (mole C:P = 122:1, Takahashi et al. (1985)) than for tDOM. Due to this and the gaps of knowledge on POM composition, we chose a C:N:P:Fe

ratio analogous to that of oceanic POM ($122:16:1:3.0 \ 10^{-4}$).

### 2.1.2   Phosphorus

#### P weathering yields

We derived the P weathering yields from a spatio-temporal model (Hartmann et al., 2014), which quantifies the P weathering release in relation to the $SiO_2$ and cation release. The model core is dependent on runoff and lithology (Hartmann and Moosdorf,





2011) and was calibrated for the extensive dataset of 381 river catchments of the Japanese Archipelago. The model was then corrected globally for temperature and soil shielding effects (Hartmann et al., 2014):

$$F_{Prelease} = \sum_{i=lith} b_{P,i} * q * F_i(T) * F_S \tag{1}$$

where $F_{Prelease}$ is the chemical weathering rate of P per area (t km$^{-2}$ yr$^{-1}$), $b_{P,i}$ is an empirical factor representing the rate of P
weathering of lithology i, q is the runoff (mm yr$^{-1}$), F(T) is a lithology-dependent temperature function and $F_S$ is a parameter for soil shielding.

The lithology types consisted of 16 lithological classes from the lithological map database GliM (Hartmann and Moosdorf, 2012), which have different weathering parameters $b_{Pi}$ as well as temperature functions $F_i(T)$. The release of P was assumed to be proportional to the release of $SiO_2$ + cations, as in Hartmann et al. (2014). The factor $b_{Pi}$ represents the chemical weathering
rate factor for $SiO_2$ + cations ($b_{SiO_2+Cat}$) multiplied with the relative P content ($b_{Prel,i}$):

$$b_{P,i} = b_{SiO_2+Cat,i} * b_{Prel,i} \tag{2}$$

The parameters $b_{SiO_2+Cat,i}$ and $b_{Prel,i}$ for each lithology i can be found in Hartmann et al. (2014).

Pre-industrial runoff model output of the MPI-ESM was used to force the model for q. The temperature correction function F(T) is an Arrhenius relationship for basic (rich in iron and magnesium) and acid (high silica content) lithological classes, with
activation energies normalized to the average temperature of the calibration catchments of the study (Hartmann et al., 2014). For acid rock lithologies, an activation energy of 60 kJ/mole was assumed, whereas for basic rock types 50 kJ/mole was used. Carbonate lithologies do not have a temperature correction due to the absence of a clear relationship to field data, as well as uncertainties in the mechanisms of a temperature effect on carbonate weathering (Plummer and Busenberg, 1982; Hartmann et al., 2014; Romero-Mujalli et al., 2018).
A soil shielding factor $F_S$ was considered due to the inhibition of weathering by certain types of soils. These soils with low physical erosion rates develop a chemically depleted thick layer, which shields them from water supply, thus preventing the maximum weathering of the soil aggregates (Stallard, 1995). Wetlands and areas with a high groundwater table have also been shown to partially inhibit weathering (Edmond et al., 1995; Goudie and Viles, 2012). The average soil shielding factor was estimated for the soils Ferrasols, Acrisols, Nitisols, Lixisols, Histosols as well as Gleysols from the Harmonized World Soil
Database (Hartmann et al., 2014). The factor used for these soils is 0.1, which was found to be the best global estimate for the calibration catchments in Hartmann et al. (2014), after having iteratively altered the parameter from 0 to 1.

**Non-weathering P sources**

Since the P cycle was already perturbed in the assumed pre-industrial state (1850) due to anthropogenic activities (Mackenzie et al., 2002; Filippelli, 2008; Beusen et al., 2016), we also accounted for P sources other than weathering ($P_{nw,catch}$). Similarly
to Beusen et al. (2016), we derived the global non-weathering source of P as the sum of fertilizer ($P_{fert,global}$), sewage ($P_{sew,global}$)





and allochthnonous P inputs ($P_{alloch,global}$):

$$P_{nw,global} = P_{fert,global} + P_{sew,global} + P_{alloch,global} \tag{3}$$

$P_{fert,global}$ was the input to the non-weathering $P_{nw,global}$ pool from the agricultural application of fertilizers, manure and organic matter (1.6 Tg P yr$^{-1}$ globally, from Hart et al. (2004)), $P_{sew,global}$ was the P input from sewage (0.1 Tg P yr$^{-1}$ globally, Morée et al. (2013)) and $P_{alloch,global}$ represented allochthnous organic matter P inputs (1 Tg P yr$^{-1}$ simplified as vegetation in floodplains in Beusen et al. (2016)). The year 1900 was used as a reference, due to previous estimates or data not being available to our knowledge. Since our framework was developed with the aim of being used in Earth System Models, we assumed soil equilibrium since this is the initial state criteria used in state-of-the-art model simulations. Therefore, P exports due to soil perturbations reported in Filippelli (2008) and Beusen et al. (2016) were not considered. The distribution of P inputs from catchment non-weathering sources was assumed to be the same as the global distribution of contemporary anthropogenic P inputs, which was derived from the NEWS2 study:

$$P_{nw,catch} = P_{nw,global} * DIP_{ant,catch} / DIP_{ant,global} \tag{4}$$

where $P_{nw,catch}$ is the non-weathering catchment P pool, whereas $DIP_{ant,catch}$ and $DIP_{ant,global}$ are anthropogenic inputs from the NEWS2 study for every catchment and their global sum, respectively.

**P river loads**

For each catchment, we estimated the total annual P inputs to the catchments ($P_{total,catch}$), as the sum of the catchment weathering yields ($P_{w,catch}$) and of non-weathering sources ($P_{nw,catch}$):

$$P_{total,catch} = P_{w,catch} + P_{nw,catch} \tag{5}$$

$P_{total,catch}$ was then fractionated into inorganic P and P contained in tDOM and POM, which was assumed to have been taken up on land or in rivers by the biology (Figure 2). After considering this net P transformation to organic matter, the remaining P was assumed to be inorganic P ($IP_{catch}$) for every catchment:

$$IP_{catch} = P_{total,catch} - P_{tDOM,catch} - P_{POM,catch} \tag{6}$$

The mole ratios of C to P in tDOM and POM were assumed to be 2584:1 (Meybeck, 1982; Compton et al., 2000) and 122:1, respectively. The IP was then fractionated into DIP and Fe-P with a ratio $r_{inorg}$ (DIP:Fe-P = 1:3) derived from the global natural P river export estimates of Compton et al. (2000):

$$DIP_{catch} = r_{inorg} * IP_{catch} \tag{7}$$

and:

$$Fe - P_{catch} = (1 - r_{inorg}) * IP_{catch} \tag{8}$$




We did not consider shale derived particulate inorganic phosphorus in this study, since it originates from physical erosion, which does not chemically transform the shale. The resulting shale-derived material is considered to not be bioavailable in rivers or the coastal ocean (Compton et al., 2000). We also did not consider in-stream retention and sinks of P, although in-stream processes might retain nutrients within river catchments (Beusen et al., 2016).

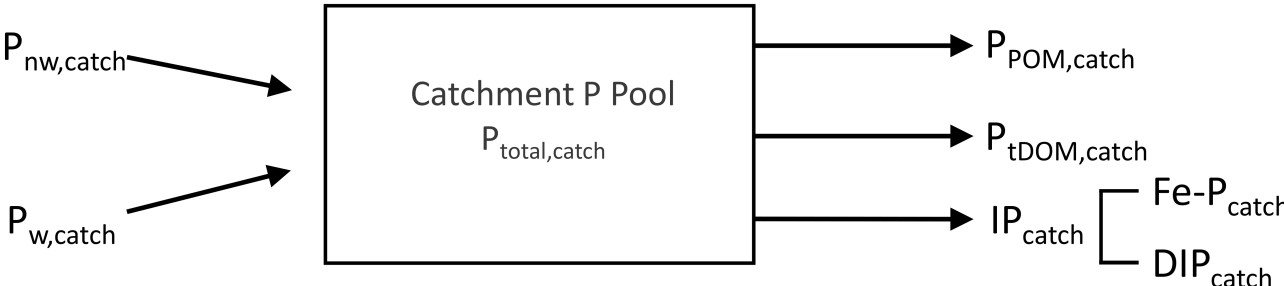

**Figure 2.** Scheme of catchment sources of P ($P_{w,catch}$ and $P_{nw,catch}$) and the export fractionation of catchment P ($P_{POM,catch}$, $P_{tDOM,catch}$, Fe-$P_{catch}$ and $DIP_{catch}$).

### 2.1.3 Nitrogen and iron

The inputs of N to riverine catchments were derived from the total P inputs to catchments at a globally fixed mole N:P ratio of 16:1 for all species, which is the same as of their oceanic removal as organic matter (Takahashi et al., 1985). While the model study of Beusen et al. (2016) suggests a total pre-industrial N:P mole ratio of 15.5:1, a synthesis of global obervations by Turner et al. (2003) suggests a higher N:P ratio for bioavailable N and P species in most major rivers. However, processes such as denitrification in river estuaries take place that remove N primarily (Meybeck, 1982; Nixon et al., 1996; Beusen et al., 2016), which exceeds the scope of our study. For Fe, we used a Fe:P mole ratio of $3.0 \ 10^{-4}$:1 to quantify Fe inputs to the catchments for all species, which is the Fe:P export ratio of organic material in the ocean biogeochemical model.

### 2.1.4 Dissolved inorganic carbon and alkalinity

The C and Alk weathering release was derived from weathering $CO_2$ uptake equations that originate from the studies of Hartmann et al. (2009) and Goll et al. (2014). Weathering reactions take up atmospheric $CO_2$ and release carbon in the form of $HCO_3^-$ during carbonate weathering:

$$CaCO_3 + CO_2 + H_2O => Ca^{2+} + 2HCO_3^- \tag{9}$$



and silicate weathering:

$$Mg_2SiO_4 + 4CO_2 + 4H_2O => 2Mg^{2+} + 4HCO_3^- + H_4SiO_4 \qquad (10)$$

The equations (9) and (10) dictate the release of 1 $HCO_3^-$ (thus 1 DIC and 1 Alk) for each mole of $CO_2$ taken up in the weathering of silicate lithologies, and that 2 $HCO_3^-$ (thus 2 DIC and 2 Alk) are released during the uptake of each mole of $CO_2$

drawn down during the weathering of carbonate lithologies.

The release equations from Hartmann et al. (2009) and Goll et al. (2014) quantify the lithology (i) dependent $HCO_3^-$ weathering release as a function of runoff (q), temperature ($F_i(T)$), soil shielding $F_S$ and a weathering parameter $b_{C,i}$:

$$F_{HCO_3^-} = \sum_{i=lith} b_{C,i} * q * F_i(T) * F_S \qquad (11)$$

Modelled pre-industrial runoff from the MPI-ESM was used for q. The parameter $b_{C,i}$ is dependent on the weathering rate

of the lithology and the composition of the lithology.

We derived the catchment Alk exports to the ocean as the $HCO_3^-$ weathered annually within the catchments, assuming conservation of Alk along the land-ocean continuum. Riverine $HCO_3^-$ is thereby considered to mainly originate from the products of silicate and carbonate weathering reactions (Amiotte Suchet and Probst, 1995; Meybeck and Vörösmarty, 1999). We did not consider additional DIC sources, for instance of $CO_2$ from respiration of organic matter in soil pore water, groundwater or in

rivers. River observational data however show that the riverine $HCO_3^-$ and total DIC mole exports rarely deviate by more than 10% (Araujo et al., 2014). Our assumptions lead to riverine exports of DIC and Alk at a mole ratio of 1:1.

### 2.1.5 Silica

To quantify the spatial distribution of Si export yields, we used the model of Beusen et al. (2009), which describes the dissolved silica (DSi) river export as:

$$F_{DSiO_2} = b_{prec} * ln(prec) + b_{volc} * volc + b_{bulk} * bulk + b_{slope} * slope \qquad (12)$$

where $F_{DSiO_2}$ is the export of DSi in Tg $SiO_2$ yr$^{-1}$ km$^{-2}$, ln(prec) is the natural logarithm of the precipitation in mm d$^{-1}$, volc is the area fraction covered by volcanic lithology (no dimension), bulk is the bulk density of the soil in Mg m$^{-3}$, slope is the average slope based on global Agro-ecological zones (FAO/IIASA) in m km$^{-1}$, and $b_{prec}$, $b_{volc}$, $b_{bulk}$, $b_{slope}$ are the estimated regression coefficients in Beusen et al. (2009). For the precipitation, we used pre-industrial model output from the MPI-ESM,

whereas the volcanic area originated from Dürr et al. (2005), the soil density from Batjes (1997) and Batjes (2002), and the average slope from the Global Agro-Ecological Zones database (FAO/IIASA). The exports were aggregated for the HD model catchments, while taking into account catchment areas. The loads that were generated by the Beusen et al. (2009) model were converted to Tg DSi loads and are given accordingly in the rest of our study. We also neglected the land-ocean export of particulate silica physically eroded from land.



## 2.2 Ocean Model Setup

### 2.2.1 Ocean Biogeochemistry

The Max Planck Institute Ocean Model (MPIOM), which was used to simulate oceanic physics, is a z coordinate global circulation model that solves primitive equations under the hydrostatic and Boussineq approximation on a C-grid with a free

surface (Jungclaus et al., 2013). The grid configuration used was GR15, which consists of a bipolar grid with poles over Antarctica and Greenland, and a grid resolution of about 1.5 degrees. Vertically, the configuration consists of 40 uneven spaced layers, with increasing thicknesses at greater depths. The surface boundary data, as well as river freshwater model inputs originate from the Ocean-Model-Intercomparision-Project (OMIP, Röske (2006)).

We used the Hamburg Ocean Carbon Cycle model (HAMOCC) to simulate the major biogeochemical processes that affect

carbon as well as nutrients in the ocean. The standard model used was an extension of the model described in Ilyina et al. (2013), which is explained in Mauritsen et al. (2018). The changes were made to incorporate dynamical nitrogen fixation through cyanobacteria (Paulsen et al., 2017), to follow recommendations from the OMIP protocol (Orr et al., 2017) and to correct errors in the model. The model represents processes in the water column, sediment, as well as air-sea exchange fluxes. All biogeochemical tracers found in the water column are thereby fully advected, mixed and diffused by the flow fields of MPIOM.

The biogeochemistry of the water column includes both organic as well as inorganic carbon cycle processes. The dynamics for the organic carbon cycle are based on a NPZD (nutrients, phytoplankton, zooplankton and detritus) model approach, which was extended to incorporate the compartments of oceanic dissolved organic material (DOM) and cyanobacteria (Six and Maier-Reimer, 1996; Ilyina et al., 2013).

A constant Redfield ratio (C:N:P = 122:16:1, Takahashi et al. (1985)) and Fe:P ratio of $3.0 \ 10^{-4}$:1 dictate the composition of

oceanic organic matter. The phytoplankton growth follows Michaelis-Menten kinetics as a function of temperature, light and nutrient availability. The phytoplankton produce opal when dissolved silica is available, and calcium carbonate ($CaCO_3$) when dissolved silica is depleted. The $CaCO_3$ and opal thereby sink at constant rates.

The mortality of the phytoplankton and its exudation as DOM, as well as the zooplankton grazing of phytoplankton is included. DOM can also be formed due to sloppy feeding. POM is formed from dead cells of phytoplankton and zooplankton,

as well as fecal pellets from zooplankton activity. Both oceanic DOM and POM are advected according to the ocean physics, and the POM also sinks as a function of depth (Martin et al., 1987). Aerobic remineralization takes place when the oxygen concentration is above a threshold oxygen concentration, whereas at low enough oxygen concentrations, denitrification and sulfate reduction can take place.

The inorganic chemistry is based on Maier-Reimer and Hasselmann (1987), with adjustments in the calculation of chemical

constants as described in the OMIP protocol (Orr et al., 2017). Total DIC and total Alk are thereby prognostic tracers from which the carbonate species are determined diagnostically.

HAMOCC also contains a 12 layer sediment module where the same remineralization and dissolution processes as in the water column take place for the solid sediment constituents (Heinze et al., 1999). The sediment consists of a fraction of pore water, which contains dissolved inorganic compounds (e.g. DIC and DIP). POM, $CaCO_3$ and opal fluxes from the water



column are deposited to the top sediment layer. There is a diffusive inorganic compound flux at the water sediment-water column interface and a particulate flux from the bottom layer to a diagenetically consolidated burial layer.

The model considers the gas exchange of $CO_2$, $O_2$ and $N_2$ at the ocean-atmosphere interface. Since we model a pre-industrial state of equilibrium in this study, we used constant atmospheric concentrations for $CO_2$ of 278ppmV (Etheridge et al., 1996).

### 2.2.2 Treatment of the river loads in the ocean biogeochemistry model

The biogeochemical riverine loads were added to the ocean surface layer in HAMOCC constantly over the whole year . The locations of major river mouths were corrected manually on a case to case basis for large rivers in order to reproduce the same locations as the freshwater inputs from OMIP. The riverine freshwater discharge on the other hand varied intra-annually according to the prescribed OMIP freshwater loads (Röske, 2006).

The dissolved riverine inorganic compounds (DIC, DIP, DIN, DSi, DIC, DFe, Alk) were added to the model in their dissolved species pools in HAMOCC. We added 80% of P contained in the riverine Fe-P to the oceanic DIP pool, in order for the amount of bio-available Fe-P to be comparable with the given range in Compton et al. (2000) (1.1-1.5 Tg P $yr^{-1}$). The rest of the Fe-P pool was considered to be unreactive in the ocean and was eliminated. The riverine POM was added to the oceanic POM pool in the ocean model, since we assumed its consistence to be the same (P:N:C:Fe mole ratio of 1:16:122:3.0 $10^{-4}$).

For tDOM, we extended HAMOCC with a new tracer that was characterized with a C:N:P:Fe mole ratio of 2584:16:1:3.0 $10^{-4}$ (Meybeck, 1982; Compton et al., 2000). tDOM was mineralized as a function of the tDOM concentration at a rate $k_{rem,tDOM}$ and also of an oxygen limitation factor ($\Gamma_{O2}$), which decreases the maximum potential remineralization rate as a function of the oxygen concentration:

$$dtDOM/dt = k_{rem,tDOM} * tDOM * \Gamma_{O2} \tag{13}$$

Since a large fraction of tDOM delivered by rivers is already strongly degraded, it is to a certain extent resistent to microbial degradation (Ittekkot, 1988; Vodacek et al., 2003). We consequently assumed a slightly slower remineralization rate of tDOM ($k_{rem,tDOM}$) compared to oceanic DOM (0.003 versus to 0.008 $d^{-1}$ for oceanic DOM), which is within the tDOM degradation range provided in Fichot and Benner (2014) for the Louisiana shelf (0.001-0.02 $d^{-1}$). The oxygen limitation function used ($\Gamma_{O2}$) was analogous to that of the oceanic DOM which is described in Mauritsen et al. (2018).

### 2.2.3 Pre-industrial ocean biogeochemistry model simulations

We performed a reference simulation (REF), where the burial loss of biogeochemical tracers was compensated by a global homogeneous flux to the surface ocean, as in the standard configuration of the model. This flux served to maintain a stable ocean state. Thus in REF, the delivery of biogeochemical inputs were added directly in the open ocean while bypassing the coastal ocean, and were fully constrained by the loss of the sediment layer. Our simulation RIV however included riverine loads of inorganic compounds (DIP, Fe-P, DIN, DSi, DFe, DIC, Alk) and organic matter (tDOM, POM), which were constrained by our framework and added to their geographical river mouth locations.



REF was simulated for 5000 years to quasi-equilibrium. RIV was simulated for 4000 years first including both the water column and sediment model components, then for 10'000 years in a model version simulating only sediment processes. In these 10'000 years, the inputs to the sediment were the means of CaCO$_3$, POM and opal from the water column of the last 100 years of the standard simulation. The sediment was then coupled back to the ocean water column, with a simulation performed for 2000 more years in the full model version. For the analysis of the resulting ocean biogeochemistry, we used 100 year means of model output.

## 2.3   Definitions of coastal regions for analysis

To investigate the impacts of riverine exports on coastal regions, we chose 10 coastal regions characterized by shallow continental shelves and high riverine loads that cover a variety of latitudes (Table 1). The shelves were defined to have depths shallower than 250m. The cutoff sections perpendicular to the coast were done according to MARgins and CATchements Segmentation (MARCATS) (Laruelle et al., 2013), except for the Beaufort Sea, Laptev Sea, North Sea and Congo shelf, where we used the COastal Segmentation and related CATchments (COSCATs) definitions (Meybeck et al., 2006) due to the vastness of their MARCATS segmentations.

**Table 1.** Comparison of the surface areas [10$^9$ m$^2$] of selected coastal regions with depths of under 250m in the ocean model setup and in segmentation approaches. The comparisons were done with the MARCATS (Laruelle et al., 2013) or COSTCATs (Meybeck et al., 2006). The shelf classes were defined as in Laruelle et al. (2013).

| Coastal Regions | Major Rivers | Model Area | MARCATS/COSCAT Area | Shelf Class |
|---|---|---|---|---|
| 1. Beaufort Sea (BS) | Mackenzie | 269 | 274 | Polar |
| 2. Laptev Sea (LS) | Lena | 397 | 326 | Polar |
| 3. North Sea (NS) | Rhine | 499 | 871 | Marginal Sea |
| 4. Sea of Okhotsk (OKH) | Amur | 245 | 992 | Marginal Sea |
| 5. East China Sea (CSK) | Yangtze, Huang He | 731 | 1299 | Tropical |
| 6. Bay of Bengal (BEN) | Ganges | 245 | 230 | Indian Margin |
| 7. Southeast Asia (SEA) | Mekong | 1795 | 2318 | Indian Margin |
| 8. Tropical West Atlantic (TWA) | Amazon, Orinoco | 448 | 517 | Tropical |
| 9. Congo shelf (CG) | Congo | 53 | 38 | Tropical |
| 10. South America (SAM) | Paraná | 1553 | 1230 | Subpolar |



## 3 Global weathering

### 3.1 Runoff, precipitation and temperature patterns

The weathering yields provided in this study are dependent on the MPI-ESM pre-industrial spatial representation of surface runoff, surface temperature and precipitation (Figure 3a,b,c). For the modelled MPI-ESM CMIP5 100 year average, we observe

high modelled precipitation and temperature in the tropics, whereas in the subtropics and temperate zones the patterns are much more spatially variable. Above the Arctic Circle, the model shows only moderate precipitation and cold annual temperatures. Spatial patterns of high precipitation are usually also reproduced in the spatial patterns of runoff. This is however not always the case due to evapotranspiration. For instance, in northern North America, northern Europe and Siberia, a pattern of high runoff can be observed despite relatively low precipitation in these regions, which is most likely due to these regions having

low vegetation densities, as well as due to the negative temperature dependency of evaporation, both of which result in lower evapotranspiration rates than in lower latitudes.

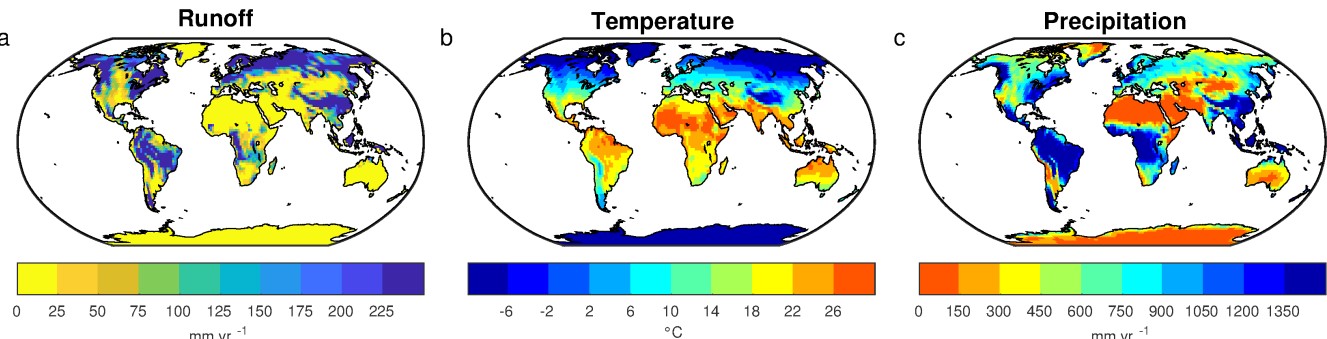

**Figure 3.** Modelled pre-industrial (a) surface runoff [mm a$^{-1}$], (b) surface temperature[°C] and (c) precipitation [mm a$^{-1}$] annual means.

Previous work by Goll et al. (2014) describes the MPI-ESM performing well when estimating surface temperatures at single grid cells with regards to observations. The global precipitation is slightly higher than is reported in the Precipitation Climatology Project (GCPC) (Adler et al., 2003), which is discussed along with spatial biases of the precipitation in Stevens

et al. (2013). Most notably, the precipitation is too strong over extratropical land surface and too little over tropical land surface. Runoff on the other hand is less well reproduced globally. For the given time period of the CMIP5 simulation, the global runoff is 23,496 km$^3$ yr$^{-1}$. This is significantly lower than the global runoff estimations of 36,600-38,300 km$^3$ yr$^{-1}$ (Fekete et al., 2002; Dai and Trenberth, 2002). The difficulty of representing several processes that control the runoff, such as evapotranspiration and condensation, is also reflected in the global runoff means of other Earth System Models, which range from 23,000 to 42,500

km$^3$ yr$^{-1}$ (Goll et al., 2014). The spatial patterns in the CMIP5 simulation are however comparable with the mean annual runoff patterns reported in Fekete et al. (2002), with high surface runoff observed in the Amazon basin, West Africa, Indo-Pacific Islands, Southeast Asia, eastern North America, Northern Europe as well as in Siberia. Due to the strong underestimation of the model regarding the runoff in relation to the combined runoff mean of Fekete et al. (2002) and Dai and Trenberth (2002),



we conclude that a scaling factor of 1.59 is necessary to produce runoff plausibly at the global scale. The global runoff from OMIP which provides freshwater to the ocean model, is on the other hand more plausible (32,542 km$^3$ yr$^{-1}$), and was therefore not scaled.

### 3.2 Global weathering yields and their spatial distribution

The described weathering release models provide global means for weathering release rates of P, Si, DIC and Alk (Table 2), as well as their spatial distributions (Figure 4).

For P, which is derived from runoff and temperature data (Hartmann et al., 2014), the global release is 1.34 Tg P yr$^{-1}$ when considering the runoff scaling factor of 1.59, whilst it is 0.84 Tg P yr$^{-1}$ when omitting the scaling factor. The release calculated when compensating the model's runoff underestimation fits within the range found in published literature of 0.8 - 4 Tg P yr$^{-1}$

(Compton et al., 2000; Wang et al., 2010; Goll et al., 2014; Hartmann et al., 2014), while it is on the lower end of this range without the scaling factor. The Hartmann et al. (2014) and Goll et al. (2014) estimates (1.1 and 0.8-1.2 Tg P yr$^{-1}$, respectively) originate from the same weathering model as is used here. The 1.9 Tg P yr$^{-1}$ reported in Wang et al. (2010) was on the other hand constructed by upscaling measurement data points. In a further study, Compton et al. (2000) provide a quantification of the prehuman phosphorus cycle while distinguishing between land-ocean fluxes from shale-erosion as well as from weathering.

Thereby, the total pre-human P riverine flux which originates from weathering is given by averaging P species concentrations from unpolluted rivers and multiplying them with global runoff estimates, yielding an estimate of 2.5 - 4 Tg P yr$^{-1}$. Both estimates originating from upscaling from river measurements are therefore higher than the P weathering flux provided in the modelling approach in this study, in Goll et al. (2014) and in Hartmann et al. (2014), which suggests further effort might be needed to better constrain the P weathering release.

Deriving the Si export from the Si model (Beusen et al., 2009) forced with MPI-ESM precipitation output amounts to a DSi global yield of 168 Tg Si yr$^{-1}$. This is within the range estimated by Beusen et al. (2009) (158-199 Tg Si yr$^{-1}$), who used the same model while using present-day observational data to drive the model for precipitation. Our estimate is also comparable with the 173 Tg Si yr$^{-1}$ estimate provided by Dürr et al. (2011). In a synthesis of the global oceanic silica cycle, Tréguer and De La Rocha (2013) conclude that rivers deliver around 200 Tg DSi to estuaries annually.

The modelled DIC and Alk release amounts to 374 Tg C yr$^{-1}$. By extrapolating from measurement data of 60 large river catchments, Meybeck (1982) suggests that the DIC export to the ocean is around 380 Tg yr$^{-1}$ and originates directly from weathering. Further modelling studies also provide similar estimates of 260 to 300 Tg yr$^{-1}$ (Berner et al., 1983; Amiotte Suchet and Probst, 1995). Mackenzie et al. (1998) provide an inorganic carbon flux of 720 Tg C yr$^{-1}$ in a conceptual model that considers mass balance. Since this estimate considers both particulate and dissolved inorganic carbon fluxes, the DIC flux is however significantly lower. Accounting for a particulate inorganic carbon flux of around 170 Tg C yr$^{-1}$ (Meybeck and

Vörösmarty, 1999), the value provided by the Mackenzie et al. (1998) study would result in a global DIC load of around 550 Tg C yr$^{-1}$.

The atmospheric $CO_2$ drawdown induced by weathering is directly related to the release of $HCO_3^-$ since silicate weathering draws down 1 mole of $CO_2$ per mole $HCO_3^-$, and carbonate weathering draws down 0.5 mol of $CO_2$ per mole $HCO_3^-$ released





(Eq. (9) and Eq. (10)). While we provide a modelled $CO_2$ drawdown of 280 Tg C yr$^{-1}$ induced by weathering, previously estimated drawdown fluxes are suggested in the range of 220 and 440 Tg C yr$^{-1}$ (Gaillardet et al., 1999; Amiotte Suchet et al., 2003; Hartmann et al., 2009), Goll et al. (2014)). As for the DIC release, the modelled estimate is therefore comparable to what was previously suggested in published literature. The results imply that of the 374 Tg C yr$^{-1}$ DIC released by weathering, 280 Tg C drawdown yr$^{-1}$ originates from atmospheric drawdown, while the rest originates from the weathering of the carbonate lithology (94 Tg C yr$^{-1}$).

**Table 2.** Weathering release of P, Si, DIC and Alk, as well as $CO_2$ drawdown, quantified by the combination of models used in this study in comparison to published literature estimates.

| Species | Modelled weathering flux | Estimates | Source |
|---|---|---|---|
| P release [Tg yr$^{-1}$] | 1.34 | 1.2 - 1.8 | Wang et al. (2009); Hartmann et al. (2014) |
| Si release [Tg yr$^{-1}$] | 168 | 158 - 200 | Beusen et al. (2009); Dürr et al. (2011); Tréguer and De La Rocha (2013) |
| DIC release [Tg yr$^{-1}$] | 374 | 260 - 550 | Berner et al. (1983); Amiotte Suchet et al. (1995); Mackenzie et al. (1998); Hartmann et al. (2009) |
| Alk release [10$^{12}$ mole yr$^{-1}$] | 18.8 | - | - |
| *CO$_2$ drawdown [Tg yr$^{-1}$]* | *280* | *220 - 440* | *Gaillardet et al., 1999; Amiotte-Suchet et al., 2003 Hartmann et al., 2009, Goll et al., 2014* |

The spatial distributions of the modelled weathering release of nutrients, carbon and alkalinity (Figure 4) show strong agreement with Hartmann et al. (2009), Hartmann et al. (2014) as well as Beusen et al. (2009), where the same models were used, but were forced with observational datasets. Generally, the patterns mostly follow the runoff and precipitation patterns, whereas the lithologies play a secondary role in explaining the spatial variability patterns. This is most likely due to runoff and precipitation having much stronger gradients of variation than the nutrient and carbon content in the different lithologies. Hartmann et al. (2014) suggest that in the case of the P release model, the temperature effect on the spatial variability is of similar magnitude as the P-content of the lithologies.

Furthermore, we observe hotspots that contribute disproportionally to the nutrients and carbon release, as is also suggested in Hartmann et al. (2009) and Hartmann et al. (2014). The spatial distributions of the weathering release yields indicate the dominance of the Amazon, Southeast Asia as well as Northern Europe and Siberia as strong sources of weathering, despite substantial soil shielding in these regions. The Amazon and Southeast Asian regions have also been identified in other studies as regions of strong carbon yields due their the wet and warm climate, as well as due to their lithology (Amiotte Suchet and Probst, 1995). The Southeast Asian islands are also areas of weathering rates significantly higher than average, due to the combination of the warm and wet climate in the region, as well as the regional abundance of volcanic and carbonate lithologies, which (Gaillardet et al., 1999; Hartmann and Moosdorf, 2011). The northern hemispheric hotspots in eastern North America



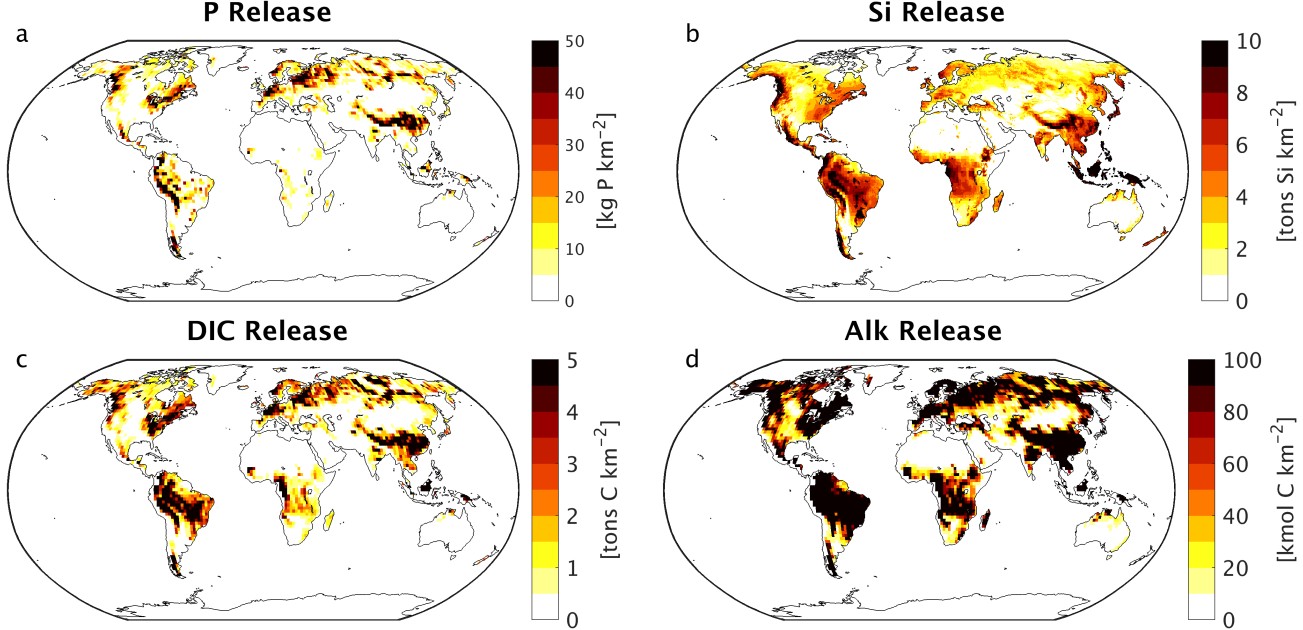

**Figure 4.** Weathering release rates of (a) P [kg P km$^{-2}$], (b) Si [tons Si km$^{-2}$], (c) DIC [tons C km$^{-2}$] and (d) Alk [kmol km$^{-2}$].

and western Europe can be explained by the carbonate lithology in these regions. While the weathering and export rates over the 6 major Arctic catchments do not appear as high as other hotspots such as in the Amazon or in Southeast Asia, the vastness of the Arctic catchments could still lead to large exports to the Arctic Ocean, and the shallowness of the Arctic shelf could result in stronger implications of the exports on the ocean carbon cycle of the Arctic Ocean (Le Fouest et al., 2013).

## 5  4   Pre-industrial rivers loads

In this section, we report the modelled riverine loads at the global scale, as well as their regional distributions, while comparing them to a wide range of estimates made for pre-industrial riverine loads until now. We also compare the modelled loads of P and N to contemporary estimates (NEWS2, reference year 1970), in order to grasp the magnitude of their anthropogenic perturbations. Since we focus on the implications of land-sea fluxes for the ocean carbon cycle in this study, our validation
10  analysis revolves around the nutrient, carbon and alkalinity loads at the river mouths, omitting validation of river concentrations further upstream.

### 4.1   Global loads in context of published estimates

The cumulative pre-industrial catchment loads amount to 3.7 Tg P yr$^{-1}$, 27 Tg N yr$^{-1}$, 158 Tg Si yr$^{-1}$ and 603 Tg C yr$^{-1}$ delivered to the ocean globally. We thereby estimate the retention of 0.3 Tg P, 2.2 Tg N, 10 Tg Si and 19 Tg C through endorheic



catchments, which do not discharge into the ocean. These values do not account for particulate inorganic compounds and in the case of P, the fraction of Fe-P assumed to not be desorbed in the ocean was also subtracted (0.2 Pg P yr⁻¹).

**Table 3.** Comparison of modelled global riverine loads (Model. global loads) with previous estimates [Tg yr⁻¹], except for the 1970 POP estimate, which includes all particulate P compounds from the NEWS2 study. The total loads thereby exclude particulate inorganic loads. [1] Compton et al. (2000), [2] Beusen et al. (2016), [3] NEWS2 (Seitzinger et al., 2010), [4] Beusen et al. (2009), [5] Dürr et al. (2011), [6] Tréguer and De La Rocha (2013), [7] Green et al. (2004), [8] Jacobson et al. (2007), [9] Meybeck and Vörösmarty (1999), [10] Resplandy et al. (2018), [11] Regnier et al. (2013), [12] Berner et al. (1983), [13] Amiotte Suchet and Probst (1995), [14] Mackenzie et al. (1998), [15] Cai (2011).

| Species | Model. global load | Estimates and Source | Species | Model. global load | Estimates and Source |
|---|---|---|---|---|---|
| P [Tg P yr⁻¹] | 3.7 | 4 - 4.8 (prehuman)[1] | N [Tg N yr⁻¹] | 27 | 19-21 (pre-industrial)[27] |
|  |  | 2 (1900)[2] |  |  | 37 (1970)[3] |
|  |  | 7.6 (1970)[3] |  |  |  |
| *DIP* | *0.5* | *0.3 - 0.5 (prehuman)[1]* | *DIN* | *3.4* | *2.4 (pre-industrial)[7]* |
|  |  | *1.1 (1970)[3]* |  |  | *14 (1970)[3]* |
| *DOP* | *0.1* | *0.2 (prehuman)[1]* | *DON + PON* | *24* | *19 (pre-industrial)[7]* |
|  |  | *0.6 (1970)[3]* |  |  | *23 (1970)[3]* |
| *POP* | 2.2 | *0.9 (prehuman)[1]* |  |  |  |
|  |  | *5.9 (1970)[3]* | C [Tg C yr⁻¹] | 603 | 450 - 950 (present-day)[8,9,10,11] |
| *Fe-P* | *0.8* | *1.1 - 1.5 (prehuman)[1]* | *DIC* | *366* | *260 - 550 (present-day)[12,13,14]* |
|  |  | *-* | *DOC* | *134* | *130 - 250 (present-day)[3,9,15]* |
|  |  |  | *POC* | *103* | *100 - 140 (present-day)[3,9]* |
| DSi [Tg Si yr⁻¹] | 158 | 158 - 200 (present-day)[4,5,6] |  |  |  |

The magnitudes of modelled land-ocean exports of P, N, Si and C, as well as their fractionations, largely agree with the wide range of estimates found in published literature (Table 3).

5      For instance, the modelled global P loads fluxes are close to the higher P export estimate range of 4 - 4.7 Tg P yr⁻¹ reported in Compton et al. (2000), which was constructed by upscaling natural river catchment P concentrations to the global river freshwater discharge (thus prehuman). A recent modelling study by Beusen et al. (2016), which takes into account a more complex retention scheme within rivers than is done in our framework, suggests a lower load of 2 Tg P yr⁻¹ for year 1900. The 1970 estimate (7.6 Tg P yr⁻¹) provided by the NEWS2 study, which considers substantial anthropogenic inputs, nevertheless

10    suggests a steep 20th century increase in the global P flux to the ocean for all three cases.

     The modelled DIP export to the ocean (0.5 Tg P yr⁻¹) is at the top range of prehuman estimates (0.3 - 0.5 Tg yr⁻¹) and well below 1970 estimates (1.1 Tg P yr⁻¹). A direct fractionation of the global P flux to DIP, DOP and POP is not provided in the Beusen et al. (2016) study. We estimate similar global loads of DOP as Compton et al. (2000) (around 0.1 and 0.2 Tg P yr⁻¹). The modelled DOP value is also much lower than the 1970 value (0.6 Tg P yr⁻¹), which was also strongly anthropogenically





perturbed for 1970 (Seitzinger et al., 2010). The modelled POP global load is larger than the estimate of Compton et al. (2000), which could be due to the POM C:P ratio of 122:1 chosen in our study. Strong uncertainties exist in the global C:P ratios for riverine POM, with Meybeck (1993) suggesting a weight ratio of around 57 C:P, whereas Ramirez and Rose (1992) estimate a ratio of around 500. The particulate P load given in the NEWS2 study is vastly higher than the POP load modelled in our study, but a large fraction of the estimate is likely to be directly shale-derived particulate inorganic P and thus biologically unreactive in the ocean. The modelled Fe-P (1.0 Tg P yr$^{-1}$) is slightly below the range estimated in Compton et al. (2000) (1.5 - 3.0 Tg P yr$^{-1}$). However, the assumed reactive fraction of the Fe-P loads (0.8 Tg P yr$^{-1}$) here is close to how much P is suggested to be desorbed in the coastal ocean in Compton et al. (2000) (1.1-1.5 Tg P yr$^{-1}$).

Despite our simplified assumption of N loads being coupled to P loads, the modelled global N load is also situated within the pre-human and contemporary land-ocean N loads given in the modelling study of Green et al. (2004) (21 and 40 Tg N yr$^{-1}$ respectively). The modelled annual DIN load (3.4 Tg N) is slightly higher than the prehuman load given in the Green et al. (2004) study (2.4 Tg N yr$^{-1}$). In Beusen et al. (2016), the global pre-industrial N load is suggested to be lower (19 Tg N yr$^{-1}$) due to in-stream retention and removal.

The modelled global load of DSi is 158 Tg Si yr$^{-1}$, which is at the lower boundary of the range of present-day estimates of 158-200 Tg Si yr$^{-1}$. We thereby assume that the change in the global DSi load over the 20th century is small, and therefore compare our pre-industrial estimate with present-day estimates from published literature. The NEWS2 study used the same Beusen et al. (2009) silica export model forced with present-day observational precipitation data, Dürr et al. (2011) and Tréguer and De La Rocha (2013) upscaled discharge weighted DSi concentrations at river mouths. Substantial amounts of particulate silica are suggested to be delivered to the ocean, yet it is not clear how much is dissolvable and biologically available (Tréguer and De La Rocha, 2013). Another point of uncertainty is the increase in river damming during the 20th century, which might have strongly increased the global silica retention in present-day rivers (Ittekkot et al., 2000; Maavara et al., 2014). The pre-industrial loads therefore might have been higher than for the present-day, but the implications of damming on the retention of biogeochemical compounds escape the scope of this study.

The modelled total C, DIC, DOC and POC fluxes are within, albeit on the lower side of the present-day estimate ranges shown in Table 3. While the carbon retention along the land-ocean continuum might have increased, enhanced soil erosion through changes in land use might have also increased the carbon inputs to the freshwater systems, leaving question marks on the magnitude of the net anthropogenic perturbation (Regnier et al., 2013; Maavara et al., 2017). The agreement of the DOC and POC loads with estimates is not surprising, since they originate directly from the NEWS2 study, which already validated the global loads extensively.

The large spread found in the literature estimates regarding all species points towards difficulties in constraining pre-industrial river fluxes. Even for the present-day, Beusen et al. (2016) note large differences between the outcomes of their study and the previous global modelling study NEWS2. Upscaling approaches, on the other hand, are often based on data collected by Meybeck (1982) for pre-1980s without taking into account more recent river measurement data. They also rely on the assumption of a linear relationship between river runoff and river specie loads. While we acknowledge a certain degree of uncertainty in the numbers provided in this study, and that possibly significant riverine processes such as in-stream retention





are omitted, the modelling approach chosen nevertheless leads to pre-industrial global river loads that are in line with what was suggested previously and to a framework that could be used within state-of-the-art Earth System Models.

## 4.2 Spatial load distribution and identified hotspots

Riverine loads of the major catchments show similar spatial distributions with regards to areas of high weathering rates (Figure 5), with warm and wet regions yielding the largest river exports to the ocean. We observe large differences between the northern and southern hemispheres. The northern hemisphere accounts for an annual total carbon input of 404 Tg C to the ocean, vastly dominating the global loads (67% of total global C). The northern hemisphere also contributes overproportionally to the oceanic nutrient supply (69% for DIP and DIN, 60% for DSi). The dominance of northern hemispheric riverine C land exports to the ocean is also reported in Aumont et al. (2001) and Resplandy et al. (2018).

We observe several regions of disproportionate contributions to global riverine loads. For one, rivers that drain into the tropical Atlantic consist of a major fraction of the global biogeochemical oceanic supply. This is due to major rivers of the South American continent debouching into the ocean basin, as well as considerable exports provided by the west African Volta, Congo and Niger rivers (Table 4). According to our framework, the seven largest rivers unloading in the region (Orinoco, Amazon, São Francisco, Paraíba do Sul, Volta, Niger and Congo) amount to a total yearly carbon flux of 123 Tg C (58 Tg DIC, 44 Tg DOC, 21 Tg POC), which consists of around 20% of the global carbon riverine exports. These regional carbon loads agree very well with estimated values derived from monthly river discharge and carbon concentrations data in Araujo et al. (2014) (53 Tg DIC, 46 Tg DOC). In terms of catchments, the pre-industrial Amazon river provides the largest inputs of biogeochemical tracers to the ocean in the region (modelled annual loads of 0.07 Tg DIP, 0.5 Tg DIN, 15.2 Tg DSi, 33.2 Tg DIC , 28.3 Tg DOC Tg, 17.1 Tg POC). Present-day data from Araujo et al. (2014) suggests annual Amazon river loads of 0.22 Tg DIP, 17.8 Tg DSi, 32.7 Tg DIC, 29.1 Tg DOC. Since DIP loads are suggested to have strongly increased due to anthropogenic inputs (NEWS2, Seitzinger et al. (2010)), the pre-industrial and present-day difference in the DIP loads is plausible. For the other tropical Atlantic catchments, the modelled DIC loads are close to estimated values for the Congo, Orinoco and Niger, but are overestimated for the smaller catchments of the Paraíba, Volta and São Fransisco. The DIP loads of the tropical Atlantic catchments tend to show much lower values with regards to present-day data, suggesting the realistic increase in the region's DIP loads from pre-industrial exports of 81.8 $10^9$ g P $yr^{-1}$ to present-day loads of 276 $10^9$ g P $yr^{-1}$ due to anthropogenic inputs.

Although less significant in terms of global loads, the major Arctic rivers (Yukon, Mackenzie, Ob, Lena, Yenisei) provide a large carbon supply, which consists of a dominant fraction of DIC, to the shallow basin of the Arctic Ocean. The Arctic rivers thereby provide 37.5 Tg DIC (10% of global DIC), 14.4 Tg DOC (11 % of global DOC) and 4.4 Tg POC annually to the Arctic Ocean. The total C loads of the Arctic therefore amount to 56 Tg C $yr^{-1}$, thus 9% of global C loads. The DIC, DOC and POC load levels are comparable to estimates of 29 Tg DIC $yr^{-1}$ (Tank et al., 2012), 17 Tg DOC $yr^{-1}$ (Raymond et al., 2007) and 5 Tg POC (Dittmar and Kattner, 2003). Total Arctic DIP loads (40.8 $10^9$ g P $yr^{-1}$) derived from our modelling approach are slightly higher with regards to published literature estimates of 35.8 $10^9$ g P $yr^{-1}$. DIP inputs from anthropogenic sources are considered to be small for Arctic catchments (NEWS2), which explains why the modelled pre-industrial DIP loads are of





comparable magnitudes to observed DIP loads for the present-day. River loads originating from weathering models (DIC and DIP loads) thereby show a slight overestimation in the Arctic with regards to published literature estimates. Since both of these modelled exports mostly originate from weathering models due to the low anthropogenic contributions to Arctic catchment riverine loads, the runoff correction of the P and DIC weathering release models of 1.59 might be too high for this region.

5     Southeast Asian rivers also provide large exports of biogeochemical tracers to the ocean. The Huang He, Brahmaputra-Ganges, Yangtze, Mekong, Irrawaddy and Salween, which have catchment areas characterized by warm and humid climates, provide 92.4 Tg C yr$^{-1}$ to the ocean (15% of global C loads). We observe vastly elevated DIP levels for the present-day estimates with regards to our pre-industrial modelled levels. The NEWS2 data suggests a strong present-day perturbation of the DIP loads due to anthropogenic inputs to the region's catchments, which can plausibly explain these differences.

10    The Indo-Pacific Islands have been identified as a region with much higher weathering yields than average Hartmann et al. (2014). Although this region only accounts for around 2% of the global land surface, it provides 7% (39 Tg) of C and in particular 10% (10 Tg C) of the global POC delivered to the ocean annually, making the region a stronger land source of POC than the entire Arctic basin. This implies that POM mobilization through soil erosion is a substantial driver of land-sea carbon exports in the region, in addition to weathering.





**Figure 5.** Modelled dissolved annual river loads of DIP (a), DSi (b), DIC (c), Alk (d), DOM (e) and POM (f).

## 4.3 Exports to chosen coastal regions

With respect to the 10 shallow shelf regions chosen for this study (Table 1), the catchments of the low latitude regions (5. CSK, 6.BEN, 7.SEA, 8.TWA, 9.CG) provide substantially more carbon and nutrients to the coastal ocean than the high latitude regions (Figure 6), although the differing size of the coastal regions and of their drainage catchments might play a strong role in explaining these differences. The tropical West Atlantic (8.TWA) has the largest input of biogeochemical tracers due to the Amazon and Orinoco rivers being a dominant source of nutrients and carbon. In the tropical regions of the Bay of Bengal (6.BEN) and Southeast Asia (7.SEA), the fraction of carbon delivered as POC is substantially higher than for the rest of the regions (Figure 6b). Furthermore, in the high latitude regions (1.BS, 2. LS, 3.NS, 10.SAM), the DIC loads are the major source



**Table 4.** Regional hotspot of C loads [Tg C yr[-1]] and DIP loads [10[9] g P yr[-1]] compared with regional estimates: [1] Araujo et al. (2014), [2] Bird et al. (2008), [3] Tank et al. (2012), [4] Raymond et al. (2007), [5] Dittmar and Kattner (2003), [6] Le Fouest et al. (2013), [7] Li and Bush (2015), [8] Yoshimura et al. (2009), [9] Tao et al. (2010), [10] Seitzinger et al. (2010). Modelled DIP is from our approach to represent pre-industrial fluxes, whereas the DIP literature estimates are from present-day data and are strongly affected by anthropogenic perturbations.

| Hotspots | Modelled | | | | Estimates | | | |
|---|---|---|---|---|---|---|---|---|
| | DIC | DOC | POC | DIP | DIC | DOC | POC | DIP |
| *Tropical Atlantic* | | | | | | | | |
| Amazon | 33.2 | 28.2 | 17.1 | 73 | 32.7[1] | 29[1] | 6.1[2] | 221[1] |
| Congo | 9 | 5.6 | 1.2 | 2.3 | 13[1] | 10.58[1] | 2.0[2] | 18 [1] |
| Paraíba | 2.4 | 0.5 | 0.4 | <0.1 | 0.3[1] | 0.10[1] | - | 0.6[1] |
| Volta | 3 | 0.5 | 0.5 | 0.1 | 0.1[1] | 0.13[1] | - | 7.0[1] |
| Niger | 1.1 | 1.5 | 0.9 | 1.0 | 2.2[1] | 0.43[1] | 0.8[2] | 7.5[1] |
| São Fransisco | 6 | 2.9 | 0.2 | 2.4 | 0.5[1] | 0.35[1] | - | 0.5[1] |
| Orinoco | 3.3 | 4.8 | 1.1 | 2.9 | 5.0 [1] | 3.9[1] | 1.7[2] | 21.4[1] |
| **Total** | **58** | **44** | **21** | **81.8** | **53** | **46** | **-** | **276** |
| *Arctic* | | | | | | | | |
| Mackenzie | 4.5 | 1.7 | 0.4 | 6.0 | 6.29[3] | 1.4[4] | - | 1.5[6] |
| Yukon | 3.1 | 6.1 | 0.9 | 3.8 | 4.45[3] | 1.7[4] | - | 1.9[6] |
| Lena | 12.7 | 2.0 | 1.1 | 8.23 | 5.82[3] | 5.83[4] | 0.47[4] | 4.4[6] |
| Yenisei | 8.6 | 2.0 | 1.2 | 8.8 | 6.96[3] | 4.69[4] | 0.17[4] | 7.9[6] |
| Ob | 8.6 | 2.6 | 0.8 | 14.1 | 5.90[3] | 3.05[4] | 0.3-0.6[4] | 20.4[6] |
| **Total** | **37.5** | **14.4** | **4.4** | **40.8** | **29.4** | **16.7** | **1.09** | **35.8** |
| *Southeast Asia* | | | | | | | | |
| Ganges | 7.7 | 5.8 | 15.6 | 21.3 | 4.2[7] | 1.4[2] | 1.7[2] | 165[10] |
| Irrawaddy | 2.5 | 6.1 | 0.9 | 4.6 | 10.8[7] | 0.89[2] | 3.25[2] | 8.7[10] |
| Salween | 0.7 | 0.3 | 0.7 | 0.8 | 8.4[7] | 0.26[2] | 2.9[2] | 1.9[10] |
| Mekong | 4.4 | 2.1 | 3.1 | 7.2 | 4.5[7] | - | - | 0.9[8] |
| Huang He | 3.6 | 0.5 | 0.3 | 8.3 | 1.3[7] | 0.1[2] | 6.3[2] | 5.0[9] |
| Yangtze | 23 | 3.5 | 1.7 | 29.9 | 24[7] | 2.1[2] | 6[2] | 92[10] |
| Xi River | 8.6 | 0.9 | 0.4 | 4.3 | - | 4.6[2] | - | 25[10] |
| **Total** | **50.5** | **19.2** | **22.7** | **76.4** | **-** | **-** | **-** | **298.5** |
| *Indo-Pacific Islands* | | | | | | | | |
| **Total** | **19.4** | **10.2** | **10.1** | **24.7** | **-** | **-** | **-** | **-** |

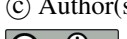


of carbon, whereas for the other regions, organic carbon is the largest contributor to the total carbon load. The increasing contribution of DIC loads to the total carbon load at high latitudes was moreover observed for Arctic catchments in Table 4.

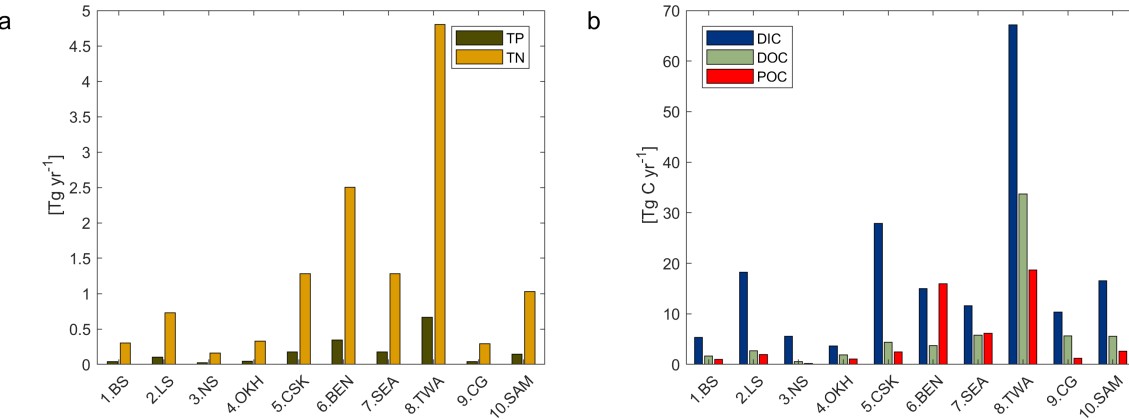

**Figure 6.** (a) Total P (TP) and N (TN) and (b) C (DIC, DOC and POC) exports to the chosen coastal ocean regions. TP and TN are the total P and N modelled in their dissolved inorganic species (DIP, DIN) and organic species (tDOM and POM). DOC and POC are the carbon loads from tDOM and riverine POM, respectively.

## 5    Implications for the ocean biogeochemistry

In this section, we investigate the long-term implications of considering the inputs of pre-industrial riverine loads in an ocean
5    biogeochemical model. We thereby compare the RIV simulation, in which the described pre-industrial riverine loads were added, to the standard model REF simulation, where biogeochemical tracers were added homogeneously to the surface ocean to compensate for particulate losses (CaCO$_3$, opal and organic matter) in the sediment and thus were necessary to maintain a stable ocean state.

### 5.1    Ocean state - An increased biogeochemical coastal sink

10    We observe that the total REF and RIV nutrient magnitudes of the P, N and Si inputs to the ocean are very similar (Table 5), implying that the differences in nutrient concentrations and NPP between RIV and REF originate from the geographic locations of inputs. Alk inputs are also added at nearly the same levels. The total carbon inputs are on the other hand increased by almost 100% in RIV in comparison to REF. These larger carbon inputs originate firstly from higher DIC to Alk ratio of the riverine loads (1:1) than is exported through the net CaCO$_3$ production (1:2). Secondly, there is a higher carbon load originating from
15    organic matter, since the tDOM C:P ratio is higher than the oceanic DOM C:P ratio. In both cases, the model inorganic (366 Tg C yr$^{-1}$) and organic (237 Tg C yr$^{-1}$) carbon inputs in RIV show stronger agreement with the riverine inorganic (260-550 Tg C yr$^{-1}$) and organic carbon (270 - 350 Tg C yr$^{-1}$) global load estimates found in literature (Meybeck, 1982; Amiotte Suchet





and Probst, 1995; Mackenzie et al., 1998; Meybeck and Vörösmarty, 1999; Hartmann et al., 2009; Seitzinger et al., 2010; Cai, 2011; Regnier et al., 2013). These higher carbon inputs result in a net long-term outgassing flux (231 Tg C yr$^{-1}$), which we will discuss in detail.

Despite slightly larger inputs of nutrients in RIV than in REF, lower global surface dissolved nutrient concentrations and lower global primary production rates are found in RIV. The coastal ocean therefore acts as an increased biogeochemical sink in the model, since river-delivered or newly produced particulate organic matter reaches the shelf sea floor faster than in the open ocean, allowing for less time for the organic matter to be remineralized within the water column. On shallow shelves (<250m depth), we find an increased organic matter flux of 0.25 Gt C yr$^{-1}$ to the sediment in RIV versus a flux of 0.18 Gt C yr$^{-1}$ for REF. Although strong uncertainties exist in literature regarding the coastal POM sediment deposition flux, the range of global values given in a review by Krumins et al. (2013) (0.19-2.20 Gt C yr$^{-1}$) hints that the coastal POM deposition flux is possibly improved in RIV.

The global mean surface concentration is lower in RIV than in the observational data of the World Ocean Atlas 2013 (WOA, Boyer et al. (2013)) for DIP (0.439 and 0.480 $\mu$M P respectively) and DIN (3.90 and 5.04 $\mu$M N), and higher for DSi (13.6 and 7.5 $\mu$M Si). The WOA dataset is constructed from present-day observations of an ocean state that might already be perturbed by a substantial increase in riverine P and especially N loads, whereas the model shows pre-industrial concentrations. A consideration of the substantial anthropogenic increase in DIN riverine loads (Seitzinger et al., 2010; Beusen et al., 2016) could plausibly shrink some of the disagreement with the WOA dataset in the case of DIN. A large part of the DIN underestimation is however most likely due to notably large tropical Pacific oxygen minimum zones, which cause a large DIN sink due to denitrification and the consumption of DIN in the anaerobic breakdown of organic matter. Furthermore, the lower surface concentrations of DIP and DIN than found in the WOA dataset suggest that the coastal sink of biogeochemical tracers might be too large. Nevertheless, the surface DIN:DIP ratio in RIV is slightly improved in comparison to REF with regards to WOA, which is most likely due to the shrinking of the tropical Pacific oxygen minimum zones.

The DIP and DIN underestimation bias with respect to the WOA datasets are also reflected in the spatial distributions of the surface concentrations (Figure 7), where in particular the DIN concentrations are underestimated in most major basins. The spatial patterns of differences with regard to WOA data are similar for RIV and for REF, suggesting ocean physics being the dominant driver of the nutrient distributions in the open ocean (see Appendix, Figure E1,F1,G1). Prominent bias of the model are lower surface DIP and DIN concentrations in the Southern Ocean, higher DSi concentrations in the Southern Ocean, and higher DIP concentrations in the tropical gyres in comparison with the WOA dataset.

## 5.2 Riverine-induced NPP hotspots

The net primary production (NPP) in the most productive open ocean regions (tropical Atlantic and Pacific, north Pacific, Southern Ocean) are reduced in RIV with respect to REF, but nevertheless remain the most dominant areas of biological production (Figure 8a,b). However, substantial enhancements in the NPP can be found near various major river mouths. In proximity to lower latitude rivers such as the Amazon, the uptake of nutrients by phytoplankton via primary production occurs





**Table 5.** Comparison of river inputs and the ocean state for REF and RIV. Additionally, we compare the modelled mean surface DIP, DIN and DSi concentrations with World Ocean Atlas 2013 (WOA) surface layer means.

| Variables | REF | RIV | WOA |
|---|---|---|---|
| ***(River) Inputs*** | | | |
| P [Tg P yr$^{-1}$] | 3.49 | 3.7 | |
| N [Tg N yr$^{-1}$] | 25.2 | 27 | |
| Si [Tg Si] | 115 | 158 | |
| Alk [Tg HCO$_3^-$ yr$^{-1}$] | 416 | 366 | |
| Inorganic C [Tg C yr$^{-1}$] | 208 | 366 | |
| Organic C [Tg C yr$^{-1}$] | 106 | 237 | |
| | | | |
| ***Ocean variables*** | | | |
| Global net primary production [Gt C yr$^{-1}$] | 48.87 | 47.09 | |
| Net CO$_2$ flux [Gt C yr$^{-1}$] | -0.05 | 0.18 | |
| Global organic material export 90m [Gt C yr$^{-1}$] | 6.84 | 6.47 | |
| Global calcium carbonate export 90m [Gt C yr$^{-1}$] | 0.66 | 0.61 | |
| Global opal production [Gt Si yr$^{-1}$] | 1.37 | 1.38 | |
| Surface Alk [mM] | 2.25 | 2.24 | |
| Surface DIC [mM C] | 1.94 | 1.94 | |
| Surface DIP [$\mu$M P] | 0.439 | 0.413 | 0.48 |
| Surface DIN [$\mu$M N] | 3.90 | 3.76 | 5.04 |
| Surface DSi [$\mu$M Si] | 13.6 | 14.6 | 7.5 |
| Oxygen minimum zones volume [km$^3$] | 2.61 | 2.45 | |

efficiently due to favorable light conditions. This freshly produced organic matter, in addition to the terrestrial supplies of organic material delivered by rivers, leads to local increases in oceanic organic material concentrations (Figure 8c,d).

In the tropical Atlantic, a region we identified to have major nutrient and carbon riverine supplies in section 4.2, the NPP is increased near the mouths of the major rivers. This is most notably the case in the Amazon plume, which mirrors the direction

5 of the freshwater plume northwestwards (see Appendix, Figure A1). However, in the open ocean of the equatorial Atlantic, where upwelling takes place from deeper water layers, the NPP is decreased. While considering both effects, the NPP increases by only 2% for the whole region, while there are nevertheless substantial NPP increases along the western African and eastern South American tropical shelves.

In the Equatorial Pacific, we observe a strong decrease in NPP, a feature that could be partly explained by the South American

10 river systems, which majorly discharge into the Atlantic (Figure 6). Although Southeast Asian rivers deliver substantial amounts



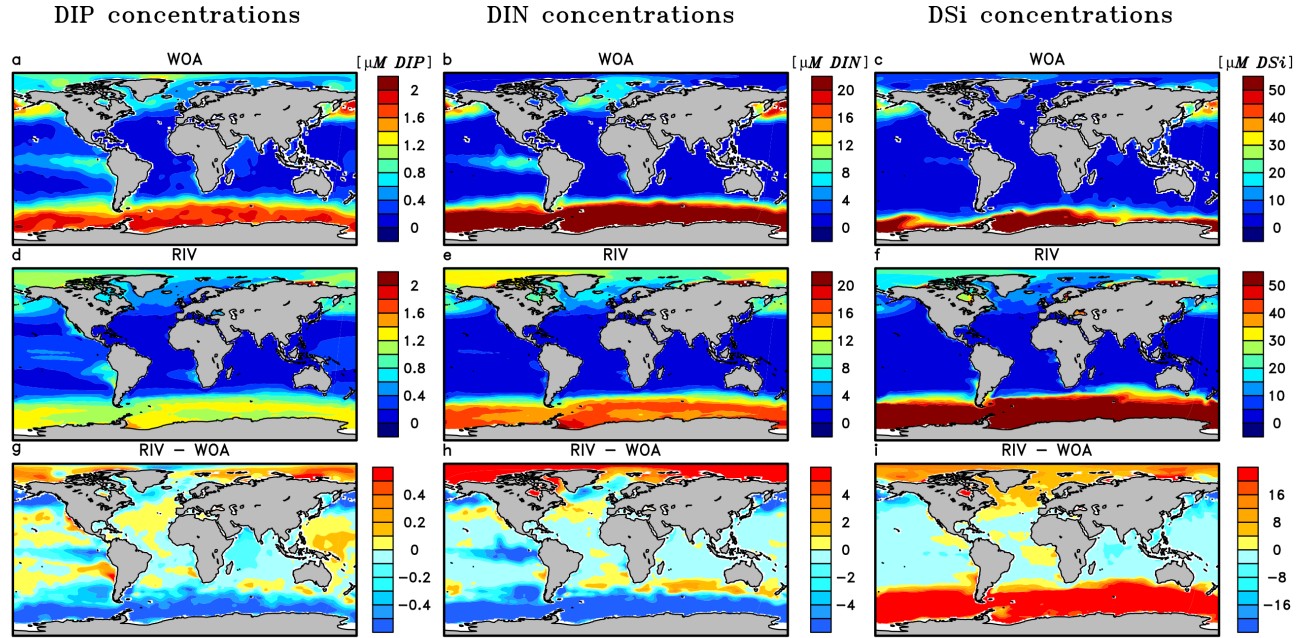

**Figure 7.** Surface DIP (a,d,g), DIN (b,e,h) and DSi (c,f,i) concentrations in WOA observations (a,b,c), RIV(d,e,f) and their differences RIV-OBS(g,h,i).

of land-derived material to the ocean, the export to the open Pacific appears to be inefficient, with model coastal salinity profiles in this region suggesting little mixing with the open ocean. Coastal parallel currents could be a key reason explaining the ineffecent export (Ichikawa and Beardsley, 2002). Furthermore, the riverine loads mostly supply semi-enclosed or marginal seas (East China, South China and Yellow Seas), which have limited exchange to the open ocean and which are affected by the

5   relatively coarse GR15 model resolution in this region (Jungclaus et al., 2013). The resulting decrease in the Equatorial Pacific NPP is responsible for most of the shrinking of oxygen minimum zones (Table 5). The NPP is decreased in the Benguela Current System, which is a major NPP hotspot in the model due to nutrients being entrained to the surface from deeper layers. Moreover, the NPP is increased in certain semi-enclosed seas such as the Caribbean Sea, Baltic Sea, the Black Sea and the Yellow Sea, where satellite observation data also suggest high chlorophyll concentrations (Behrenfeld and Falkowski, 1997).

10   The Arctic Ocean does not show a noticeable increase in NPP, despite high nutrient concentrations in the basin. This can be explained by the light limitation, as well as sea ice coverage inhibiting the primary production especially during the winter. In the entire basin, the nutrient concentrations are much higher than what is suggested in the WOA database. In Bernard et al. (2011), where nutrient inputs were added to the ocean according to the NEWS2 study, similarly high concentrations of DSi were found in the Arctic. Furthermore, Harrison and Cota (1991) suggest that nutrients limit phytoplankton growth in the late





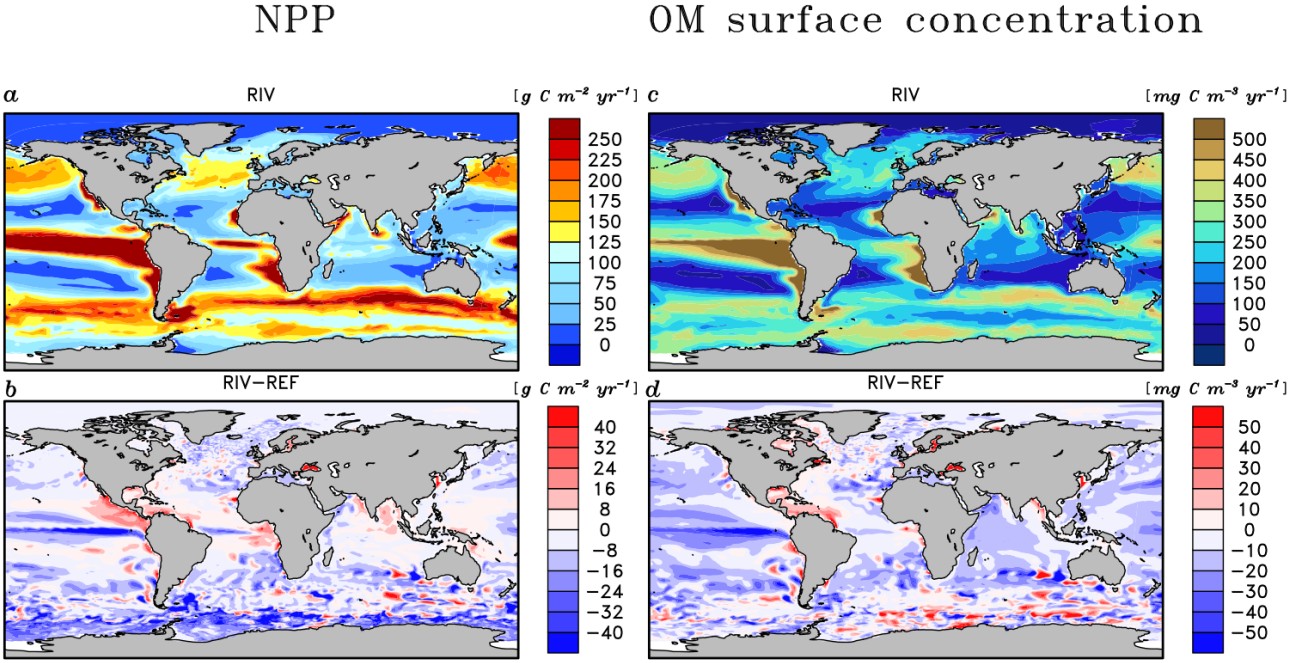

**Figure 8.** Depth integrated annual NPP (a,b) and total annually accumulated organic concentration (c,d) in the surface layer in the RIV simulation and RIV - REF.

Summer in the Arctic Ocean. Although the summer primary production in the model is substantially higher than for other seasons, the NPP is never nutrient limited for the vast majority of the Arctic.

### 5.3 Riverine-induced CO$_2$ Outgassing

The addition of riverine carbon loads causes an oceanic CO$_2$ source of 231 Tg C yr$^{-1}$ to the atmosphere (Table 5). The out-
5    gassing flux is thereby caused by carbon contained in tDOM, as well as considering the inputs of DIC. While the hotspots of the riverine-induced carbon outgassing are regions in proximity to major river mouths (Figure 9b), a widespread, albeit weaker outgassing signal can be observed in open ocean basins. The largest outgassing fluxes are found in the Atlantic and Indo-Pacific (31% and 43% of global outgassing flux respectively), likely due to the tropical Atlantic and Southeast Asian hotspots of riverine carbon supplies (see section 4.2). In the Southern Ocean, we observe an increase in the outgassing flux of
10    17 Tg yr$^{-1}$ when comparing RIV to REF, which is almost 10% of the total riverine-caused outgassing. The southern hemisphere shows an oceanic outgassing flux of 113 Tg yr$^{-1}$ (49%), despite southern hemisphere land exports contributing only 227 Tg (36 %) of total riverine carbon loads to the ocean, which suggests a substantial interhemispheric transfer of carbon from the northern hemisphere to the southern hemisphere. The interhemispheric transfer of carbon in the ocean has been a topic of



discussion in literature, with studies of Aumont et al. (2001) and Resplandy et al. (2018) suggesting the transport of carbon between latitudinal regions of the ocean to compensate for the heterogeneous terrestrial supplies.

The high latitude Arctic rivers (Lena, Mackenzie, Yenisei, Ob, Oder, Yukon) provide a source of carbon to their respective shelves, which causes outgassing on the Laptev shelf and in the Beaufort Sea (2.2 Tg C yr$^{-1}$ and 2.3 Tg C yr$^{-1}$, respectively).

The impacts of riverine carbon loads in these regions can also be observed in the present-day coastal ocean $pCO_2$ dataset of Laruelle et al. (2017), in which these regions display very high $pCO_2$ values.

While all areas in proximity to the river mouths show increases in $CO_2$ outgassing caused by the addition of riverine inputs of carbon (Figure 9b), determining the net sign of the $CO_2$ of individual plumes in RIV is not as straightforward. A $CO_2$ undersaturation in many river plumes can still be observed despite the addition of riverine carbon (Figure 9a). Riverine nutrient,

carbon, alkalinity and freshwater inputs, as well as physical and biogeochemical oceanic features all interact to affect the net $CO_2$ flux. The Amazon plume is a prominent example, which is a net carbon sink near the river mouth in the model despite being supplied by large amounts of carbon from the Amazon river. The near-shore Amazon plume is thereby also identified as an atmospheric carbon sink in literature (Cooley et al., 2007; Lefèvre et al., 2017).

### 5.4 Sensitivity of the NPP and $CO_2$ flux in chosen coastal regions

The areas of the 10 chosen coastal ocean regions (Table 1 in section 4.3) are better represented for the Atlantic shelves than eastern Asian shelves due higher resolutions of the GR15 model in the Atlantic (Jungclaus et al., 2013). We observe strong latitudinal differences in the regional riverine inputs (section 4.3), and analyze their implications for the coastal ocean regions here. We observe strong differences in the regional responses to the riverine loads, with a tendency of stronger relative changes in NPP on lower latitudinal shelves, and stronger relative changes in $CO_2$ in the higher latitudes (Figure 10).

For tropical and subtropical regions, we observe major NPP increases of 166%, 377% and 71% for the tropical West Atlantic (3.TWA), Bay of Bengal (5.BEN) and East China Sea (6.CSK), respectively. The availability of light, as well as the large supplies of nutrients to these regions provide optimal conditions to enhance the biological production. Surprisingly however, the Southeast Asian shelf (6.SEA) does not show a similar substantial NPP increase as the other tropical regions despite considerable riverine fluxes to the region (Table 4 in Section 4.2). This is on one hand due to the large area of the defined

region; the shelf is the largest that is analyzed in this study (1795 $10^9$ m$^2$), which reduces the impact of the river loads per area. Secondly, there is a larger connection area to the open ocean due to not sharing a coastal border with a continent, which implies a larger open ocean exchange, thus reducing the influence of the riverine supply with regard to open ocean supplies. The Congo shelf (9.CG) on the other hand has a very small area (53 $10^9$ m$^2$) due to a steep coastal slope. The NPP here is however already one of the highest of the chosen regions without considering rivers, suggesting that the region is already strongly supplied with

nutrients from coastal upwelling.

On the temperate shelves, where there is a stronger seasonal cycle of the light limitation, the North Sea (2.NS) shows only weak enhancement in the NPP (2%) due to riverine inputs. The South American (10.SAM) and Sea of Okhotsk (4.OKH) also do not show significant NPP increases. Although the NPP is strongly enhanced in the direct proximity to the Paraná river (Figure 8b), the vastness of the South American shelf (1553 $10^9$ m$^2$) also makes the region less sensitive to river inputs. In published



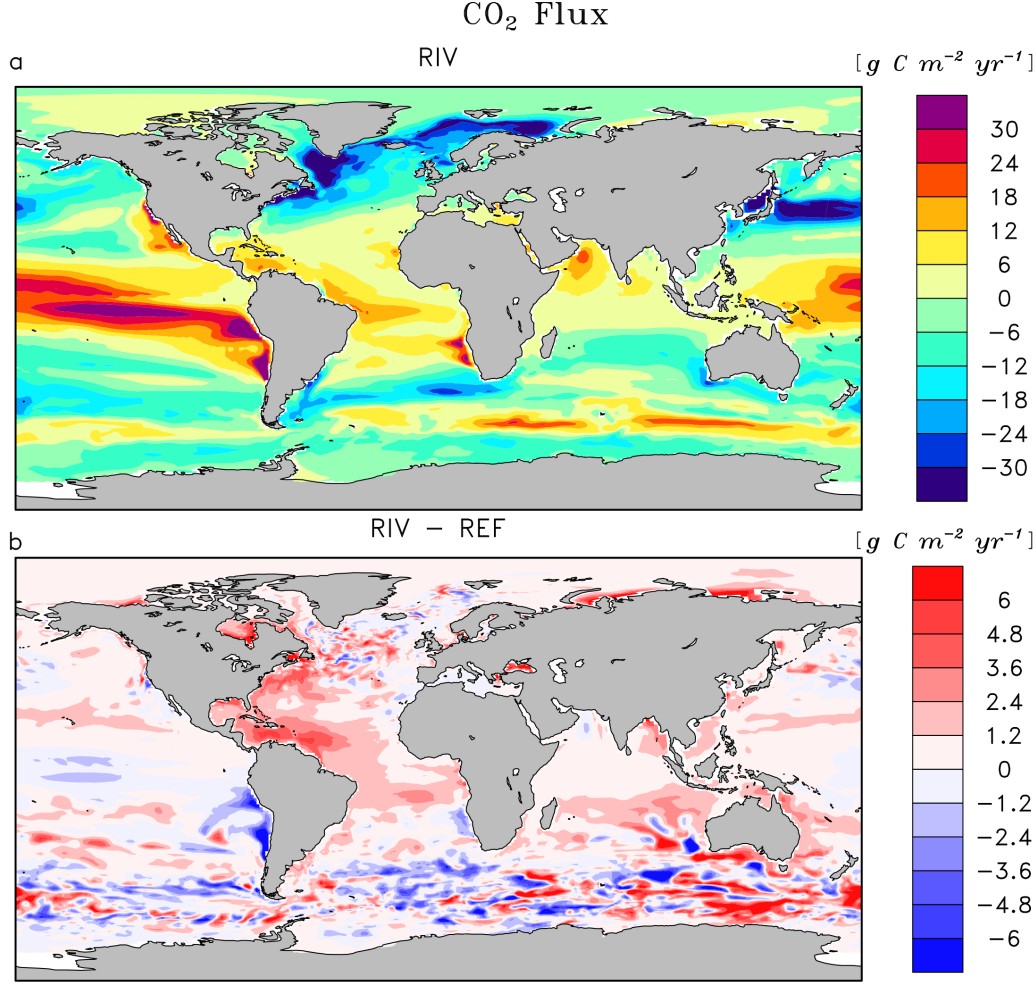

**Figure 9.** Annual pre-industrial air-sea CO₂ exchange flux of (a) RIV and (b) RIV-REF. A positive flux describes an outgassing flux from the ocean to the atmosphere, whereas a negative flux is from the atmosphere to the ocean.

literature, the nutrient supply which drives the NPP on the Patagonian Shelf is also confirmed to be strongly controlled by the open ocean inflows (Song et al., 2016).

The Arctic shelf regions do not show a strong NPP response to the river inputs (8% and 5% increases for the Beaufort Sea ,1.BS, and Laptev Sea ,2.LS, respectively). We however do not consider seasonality of the riverine inputs. Larger inputs
5 of nutrients in months of larger discharge (Le Fouest et al., 2013) of April to June, which are also months of better light availability, could cause a more efficient usage of the riverine nutrients, since the sea-ice coverage is strongly reduced in these months.





All regions show an increase in $CO_2$ outgassing due to the carbon inputs to the ocean. In the Arctic regions (Beaufort Sea and Laptev Sea), the relative change is much more pronounced, whereas the impact is generally not as strong in the lower latitude regions due to the enhancement of biological carbon uptake by the nutrient inputs. The tropical West Atlantic is an exception to this latitudinal pattern, since the large carbon riverine supplies also cause a substantial change in the $CO_2$ flux of

5   the region. In the North Sea, we observe an enhancement of carbon outgassing, but the region remains a substantial sink of atmospheric $CO_2$, as is still suggested for the present-day by Laruelle et al. (2014).



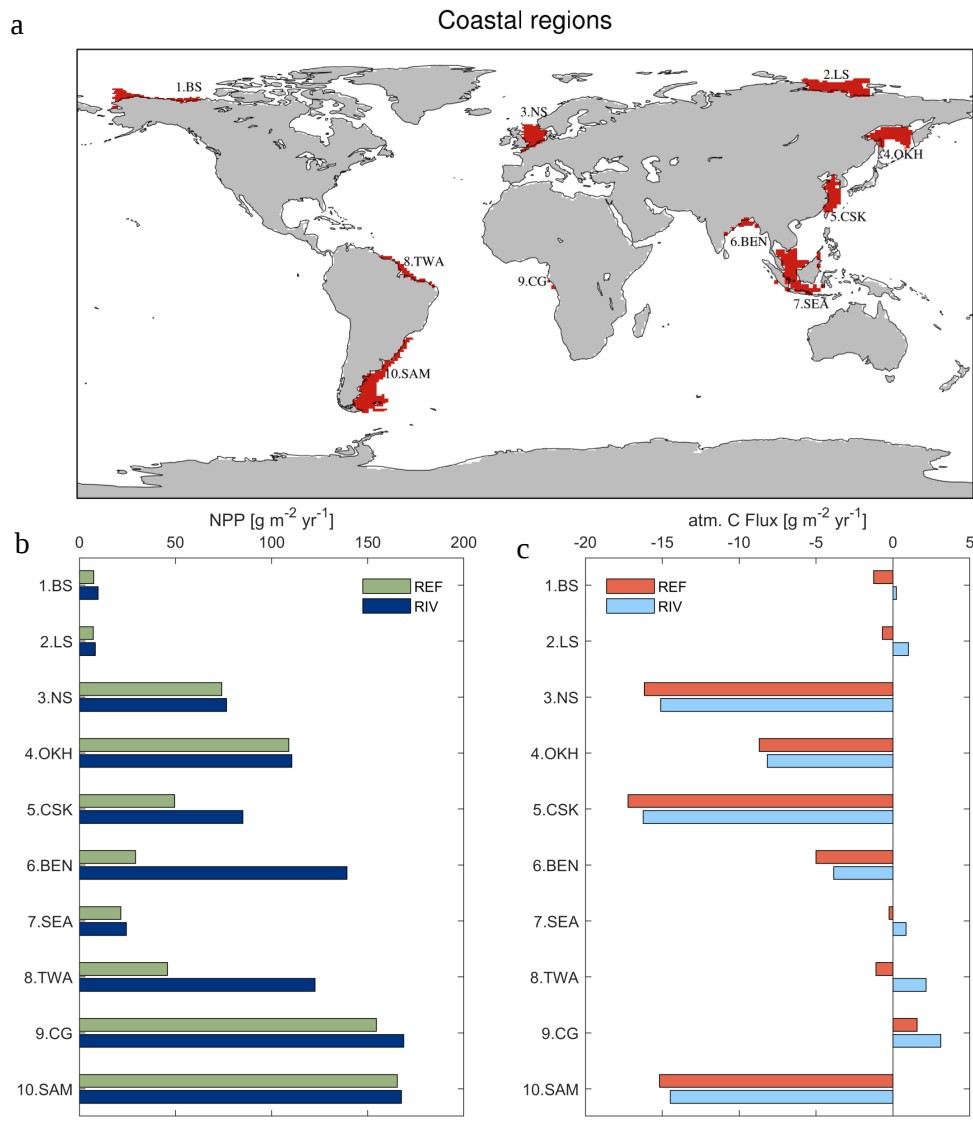

**Figure 10.** (a) Global map of the 10 chosen coastal regions with less than 250m depths and (b) pre-industrial annual NPP per area and $CO_2$ flux in the given regions [g m$^2$ yr$^{-1}$].





## 6 Origins and fate of riverine carbon

In our simplified land-ocean system (Figure 11), we quantify the land sources of riverine carbon (**1-3**), its riverine transfer to the ocean (**4**), and the long-term fate of the riverine carbon in the ocean (**5-10**). Here, we briefly explain the fluxes to focus on their implications. While the terrestrial fluxes are derived from the weathering and organic matter export models, the long-

term oceanic fluxes are based on fluxes given by the ocean biogeochemical model. The long-term $CO_2$ flux is furthermore decomposed to illustrate the contributions of inorganic and organic carbon inputs to the oceanic outgassing flux in a model equilibrium analysis. The detailed derivation of the land and ocean fluxes are explained in detail in the Appendix C (C1 for terrestrial and C2 for oceanic fluxes).

The net pre-industrial terrestrial uptake of atmospheric $CO_2$ and its export of rivers amounts to 529 Tg C yr$^{-1}$ in our frame-

work. The sink consists of 280 Tg C yr$^{-1}$ from the $CO_2$ drawdown induced by weathering (**1**) and 249 Tg C yr$^{-1}$ due to the land biological uptake (**3**). During the weathering process 94 Tg C yr$^{-1}$ is moreover released from the lithology during carbonate weathering (**2**), which is also reported in Hartmann et al. (2009). During silicate weathering, all the carbon originates from atmospheric $CO_2$. The land biological uptake (**3**) is derived from the net global export of organic carbon to the ocean. It therefore implicitly takes into account the net soil carbon uptake and export to freshwaters, as well as all net sinks and sources in river

systems. A total 603 Tg C yr$^{-1}$ is transferred laterally to the ocean (**4**) while taking into consideration an endorheic catchment loss of 19 Tg C yr$^{-1}$.

In the ocean, riverine exports of carbon cause a long-term net annual carbon source of 231 Tg C yr$^{-1}$. We propose a decomposition of the long-term $CO_2$ flux into sources and sinks induced by the inputs of riverine species (Appendix C.2). Assuming model equilibrium, the oceanic outgassing flux can be decomposed into a source from inorganic carbon supplied by weathering

(183 Tg C yr$^{-1}$, **5**), a source from terrestrial organic carbon (128 Tg C yr$^{-1}$), **6**), a sink caused by the enhancement of the biology due to the addition of dissolved inorganic nutrient and corresponding alkalinity production (69 Tg C yr$^{-1}$, **7**) and a sink due to disequilibrium at the atmosphere-water column interface in the model (11 Tg C yr$^{-1}$, **D1**). The production and the sinking of $CaCO_3$ and POM within the ocean lead to simulated sediment deposition fluxes of 188 Tg C yr$^{-1}$ for $CaCO_3$ (inorg. C. flux, **8**) and 582 Tg C yr$^{-1}$ for POM (org. C flux, **9**). The dissolution of $CaCO_3$ and the remineralization of POM within the sediment

lead to a DIC flux from the sediment to the water column of 385 Tg C yr$^{-1}$. The net C flux at the sediment interface (**8+9-10**) is therefore a burial flux of 385 Tg C yr$^{-1}$. The calculated equilibrium carbon burial flux, which is the difference between the riverine carbon inputs and the equilibrium $CO_2$ outgassing (**4-5-6+7**), is 361 Tg C yr$^{-1}$, which implies that there is a deviation of 24 Tg C yr$^{-1}$ (**D2**) towards the sediment in the simulated model burial (385 Tg C yr$^{-1}$) with respect to the calculated equilibrium state burial (361 Tg C yr$^{-1}$). The similar deviations from the equilibrium state at the atmosphere-water column and water

column-sediment interfaces suggest that the model drift in alkalinity, which most likely originates from disequilibrium in the sediment layers, translates efficiently into a perturbation of the $CO_2$ at the atmosphere-water column interface.

The riverine induced oceanic $CO_2$ outgassing flux of 231 Tg C yr$^{-1}$ is consistent with the estimate range of 200-400 Tg C yr$^{-1}$ given in Sarmiento and Sundquist (1992), who assume an annual riverine C flux of 300-500 Tg C, and with Jacobson et al. (2007) and Gruber et al. (2009), who suggest a slightly higher natural $CO_2$ outgassing flux of 450 Tg C yr$^{-1}$. Resplandy et al.





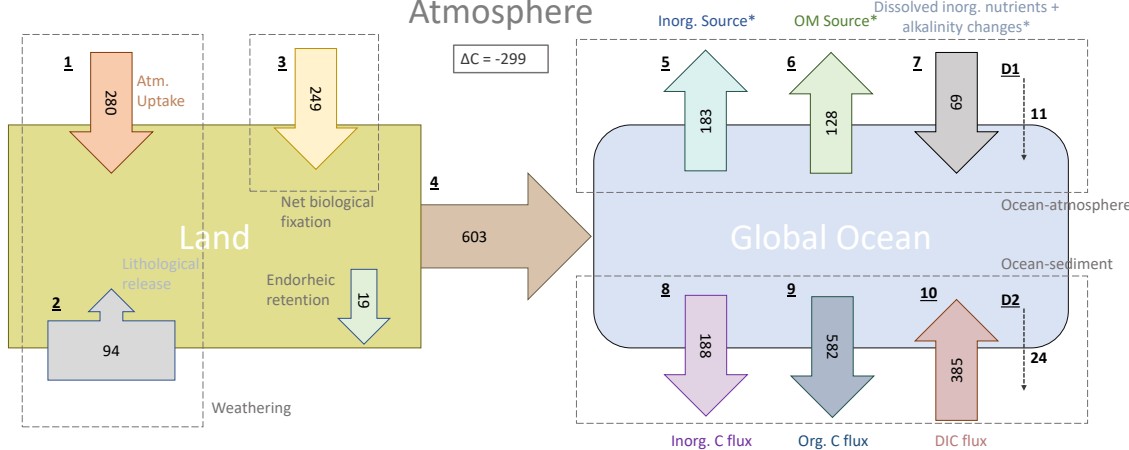

**Figure 11.** Origins and oceanic fate of riverine carbon in our simplified land scheme coupled to HAMOCC (RIV simulation) [Tg C yr$^{-1}$].**1.** Land C uptake through weathering. **2.** Carbonate weathering lithological C flux. **3.** Net land biological C uptake (derived directly from riverine organic carbon exports). **4.** Riverine C exports. **5.** Oceanic outgassing from riverine DIC. **6.** Oceanic outgassing resulting organic material (OM) loads. **7.** Oceanic C uptake due to the enhanced primary production by dissolved inorganic nutrients and the corresponding alkalinity production. **8.** Simulated inorganic C deposition to the sediment. **9.** Simulated net organic C deposition to sediment. **10.** Diffusive DIC flux from the sediment back to the water column. * are calculated fluxes for ocean model equilibrium, whereas the other fluxes are simulated fluxes by the terrestrial and ocean models. D1 and D2 are the calculated drifts between the oceanic modelled carbon fluxes and the calculated equilibrium fluxes (derived from model equations, Appendix C2) for the ocean-atmosphere and ocean sediment interfaces. See Appendix C for the derivation of the fluxes.

(2018) suggest a higher land-ocean carbon (780 Tg C yr$^{-1}$) from the derivation of natural outgassing of carbon in the ocean. It is unclear if and how the method considers oceanic carbon removal of the riverine-delivered carbon through sediment burial.

Furthermore, we observe an imbalance in the calculated pre-industrial CO$_2$ land uptake from the atmosphere and the oceanic outgassing in our approach, with the land uptake outweighing the oceanic outgassing, resulting in a total net atmospheric sink

5 of 299 Tg C yr$^{-1}$. Accounting for further sources of atmospheric CO$_2$ such as volcanic emissions and shale organic oxidation would therefore be necessary to achieve a stable atmospheric carbon budget in a fully coupled land-ocean-atmosphere setting, since Earth System Models assume constant pre-industrial atmospheric CO$_2$ levels. For instance, Mörner and Etiope (2002) suggest long term volcanic annual emissions in the range of 80-160 Tg C yr$^{-1}$, whereas Burton et al. (2013) estimate volcanic CO$_2$ fluxes as high as that of silicate weathering drawdown, which would reduce the disequilibrium in the pre-industrial

10 atmospheric CO$_2$ budget. In our approach, the silicate weathering CO$_2$ drawdown is of 196 Tg C yr$^{-1}$. Additionally to the volcanic CO$_2$ emissions, the global atmospheric CO$_2$ land source of around 100 Tg C yr$^{-1}$ given by Sarmiento and Sundquist (1992), due to the oxidation of organic carbon in rocks, would then approximately close the atmospheric carbon budget in our framework.





## 7    Approach advantages and limitations

### 7.1    Rivers in an Earth System Model setting

Our approach to represent riverine loads as a function of the climate variables runoff, precipitation and temperature can be used to estimate land-sea fluxes in an Earth System Model (ESM) setting. For one, this could help tackle questions of the past, since

weathering rates are dependent on climate variables (Berner, 1991). For instance, strong differences in weathering for the last glacial maximum have been suggested (Brault et al., 2017). The impacts of future climate change on weathering rates and land-sea fluxes could be addressed, as Gislason et al. (2009) and Beaulieu et al. (2012) suggest major changes in weathering due to changing climatic conditions on a decadal timescale. Furthermore, the RIV simulation can serve as a pre-industrial initial state to investigate temporal changes to the riverine loads over the 20th century, and in the future. Conclusions of Seitzinger et al.

(2010) and Beusen et al. (2016) reveal strong increases DIP and DIN river loads during the 20th century, for which the oceanic impacts could be assessed.

Our approach provides a basis in order to address further questions regarding the land-ocean transfer dynamics. For one, improvements in the weathering mechanisms are possible (biological weathering enhancement and secondary mineral weathering). Ecosystem and the dynamical uptake of nutrients by the land biology, and their storage in the soil could be modelled, as

well as considering hydrological flow characteristics and river biogeochemical transformation processes. The consideration of groundwater fluxes could also be included. Coastal ocean dynamics could moreover be more accurately represented by using a higher model resolution.

### 7.2    Fate and consistence of terrestrial organic matter in the ocean

The composition of terrestrial organic matter, as well as its fate in the ocean are associated with a large degree of uncertainty.

Recent work shows that tDOM is mineralized efficiently by abiotic processes in the coastal zone (Fichot and Benner, 2014; Müller et al., 2016; Fichot and Benner, 2014), despite having already been strongly degraded along the land-ocean continuum. On the other hand, few studies tackle the composition of POM, although it is thought to also be efficiently remineralized in the coastal zone sediment (Hedges et al., 1997; Cai, 2011). While the carbon loads from POM are the lowest loads of all carbon compounds considered in this study, a differing C:P ratio to the one chosen in this study would also affect the model outgassing

flux estimated here.

Rates of coastal remineralization processes have been suggested to differ from those of the open ocean (e.g. Krumins et al. (2013)). A higher sediment organic matter remineralization rate observed in coastal sediments (Krumins et al., 2013) could potentially reduce the coastal biogeochemical sink described in this study.

### 7.3    Arctic Ocean

The simulated nutrient concentrations in the Arctic Ocean are particularly high with regards to WOA data. Furthermore high dissolved organic material concentrations are found in observations for the Arctic (Benner et al., 2005). These characteristics





suggest that this region with strong riverine inputs might be poorly represented in the ocean biogeochemistry model. Difficulties to represent the region could be due to fine circulation features, with outflows through narrow passages having been shown to be affected by model resolution (Aksenov et al., 2010). Moreover, the primary production in the region might be underestimated due to photosynthesis taking place under ice, in ice ponds and over extended daytime periods in the summer months (Deal

et al., 2011; Sørensen et al., 2017), all of which are not represented in the model.

## 8    Summary and conclusions

In this study, we provide global and spatial weathering release yields for P, Si, DIC and Alk, that are derived from driving spatially explicit models with MPI-ESM output of runoff, surface temperature and precipitation. These yields show good agreement with previous assessments found in published literature. The weathering yields are of disproportionate magnitude in

warm and wet regions, confirming what has been suggested until now (Amiotte Suchet and Probst, 1995; Beusen et al., 2009; Hartmann et al., 2009, 2014). Since ESMs tend to have substantial bias when quantifying the global runoff (Goll et al., 2014), runoff correction terms are needed to produce plausible weathering yields at the global scale. In the case of the MPI-ESM used in this study, which substantially underestimates the global runoff with regards to estimates of Dai and Trenberth (2002) and Fekete et al. (2002), a factor of 1.59 is necessary.

Accounting for weathering and non-weathering inputs to river catchments results in annual pre-industrial loads of 3.7 Tg P, 27 N, 168 Tg $SiO_2$, and 603 Tg C to the ocean. These loads are consistent with published literature estimates, although we acknowledge a certain degree of uncertainty regarding the magnitude of riverine fluxes. Even for the present-day, substantial differences can be found between different approaches to derive land-ocean exports (Beusen et al., 2016). While we omit the in-stream retention of P during its riverine transport, which reduces the global P exports to the ocean (Beusen et al., 2016),

our estimate of global P export to the ocean is comparable in magnitude to an approach that determines riverine P exports by upscaling from pristine river measurements (Compton et al., 2000).

We identify the tropical Atlantic catchments, the Arctic Ocean, Southeast Asia and Indo-Pacific islands as regions of dominant contributions of riverine supplies to the ocean. These 4 regions account for over 51% of land-ocean carbon exports in total, with tropical Atlantic catchments supplying around 20% of carbon to the ocean globally. We also observe that the contributions

of different carbon species differ between the regions. Most prominently, the carbon supply of the Indo-Pacific islands is dominated by particulate organic carbon loads, which have been identified to be more strongly controlled by extreme hydrological events than other C species (Hilton et al., 2008).

In the ocean, riverine inputs of carbon lead to net global oceanic outgassing of 231 Gt C $yr^{-1}$, a comparable value with regards to previous estimates of 200-450 Tg C $yr^{-1}$ (Sarmiento and Sundquist, 1992; Jacobson et al., 2007; Gruber et al., 2009).

This outgassing flux can be decomposed into two source terms caused by inorganic C inputs (183 Tg C $yr^{-1}$) and organic C inputs (128 Tg C $yr^{-1}$), and a net sink term (80 Tg C $yr^{-1}$) caused by the enhanced biological C uptake due to riverine inorganic nutrient supplies, corresponding alkalinity production and a slight model drift in alkalinity. The magnitude of the outgassing is however strongly dependent on the magnitude of riverine carbon loads, for which uncertainties still exist.



We observe evidence of a substantial interhemispheric transport of carbon from the northern to the southern hemisphere, with a larger relative carbon outgassing flux in the southern hemisphere (49% of global outgassing) than its relative riverine carbon inputs to the ocean (36% of global C loads). We also show that the Southern Ocean outgasses 17 Tg of riverine carbon, despite not having a direct riverine source of carbon, meaning that riverine carbon is transported within the ocean interior to the

Southern Ocean. This interhemispheric transfer of riverine carbon in the ocean has been previously suggested to contribute to the pre-industrial Southern Ocean source of atmospheric $CO_2$ for the pre-industrial time-frame (Sarmiento et al., 2000; Aumont et al., 2001; Gruber et al., 2009; Resplandy et al., 2018). Here we show that riverine carbon fluxes derived from state-of-the-art land export models confirm the larger contribution of the northern hemispheric terrestrial carbon supply to the ocean. Part of the uneven hemispheric terrestrial carbon supply is then compensated by the transport of carbon within the ocean and is

outgassed remotely to the atmosphere.

Our results help identify oceanic regions that are sensitive to riverine fluxes. Riverine-induced changes in the regional NPP are mostly found in coastal regions, but significant riverine-derived $CO_2$ outgassing can also be observed in the open ocean of the tropical Atlantic. In general, latitudinal differences can also be observed in the sensitivity of the NPP and the $CO_2$ fluxes of various shallow shelves to riverine fluxes. While a high sensitivity in the NPP is found in tropical latitudes, with the tropical

West Atlantic, the Bay of Bengal and the East China Sea showing large increases of 166%, 377% and 71% respectively, the relative changes in the regional $CO_2$ fluxes are larger at higher latitudes. For instance, the Laptev Sea and the Bay of Beaufort become a atmospheric sources of 2.2 Tg C yr$^{-1}$ and 2.3 Tg C yr$^{-1}$ respectively, despite previously being sinks of atmospheric $CO_2$ in the model. While our analysis revolves around pre-industrial riverine exports, regions that show high sensitivity might also be more strongly affected by 20th century anthropogenic perturbations of land-ocean exports.

Deriving riverine exports as a function of Earth System Model variables (precipitation, temperature and runoff) enables a representation of the riverine loop, from the terrestrial uptake of carbon, its riverine export and to its long-term outgassing in the ocean and export to the oceanic sediment. In the case of implementing the framework in a coupled land-atmosphere-ocean setting such as an ESM, the atmospheric pre-industrial budget would have to be balanced. In our study, we emphasize the need to consider a land $CO_2$ source originating from long-term volcanic activity and from shale organic carbon oxidation in order to

close the pre-industrial atmospheric C budget.

Throughout this study, we find global heterogeneity in the spatial features of weathering fluxes, riverine loads and their implications for the ocean biogeochemistry. This, for instance, leads to the observed interhemispheric transfer of carbon in the ocean, with the dominance of northern hemispheric land-ocean carbon exports being evened out by remote oceanic carbon outgassing fluxes. Our results confirm the importance of nutrient and carbon fluxes for the biogeochemistry of various shallow

shelves and for the Arctic Ocean $CO_2$ flux. Our study also shows the necessity to account for the riverine-induced oceanic outgassing of carbon in ocean biogeochemistry models, since our conservative estimate consists of around 10% of the magnitude of the present-day ocean carbon uptake.





*Code and data availability.* Code, primary data and scripts needed to reproduce the analyses presented in this study are archived by the Max Planck Institute for Meteorology are available upon request (publications@mpimet.mpg.de)

## Appendix A: Freshwater Fluxes

The freshwater loads from OMIP were already implemented in previous standard HAMOCC version, in order to close the hydrological cycle. Globally, they add 32,542 km$^3$ yr$^{-1}$ of freshwater to the ocean. These impact the salinity and the ocean stratification in the proximity of our biogeochemical riverine inputs, and therefore their advection within the ocean.

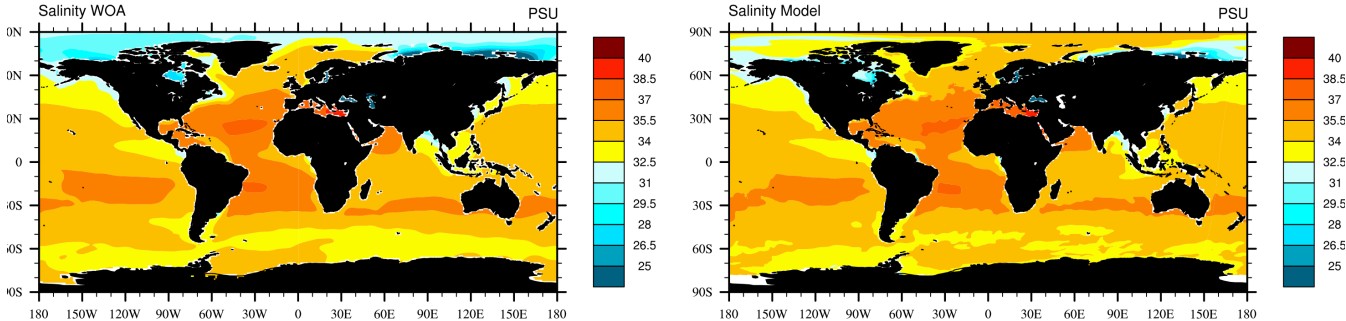

**Figure A1.** Comparison of salinity from observational data (WOA) and modelled salinity in RIV (Salinity Model).

.



## Appendix B: Coastal Salinity and Nutrient Profiles

Coastal vertical profiles show that despite the coarse resolution of our model, the features of salinity in selected coastal regions
is comparable to WOA data, although vertical gradients are often not as strong as in the observational data (Figure A.2).
For the Amazon Northward section, we see a vertical the salinity gradient in the WOA data, which is also reproduced albeit

5   not extensively in the modelled data which induces stronger stratification of the water layers. In terms of absolute values the
salinity is also well represented for the Ganges river, although the vertical stratification is also not quite as extensive further
away from the coast. The bathymetry of the Laptev Sea at the Lena river mouth is poorly represented, as WOA data shows
a height increase of the ocean floor at connection of the Sea with the Arctic Ocean, which is not represented in the model
bathymetry. The salinity gradient in the East Chinese Sea is very strong due to currents parallel in the model which are also

10   shown in observations (Ichikawa and Beardsley, 2002). Smaller currents in this region and in the Yellow Sea are however not
well represented in the model. The Congo river has a very steep coastal shelf, and therefore the salinity relatively strong vertical
horizontal gradient, due to little bathymetry induced vertical mixing, as well as heating of the surface waters.



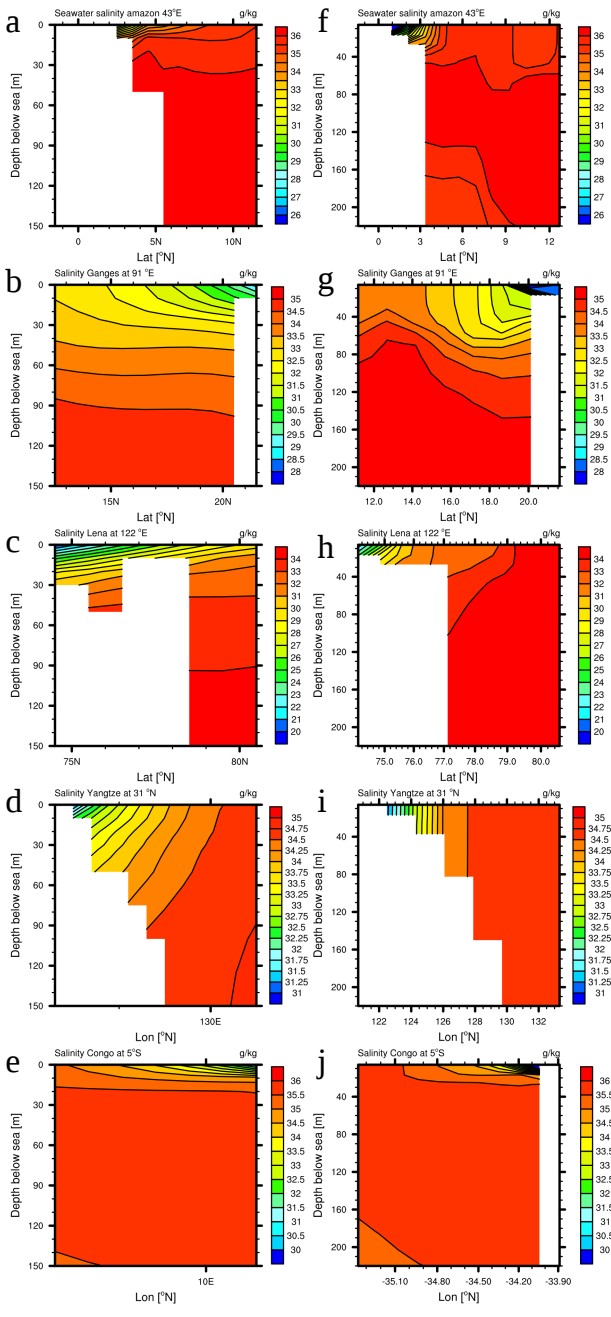

**Figure B1.** Vertical profiles of the coastal bathymetry and salinity, along specific longitudes and longitudes for chosen river mouths. The left column (a-e) is made of WOA salinity profiles, whereas the right column (f-j) of profiles from model simulation RIV. a & f are shelf and salinity profiles at the Amazon river (TWA), b & g for the Ganges river, c & h Lena river, d & i Yangtze river, e & j Congo river.



**Appendix C: Derivation of carbon fluxes in the simplified coupled system**

**C1    Terrestrial fluxes**

**1**. Carbonate and silicate weathering cause a land uptake flux of 280 Tg C yr[-1] from the atmosphere according to the weathering model simulations (Table 3).

**2**. Carbonate mineral weathering causes a lithological carbon release flux of 94 Tg C yr[-1] DIC, as shown in Section 3.2.

**3**. Carbon from terrestrial organic matter originates from the atmosphere (Meybeck and Vörösmarty, 1999). The net carbon uptake by the terrestrial and riverine biology is therefore the same as the lateral organic carbon export, which we derived from NEWS2 (Seitzinger et al., 2010) DOC and POC exports. The net uptake by the terrestrial biology, while taking into account all respiration processes on land and in rivers, is the sum of the lateral POC and DOC exports (249 Tg C yr[-1]).

**4**. The riverine carbon export to the ocean consists of the sum of from weathering and organic matter carbon exports (623 Tg C yr[-1]), minus a loss term due to endorheic rivers (19 Tg C yr[-1]), which results in 603 Tg C yr[-1] (values are rounded).

**C2    Long-term ocean fluxes**

In the model, riverine loads cause oceanic outgassing through the inputs of inorganic C and tDOM, while the inputs of dissolved inorganic nutrients cause a sink of atmospheric carbon through the biological enhancement of C uptake as well as increasing alkalinity while doing so.

The inorganic C is delivered by rivers as 1 mol DIC and 1 mol alkalinity ($HCO_3^-$) and exported as 0.5 mol DIC and 1 mol alkalinity ($CaCO_3$), leaving a surplus of 0.5 mol DIC and 0 mol alkalinity. Increasing the DIC pool without increasing the alkalinity directly increases the dissolved $CO_2$ concentrations, which in its turn causes outgassing:

$$H_2O + Ca^{2+} + 2HCO_3 => CaCO_3 + CO_2 \tag{C1}$$

Equilibrium model outgassing caused by inorganic carbon inputs is therefore 0.5 fold of the riverine DIC loads.

tDOM model inputs, in contrary to POM, are not exported to the sediment. Outgassing from organic material results from the high C:P ratio of tDOM. It is mineralized in the ocean providing dissolved inorganic compounds in the C:P ratio of 2584:1, but the subsequent uptake of the released inorganic compounds happens at a C:P ratio of 122:1, resulting in a DIC overshoot. Since the net alkalinity over the entirety of Equation 15 is also constant, the DIC increase causes a pCO$_2$ increase (*simplified equation):

$$C_{2584}PN_{16}^* => 2584DIC + DIP + 16DIN + ^* => C_{122}PN_{16}^* + 2462CO_2 \tag{C2}$$

The organic outgassing caused by organic matter inputs is therefore 2462/2584 multiplied with tDOM carbon loads.





The riverine loads of DIP and DIN on the other hand cause C uptake through their enhancement of biological primary production. DIC is thereby removed, thus sinking $pCO_2$ (*simplified equation):

$$DIP + 16DIN +^* + 122CO_2 => C_{122}PN_{16}^* \tag{C3}$$

The resulting C uptake from the equation is therefore 122-fold the mole DIP inputs. Additionally, alkalinity is produced

in equation (16), which enhances C uptake. The uptake of DIN and DIP through primary production causes a net alkalinity increase by a factor of Alk:P = 17:1 (Wolf-Gladrow et al., 2007). The alkalinity is exported in a C:Alk ratio of 1:2 through calcium carbonate production (Equation 14). The C uptake enhancement from the alkalinity increase is therefore the 17 * 1/2 - fold of the (bioavailable) DIP loads.

**5**. According to the $CaCO_3$ export equation (Eq C1), half of the DIC input (assuming $HCO_3^-$ is exported to the sediment as

$CaCO_3$ and the other half is outgassed as $CO_2$ in model equilibrium state. The contribution of outgassing caused by inorganic carbon in the ocean is therefore half (0.5-fold) the DIC inputs (366 Tg C yr$^{-1}$) and therefore 183 Tg C yr$^{-1}$ assuming model equilibrium.

**6**. tDOM input C:P rations vastly exceed the oceanic sediment export C:P ratios of organic matter, which causes model equilibrium outgassing in the ocean. Equation C2 shows that for every mol tDOM supplied to the ocean, in model equilibrium

122/2584 of C is exported to the sediment and 2462/2584 of C increases the dissolved $CO_2$ pool, which is outgassed in the long term. The equilibrium outgassing is therefore the tDOM carbon load (134 Tg C yr$^{-1}$) multiplied by 2462/2584, which results in 128 Tg C yr$^{-1}$. In the case of POM, since the C:P ratio of the riverine input (122:1) is the same as the ratio of the export to the sediment in the ocean, there is no effect on the longterm equilibrium outgassing flux.

**7**. Since P has no further sinks or sources in the model other than riverine inputs and sediment burial as organic matter,

in equilibrium the same amount of P supplied by rivers is buried in the sediment. When DIP is taken up by the biology and transformed to organic matter, carbon is also taken up in a mole ratio of C:P = 122:1. Accounting for this uptake through the biological production enhancement by DIP inputs (including bioavailable DIP) from rivers ((1.4 Tg P yr$^{-1}$) results in the uptake of 67 Tg C yr$^{-1}$ throug the biological enhancement. Furthermore, when DIP and DIN are transformed to organic matter as organic matter, an alkalinity increase of 17 mol per mol DIP uptake takes place (Eq. C3, Wolf-Gladrow et al. (2007). This

increase in alkalinity causes the further uptake uptake and export of 2 Tg C yr$^{-1}$, resulting in a total sink of 69 Tg C yr$^{-1}$.

**D1**. We attribute the difference between the equilibrium $CO_2$ flux of 242 Tg C yr$^{-1}$ and the modelled net $CO_2$ flux of 231 Tg C yr$^{-1}$ (=11 Tg C yr$^{-1}$) to the small surface alkalinity increase over the analysis time period.

**8**. The simulated global particulate inorganic C sediment deposition flux in the ocean biogeochemistry model is 188 Tg C yr$^{-1}$.

**9**. The simulated global organic C sediment deposition flux in the ocean biogeochemistry model is 582 Tg C yr$^{-1}$.

**10**. The global modelled DIC flux from the sediment back to the water, which originates from POM remineralization in the sediment, column is 385 Tg C yr$^{-1}$.





**D2**. In model equilibrium the net sediment burial C flux is the total riverine C inputs of 603 Tg C yr$^{-1}$ (**4**) subtracted by the equilibrium outgassing of 242 Tg C yr$^{-1}$ (**5+6-7**), which results in 361 Tg C yr$^{-1}$. The simulated burial flux in the model of 385 Tg C yr$^{-1}$ (**8+9-10**) deviates from the calculated model burial equilibrium flux. Therefore, the drift at the sediment-ocean interface is 24 Tg C yr$^{-1}$ (385 Tg C yr$^{-1}$ - 361 Tg C yr$^{-1}$).

5    **Appendix D: Surface Nutrient profiles**

# Phosphate concentrations

**Figure E1.** Phosphate (DIP) concentrations in OBS (WOA observations), REF and RIV.



# Dissolved silica concentrations

**Figure F1.** Dissolved silica (DSi) concentrations in OBS (WOA observations), REF and RIV.





# Nitrate concentrations



**Figure G1.** Nitrate (DIN) concentrations in OBS (WOA observations), REF and RIV.



*Competing interests.* All contributing authors declare that no competeting interests are present.

*Acknowledgements.* All simulations were performed at the German Climate Computing Center (DKRZ). The research leading to these results has received funding from the European Union's Horizon 2020 research and innovation programme under the Marie Sklodowska-Curie grant agreement No 643052 (C-CASCADES project). We aknowledge constructive comments and suggestions received from Pierre Regnier, Irene Stemmler, Katharina Six and Philip Pika.



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
