# Peer review of "Oceanic CO2 outgassing and biological production hotspots induced by pre-industrial river loads of nutrients and carbon in a global modelling approach"

_Biogeosciences, 2019_

## Referee Comment (RC1) · Anonymous Referee #1 · 10 Jun 2019

This paper assesses the effect of riverine nutrient and C inputs on C cycling in global oceans, and notably on CO2 outgassing in a pre-industrial state. The work includes quantification of riverine inputs of different elements under inorganic and organic forms, regional coastal shelves analysis, and global ocean biogeochemical modelling. Finally, the authors derive a global land-ocean-atmosphere pre-industrial budget, and discuss the effect of riverine inputs and gaps that need to be further addressed in the future (e.g., including volcanic emissions and the effect of shale organic oxidation).

I believe this is a substantial piece of work, improving the current understanding of

global C cycling, and totally fitting Biogeosciences' scope.

I however believe the author should improve the manuscript by addressing the 2 following points, for it to become clearer and meet the journal's standards.

1) In general, the paper is extremely dense and would benefit from being shortened and making it more to the point. Major conclusions should be better highlighted, and repetitions avoided. Also, consider deleting sections with discussions out of the main scope of the study and already addressed in previous work (e.g., on weathering). A few suggestions to improve this point are also listed hereafter, in the specific comments.

2) Further details should be provided on the construction of 2 ocean simulations, RIV and REF. The understanding of the differences between these 2 simulations is key to understand the paper's major conclusions.

——Specific comments——

Abstract

L7-8p1. "Thirdly, we quantify the terrestrial origins... in the framework, ..." -> It is not clear from this sentence if this is purely a modelling exercise of if you are assessing global oceanic C budgets using the coupled land-atmosphere-ocean model.

L11p1. "leads to a global oceanic source of CO2" -> "leads a net global CO2 emission to the atmosphere of". To be consistent with the following sentence, the "source" would be 183+128 TgC/yr (Fig. 11).

L14p1. It is not clear what a sink due to a model drift is without reading the whole paper in depth...

Introduction. The introduction provides a lot of information, some of it not totally relevant to the focus of the study. It should be shortened and re-organized to better highlight current state of knowledge, gaps and how they are addressed in this work.

L15p2. "these knowledge gaps" -> all knowledge gaps are presented in paragraph 7

(L3-23 p4).

Paragraphes 2-6. These present in detail processing in watersheds, gaps in knowledge on tDOM and POM degradation and transfers, etc. These could be substantially shortened, since these are not points tackled in the paper, which waters down the main message/focus of the paper. Parts of it (e.g., uncertainties on degradation of different OM forms, desorption of P as it enters saline waters etc.) could however be used in the Discussion (e.g., subsection 7.2).

Methods

L2p5. "Pre-industrial" could be defined here (state in 1850).

L5p5. "spatially explicit quantification of global riverine loads" -> "spatially explicit quantification of riverine exports to global coasts". It is not clear otherwise if the loads within watersheds are quantified as well.

L10p5. "We first briefly..." -> Description of the Method's content is already described in the previous paragraph. These introductory sentences could be most of the time removed to make the paper shorter and more to the point.

L12p5. Define what you consider like alkalinity here.

L20-32p5. This is described in detail in the subsections; it could be deleted to avoid repetitions.

Fig.1. Precise that the C from weathering sources is DIC (OC assumed to originate only from the uptake of atmospheric CO2).

L12-14p7. Doesn't river damming affect POM loads between pre-industrial and the 1970s? This point could be considered later in the discussion on river loads.

L13p8. "Pre-industrial runoff..." -> this is already explained L1-5p7, not necessary. This repetition occurs several times throughout the paper.

L24p8. Are the soil types listed here those with typically low erosion rates? Is the 0.1 factor also used for wetlands and areas with a high groundwater table? If yes, what definition did you use for "high water table"?

L4p9. What do the 1.6 TgP/yr used here correspond to? Fertilizer P in surface runoff reaching rivers? Hart et al. (2004) report an annual fertilizer consumption of 873 TgP in 1913.

L7-9p9. Why is it reasonable to assume P equilibrium in soils at pre-industrial state (besides that state-of-the-art models usually use this initialization)?

L9-11p9. The assumption that spatial distribution of P river loads is the same in pre-industrial and 1970 is quite strong. For example, agriculture was probably much more developed in North America and Western Europe than in Asia at the beginning of the 20th century. Implications of this assumption should be discussed later in the discussion on river loads. Why not use load distribution from models describing earlier states (e.g., Beusen et al., 2016)?

L23-24p9. Molar C:P ratios are already provided earlier. You can just say L19 that P is incorporated in organic matter accordingly to C:P molar ratios for tDOM and POM.

L6-12p10. Precise here that DIN export was calculated by subtracting the part contained in organic matter. Was DFe export equal to Fe inputs to catchments?

L11-16p11. This is not totally clear. Do you assume that only HCO3- affects alkalinity and that DIC is only HCO3- as well (Alk:DIC = 1:1)?

Sub-subsection 2.1.5 Silica. Why not use the same weathering model type as for P (Hartmann et al., 2011)?

Sub-subsection 2.2.1 Ocean biogeochemistry. A scheme with model state variables and processes would be helpful here (as well as a description of biogeochemical processing equations and parameters as Supplementary Information).

L22p13. How did you choose the 0.003 d-1 value for tDOM degradation? 0.008 d-1 for the oceanic DOM is also in the literature range for tDOM degradation.

2.2.3 Pre-industrial ocean biogeochemistry model simulations. Please provide more details here, since the distinction between the REF and RIV simulations is key to understanding the paper's results. In the REF simulation, are input fluxes (per surface area?) globally homogeneous? Or do they compensate sediment losses to reach equilibrium locally (at the cell scale)? In the RIV simulation, are there also open ocean surface inputs in addition to river inputs?

L1-6p14. What do you call quasi-equilibrium? How were the lengths of the different simulation chosen? Why did you perform a succession of 3 runs for the RIV simulation? Does "standard simulation" (L4) refer to REF? Are the 100-year means (output results) calculated on the last 100 simulated years for each simulation? What are the simulation timesteps? It is mentioned that water loads vary inter-annually based on runoff inputs from OMIP. Are the ocean physics' inter-annual variations modelled with the same patterns for every simulated year?

Subsection 3.1 Runoff, precipitation and temperature patterns. Hydrology model performance is not the scope of the paper; this section seems to water down once more the message/goal of the study. The scaling of the MPI-ESM runoff should be detailed earlier, in the Methods section. How is the factor 1.59 chosen? Why is the runoff not scaled to the OMIP one, to be fully consistent with the freshwater inputs?

L20-24p16. For which periods were these Si loads from the literature estimated?

L26-32p16. Comparison with Mackenzie et al. (1998) does not seem necessary here, especially since they assess only the TIC load.

Subsection 4.1 Global loads in the context of published estimates L2p19. Does the Fe-P load given in Table 3 only corresponds to the fraction that is desorbed when entering the estuary?

Table 3p19. Precise for the POP comparison that the 5.9 value (for 1970) is for PP, and not only POP.

L20-23p20. Doesn't the potential increase in Si retention at the end of the 20th century mainly concern particulate forms?

L20-26p21. Please better organize text.

L4p22. Could the discrepancies also be due to the fact that weathering formalisms are less adapted to Arctic regions?

5.1 Ocean state – An increased biogeochemical coastal sink

L14-15p25. "There is a higher carbon load originating from organic matter, since tDOM C:P ratio is higher than the oceanic DOM C:P ratio" -> This explanation is not straight-forward, since the estimation of DOC is not related to P in the model.

L15p25-L2p26. Comparison with other studies was already discussed earlier.

L17-19p26. Isn't the effect of large oxygen minimum zones on DIN concentrations also visible in WOA dataset?

What explains the major differences in DIP, DIN and DSi concentrations in the Southern Ocean?

Table 5.p27. Do N inputs include N2 uptake from the atmosphere in the two simulations?

L12-14p28. Comparison of Arctic concentrations to the WOA database could be presented earlier in text, with the rest of the comparisons.

7.1 Rivers in an Earth System Model setting. Paragraph 2 (L12-17p36). These improvements could further avoid strong assumptions, such as globally constant N:P ratios for river inputs, and on the spatial distribution of non-weathering sources.

8. Summary and conclusions

L11-14p37. Is the need for a runoff scaling factor a major conclusion of this work?

L17-21p37. "Even for present-day..." -> comparison with literature is already discussed earlier, this could be removed.

L26-27p37. "which have been identified to be more strongly controlled by extreme hydrological events than other C species" -> Was this considered in the study? What does it imply? It seems that this belongs to some discussion (that should be more developed if added).

L17p38. "despite previously being sinks" -> be more explicit by explaining that this is without accounting for river inputs, and not "previously" in a temporal way.

L27-29p38. Interhemispheric C transfers are already mentioned earlier in this section (L1-3p38). Please focus in this last paragraph only on the major points that this study shows have to be included in ESMs to better assess land-ocean-atmosphere C transfers.

Appendix C

L4p42. "Table 3" -> "Table 2". How do you get to the 280 Tg/yr of CO2 drawdown? Some calculations could be shorter explained (e.g., CO2 emissions explained L16-20p42 and L9-12p43).

—— Other minor comments ——

L5p1. "in a regional shelf analysis" -> "in regional shelf analysis"; there are more than one.

L10-11p1. "total C" -> "global C"

L12p1. "which is largely a result of a source from" -> "which mainly results from"

L22p1-L2p2. The last sentence could be split in 2.

L13p2. "exported to the sediment" -> "stored in the sediment"

L18p2. Define NPP

L25p5. "their its ratios" -> "their ratios"

L30p5. "a fraction P" -> "a fraction of P"

L4p11. "and that 2 HCO3-" -> "and of 2 HCO3-"

L11p12. "dynamical nitrogen fixation through cyanobacteria" -> "dynamic N fixation by cyanobacteria"

L14p13. "its consistence" -> "its composition"

L11p17. "much stronger gradients of variation" -> "much higher spatial variability"

L21p17. End of sentence is missing or "which" should be deleted.

L17p20. Start new sentence at "Dürr et al. . ."

L2p21. "and to a framework" -> "and could be fed/incorporated into a framework"

L2p30. "compensate" -> "counterbalance"

L15p30. "due higher" -> "due to higher"

L18 p36 "consistence" -> "composition"

L19-25p43. Check parentheses.

---

## Referee Comment (RC2) · Anonymous Referee #2 · 19 Jun 2019

The manuscript addresses an important question about material transport across the land-ocean aquatic continuum, and is of particular interest given its global application. I appreciate the substantial amount of work presented here, which includes a synthesis of existing methods to derive a global data set of riverine sources of nutrients and carbonate species to the ocean, long-term simulations of a global ocean-biogeochemical model, and analyses of CO2 outgassing hotspots and the origins and fates of riverine carbon from land/atmosphere to the ocean/atmosphere. I am particularly impressed that the authors were able to run the global model simulations for several thousand

years long (even though the model is relatively coarse). The scope of this study is certainly appropriate for publication in Biogeosciences, however the manuscript in this current format requires clarifications in some places while in others the text needs to be shortened and/or streamlined in order to avoid distraction and help readers better capture the key points of this study.

My major concern about the study is the comparison of carbon (and other) budgets between the standard simulation (RIV, what does it stand for?) and reference simulation (REF), which led to most key conclusions made in the manuscript. The authors described REF as a configuration where no river inputs are added to the ocean but "the burial loss of biogeochemical tracers was compensated by a global homogenous flux to the surface ocean", such that REF is "fully constrained by the loss of the sediment layer" (P13L26-29). This REF configuration seems a bit odd or at least not so clear to me. Is this addition of homogenous flux occurs during the same or next time step? Why choosing to use a homogenous flux instead of a spatially varying flux that directly compensates for the bottom loss at the same location? Wouldn't this framework to some degree arbitrarily homogenize the resource distribution across the ocean? Why adding the flux at the surface instead of evenly distributing it throughout the water column? How about distributing this flux only along the coastline (acting as a riverine source)?... It seems that in this framework carbon (and other materials) is being relocated from the bottom to the surface and from some places to others without any explicit transport processes involved. And this would potentially make a HUGE impact on NPP and CO2 outgassing patterns regardless of riverine inputs. Also, what are other inputs to the REF beside this surface flux? Does REF also include N2 fixation? How is carbon synthesis associated with N2 fixation being handled in the model? These are important details for making the conclusions of the manuscript and should be clearly described.

Depending on these details and whether the comparison between RIV and REF is justifiable, I recommend either a major (which would require a re-configuration and rerun of the 5000-year REF experiment) or a moderate revision (which would be focused on streamlining and shortening the text plus clarification on some details as suggested below) of the paper before considering it for publication in Biogeosciences.

Minor comments:

P2L13: and also released to the atmosphere

P2L19-P2L6: introductory information in these a few paragraphs needs to be streamlined. I suggest shortening it to 5-8 lines and expand Fig.1 to include more details on the processes to be considered or discussed.

P3L9: without reading the referenced literature, it is not so clear to me why riverine DIC would cause $CO_2$ outgassing. Is it due to solubility change? Would riverine DOC/POC be also, if not more, likely to cause $CO_2$ outgassing as a result of microbial respiration?

P3L20: where does photodegradation most likely occur? In the rivers? Coastal margins? Is this process not reflected in the extremely high C:P ratio (2584:1) considered here?

P3L24: why does POM control the availability of nutrients? By being remineralized?

P4L13: which 10 years? Contemporary?

P5L8: Is there a particular reason to use 250m rather than the more commonly defined 200m?

P7L18: C:P=1000:1 here but 2584:1 in P9L23.

P10 section 2.1.3: why not deriving N:P ratios for different rivers from Global NEWS data set? Or at least to make a comparison with?

P12L7: "river freshwater model" -> is this the MPI-ESM model mentioned in P7L3?

P12L9-18: need some clarification on the biogeochemical model configuration and references for different versions of the model. The original model is described in Ilyina

et al. 2013, and the version being used in this study is the same with Mauritsen et al., 2018? The major changes include cyanobacterial N2 fixation and incorporation of DOM? Was the DOM improvement also made in Mauritsen et al., 2018? What about Paulsen et al., 2017 and Six and Maier-Reimer, 1996? Are these studies relevant to the HAMOCC model development? Also, is the model (e.g. photosynthesis) N or P or C based? Are there different pools for D/POC, D/PON, D/POP, etc? Beside river inputs, are there other inputs from e.g. atmosphere deposition? Is N2 fixation also a carbon input or just N? These important details, particularly on the sinks and sources of N,P,C, need to be provided here.

P13L22: first 0.003 is not just "slightly" slower than 0.008, and second, how did you choose these two values?

Suggestions on shortening the text:

Abstract: focus on the riverine impact on CO2 outgassing and NPP hotspots in the ocean and leave the details of the river export (e.g numbers in P1L8) to the result section. P1L15-19 can be removed or shortened to one sentence.

P2L19-P2L6: can be shortened as suggested above.

P15 Section 3.1 is not a focus of this study and does not seem too necessary to me. P16 Section 3.2 and P18 Section 4.1 can be reasonably shortened to half of its present length. I appreciate the effort made here to validate the results but this is a bit too much details. The readers need to reach P25 Section 5 and P34 Section 6 for the key points before being distracted by these details.

---

## Author Comment (AC1) · 18 Jul 2019

**Response to Review#1**

**General Comments**

This paper assesses the effect of riverine nutrient and C inputs on C cycling in global oceans, and notably on CO2 outgassing in a pre-industrial state. The work includes quantification of riverine inputs of different elements under inorganic and organic forms, regional coastal shelves analysis, and global ocean biogeochemical modelling. Finally, the authors derive a global land-ocean-atmosphere pre-industrial budget, and discuss the effect of riverine inputs and gaps that need to be further addressed in the future (e.g., including volcanic emissions and the effect of shale organic oxidation). I believe this is a substantial piece of work, improving the current understanding of global C cycling, and totally fitting Biogeosciences' scope.

**Response: We would like to thank the reviewer for the comments and very helpful contributions to improving the manuscript.**

**Major Comments**

1) In general, the paper is extremely dense and would benefit from being shortened and making it more to the point. Major conclusions should be better highlighted, and repetitions avoided. Also, consider deleting sections with discussions out of the main scope of the study and already addressed in previous work (e.g., on weathering). A few suggestions to improve this point are also listed hereafter, in the specific comments.

**Response: We think that the suggestion to streamline and shorten the manuscript is a good point that has been mentioned by both reviewers, and we will therefore consider this for the revised manuscript.**

**We suggest a revised manuscript with a shortening of the introduction (section 1), the elimination of repetitions in the methods section (section 2), and the merging and shortening of sections 3. and 4.**

2) Further details should be provided on the construction of 2 ocean simulations, RIV and REF. The understanding of the differences between these 2 simulations is key to understand the paper's major conclusions.

**Response: While this is partly done in paragraphs in the methods (section 2.2.3) and in the results and discussion (section 5.1), we also see the need to further highlight the main differences between REF and RIV, and discuss the analysis strategy. These are two things: Firstly, the geographical differences of inputs to the ocean (riverine inputs vs inputs to the open ocean), and secondly the increased carbon inputs to the ocean due to the transfer of carbon from land to ocean.**

**We will more clearly state these points in the revised manuscript. We also believe that the improvements from comment 1) will help improve the overview of these differences in the revised manuscript.**

**Minor comments**

L7-8p1. "Thirdly, we quantify the terrestrial origins. . . in the framework, . . ." -> It is not clear from this sentence if this is purely a modelling exercise of if you are assessing global oceanic C budgets using the coupled land-atmosphere-ocean model.

**Response: Here, we solely budget for the carbon fluxes from the "stand-alone" models, which are forced with the same pre-industrial atmospheric state. The models do not however give a feedback, which alters the atmospheric state, since the atmospheric CO2 concentration would decline due to the carbon imbalance of the atmosphere (Figure 11). On the terrestrial side, the carbon fluxes are derived from the weathering and the organic carbon export models. On the ocean side, they are derived from the the ocean biogeochemistry model HAMOCC. We therefore do not take into account feedbacks in the system, as would be done in a fully coupled land-atmosphere-ocean model, but if so, one would need to consider further carbon outgassing sources (long-term volcanic emissions, shale oxidation), in order to have a balanced budget for the pre-industrial atmosphere.**

**We will clarify this sentence in the revised manuscript.**

L11p1. "leads to a global oceanic source of CO2" -> "leads a net global CO2 emission to the atmosphere of". To be consistent with the following sentence, the "source" would be 183+128 TgC/yr (Fig. 11).

**Response: We will implement this correction in the revised manuscript.**

L14p1. It is not clear what a sink due to a model drift is without reading the whole paper in depth. . .
**Response: This is a good point, and we think that the word "model disequilibrium" is more understandable for the reader.**

Introduction. The introduction provides a lot of information, some of it not totally relevant to the focus of the study. It should be shortened and re-organized to better highlight current state of knowledge, gaps and how they are addressed in this work.

**Response: We will improve the structure and points of focus of the introduction in the revised manuscript. We will shorten the paragraphs p2.l4-p2.l18 and p3.l7-p3.l28 and increase the focus of the paragraphs.**

L15p2. "these knowledge gaps" -> all knowledge gaps are presented in paragraph 7.

**Response: We agree that this part of the sentence might be redundant and therefore will remove it.**

Paragraphes 2-6. These present in detail processing in watersheds, gaps in knowledge on tDOM and POM degradation and transfers, etc. These could be substantially shortened, since these are not points tackled in the paper, which waters down the main message/focus of the paper. Parts of it (e.g., uncertainties on degradation of different OM forms, desorption of P as it enters saline waters etc.) could however be used in the Discussion (e.g., subsection 7.2).

**Response: We understand the point of view of the reviewer, and also think the paragraphs can be shorted. However, some of the information within these paragraphs is also vital in order to understand the assumptions chosen later in the methods section. For instance, due to the previous degradation of tDOM within rivers, we assume it is less reactive once it reaches the ocean than oceanic DOM.**

**In the revised manuscript, we will shorten these paragraphs and increase their focus.**

L2p5. "Pre-industrial" could be defined here (state in 1850).

**Response: We will define pre-industrial here in the revised manuscript.**

L5p5. "spatially explicit quantification of global riverine loads" -> "spatially explicit quantification of riverine exports to global coasts". It is not clear otherwise if the loads within watersheds are quantified as well.

**Response: The loads are indeed quantified for every watershed. We agree this might be somewhat unclear here and will correct this in the revised manuscript.**

L10p5. "We first briefly..." -> Description of the Method's content is already described in the previous paragraph. These introductory sentences could be most of the time removed to make the paper shorter and more to the point.

**Response: We will remove this sentence since it is a repetition.**

L12p5. Define what you consider like alkalinity here.

**Response: Alkalinity (which is carbonate alkalinity in the case of our derived riverine loads) will be more clearly defined here in the revised manuscript.**

L20-32p5. This is described in detail in the subsections; it could be deleted to avoid repetitions.

**Response: We believe it is important to give a brief overview of all derived loads before addressing the individual elements. Otherwise we fear the reader will be missing the "big picture" if we start with individual details of every compound load. We will however remove some parts from this paragraph, which might be unnecessarily repeating information in the revised manuscript.**

Fig.1. Precise that the C from weathering sources is DIC (OC assumed to originate only from the uptake of atmospheric CO2).

**Response: This will be added to the figure in the revised manuscript.**

L12-14p7. Doesn't river damming affect POM loads between pre-industrial and the 1970s? This point could be considered later in the discussion on river loads.

**Response: We assume in the whole study that the organic matter loads have remained constant globally since the pre-industrial time period. While increased organic matter supplies to catchments have been suggested in literature (for instance in Regnier et al. 2013), increased retention and remineralization (for instance due to damming, Maavara et al., 2017) has also been reported. There is however yet a study to spatially quantify these combined effects, while comparing the pre-industrial time-frame to the present day. This is briefly already discussed in 4.1. (l25p20.), and we believe a more comprehensive discussion of these combined anthropogenic influences escape the scope of our study.**

L13p8. "Pre-industrial runoff..." -> this is already explained L1-5p7, not necessary. This repetition occurs several times throughout the paper.

**Response: We will remove this repetition in the revised manuscript.**

L24p8. Are the soil types listed here those with typically low erosion rates? Is the 0.1 factor also used for wetlands and areas with a high groundwater table? If yes, what definition did you use for "high water table"?

**Response: The soil types listed here are indeed the ones assumed to have very low weathering rates. The wetland areas are the Gleysols. These (along with the other soil types) are defined in the Harmonized World Soil Database (Fao et al., 2009). The description found in the database is that gleysols are soils with permanent or temporary wetness near the surface. The derivation of the soil shielding factor is extensively described in Hartmann et al. (2014). In our opinion, further details escape the scope of our study.**

L4p9. What do the 1.6 TgP/yr used here correspond to? Fertilizer P in surface runoff reaching rivers? Hart et al. (2004) report an annual fertilizer consumption of 873 TgP in 1913.

**Response: This number was wrongly cited. The 1.6 Tg P to catchments was derived from Figure 3 in Beusen et al. (2016), which on its side derived from inputs described in Hart et al. (2004), as is cited in the corresponding technical manuscript (Beusen et al., 2015). Assuming 873 million tonnes of P inputs of fertilizer for the year 1913 (Hart et al., 2004), this relatively high value seems plausible.**

L7-9p9. Why is it reasonable to assume P equilibrium in soils at pre-industrial state (besides that state-of-the-art models usually use this initialization)?

**Response: We agree this assumption is a limitation of our study. For instance, Filipelli et al. (2008) suggest an increase of P inputs to catchments due to increased soil erosion caused by deforestation and land use change. This likely already had an impact on land-ocean fluxes prior to industrialization, although it is completely unknown how larg these fluxes could be. This point goes hand-in-hand with our assumption that the organic matter fluxes remain near constant, which is at least consistent. In the revised manuscript, we will add this point to 7.1 when discussing limitations and aspects that could be potentially improved.**

L9-11p9. The assumption that spatial distribution of P river loads is the same in pre-industrial and 1970 is quite strong. For example, agriculture was probably much more developed in North America and Western Europe than in Asia at the beginning of the 20th century. Implications of this assumption should be discussed later in the discussion on river loads. Why not use load distribution from models describing earlier states (e.g., Beusen et al., 2016)?

**Response: This is a good point, we however are not aware of available data from which such a differing distribution of the anthropogenic inputs from the present-day could be constructed. It is also not clear from the Beusen et al. (2016) study if and how the study assumes differing anthropogenic input distributions to the present day. We however suggest shortly investigating the differences between the approach chosen here and the approach chosen in Beusen et al. (2016) in a supplementary information section, if this data is available. We will also discuss this uncertainty in the discussion section 7.1.**

L23-24p9. Molar C:P ratios are already provided earlier. You can just say L19 that P is incorporated in organic matter accordingly to C:P molar ratios for tDOM and POM.

**Response: We will correct this in the revised manuscript to avoid repetition.**

L6-12p10. Precise here that DIN export was calculated by subtracting the part contained in organic matter. Was DFe export equal to Fe inputs to catchments?

**Response: The Fe composition of the organic matter is also subtracted from the Fe inputs to the catchments, which results in DFe. We will clarify this in the revised manuscript.**

L11-16p11. This is not totally clear. Do you assume that only HCO3- affects alkalinity and that DIC is only HCO3- as well (Alk:DIC = 1:1)?

**Response: We asumed that only $HCO_3^-$ release from weathering affect the alkalinity and that the Alk:DIC ratio is 1:1. This is stated and justified in the paragraph, but we will attempt to make the paragraph more understandable.**

Sub-subsection 2.1.5 Silica. Why not use the same weathering model type as for P (Hartmann et al., 2011)?

**Response: In theory, this would be possible, but the model would first need to be calibrated for Si. Since the Beusen et al. (2009) model is already calibrated for Si, we used this one out of simplicity.**

Sub-subsection 2.2.1 Ocean biogeochemistry. A scheme with model state variables and processes would be helpful here (as well as a description of biogeochemical processing equations and parameters as Supplementary Information).

**Response: We will add a scheme which illustrates the main processes represented in HAMOCC in the revised manuscript. Regarding the equations, this is too extensive in the case of this manuscript. We refer to Ilyina et al. (2013) for the model description, equations and parameters, with more recent model developments described in Mauritsen et al. (2019). If necessary, we can provide a list of changed parameters in comparison to the Ilyina et al. (2013) study in a Supplementary Information document.**

L22p13. How did you choose the 0.003 d-1 value for tDOM degradation? 0.008 d-1 for the oceanic DOM is also in the literature range for tDOM degradation.

**Response: The assumption chosen here was that the tDOM entering the ocean at the river mouths is less reactive than the "freshly produced" oceanic DOM, since tDOM is reported to have been strongly degraded previously during its transport in rivers. This is already stated in paragraph p13.l20-p13.l24.**

2.2.3 Pre-industrial ocean biogeochemistry model simulations. Please provide more details here, since the distinction between the REF and RIV simulations is key to understanding the paper's results. In the REF simulation, are input fluxes (per surface area?) globally homogeneous? Or do they compensate sediment losses to reach equilibrium locally (at the cell scale)? In the RIV simulation, are there also open ocean surface inputs in addition to river inputs?

**Response: In short, the REF inputs are added globally homogeneously (per surface area) to the ocean surface. This represents oceanic inputs by (to a major part) passing the coastal ocean and does not at all represent geographic riverine inputs. Their magnitudes are dictated by losses to the sediment: every few hundreds of years, the averaged global burial loss of biogeochemical compounds was computed, and these were the fluxes added to the surface ocean. They are**

indeed homogeneous per surface area. In the RIV simulation, there is no open ocean surface inputs, since these were replaced by riverine inputs, derived as shown in the previous sections of the manuscript.

**We will discuss and clarify these points extensively in the revised manuscript.**

L1-6p14. What do you call quasi-equilibrium? How were the lengths of the different simulation chosen? Why did you perform a succession of 3 runs for the RIV simulation? Does "standard simulation" (L4) refer to REF?

**Response: The lengths of the simulations were chosen accordingly to the model state, meaning the simulation were performed until no strong drifts of model variables was remaining. Since biogeochemical processes in the ocean, and especially in the sediment, likely need larger simulation time-periods to perfectly equilibrate than is feasibly possible with current computing resources, there will however always remain a small drift in model variables in such simulations. Therefore, this is a state of quasi-equilibrium and not a perfect equilibrium state.**

**The 3 sequential simulations for the RIV simulation were done in order to achieve a more stable state in the ocean sediment. The motivation here is that it is much less expensive in terms of computing power to simulate the sediment separately from the ocean, and since the time-scales of processes taking place in the sediment are much longer than in the water-column, it makes sense to perform simulations for the sediment alone once the oceanic water column has reached a state close to equilibrium. We therefore firstly perfomed a simulation of 4'000 years including both the water column and sediment biogeochemistry, until the global particulate fluxes from the water column to the sediment were approximately stable. Then simulations of the sediment component of the model, while being given the stable global particulate fluxes from the previous simulation. Finally, the sediment state was re-coupled to the ocean water-column for the final 2000 year simulation.**

**We believe the full description of the model spin-up mentioned above are too technical for the scope of this journal, and would substantially unnecessarily lengthen the manuscript.**

Are the 100-year means (output results) calculated on the last 100 simulated years for each simulation? What are the simulation timesteps?

**Response: The last 100 year mean of both REF and RIV were used to the analysis. Using a 100 year mean prior to this would not affect the results much. The simulation timestep is one hour.**

It is mentioned that water loads vary inter-annually based on runoff inputs from OMIP. Are the ocean physics' inter-annual variations modelled with the same patterns for every simulated year?

**Response: The parameters within the physical circulation model MPI-OM remain the same. We would like to point out that the model simulates the response of the ocean physics dynamically to the atmospheric state, from which the information is given to the model for every time-step. There is therefore inter-annual variation in the physical features of the ocean, and even variation at every model time-step.**

**It escapes the scope of our study to completely explain mechanisms of such a circulation model in the scope of our study, but this has been done before and we would like to refer to the cited references (Junclaus et al., 2013 and Mauritsen et al., 2019).**

L20-24p16. For which periods were these Si loads from the literature estimated?

**Response: These estimates are for the present day. We will mention this in the revised manuscript.**

L26-32p16. Comparison with Mackenzie et al. (1998) does not seem necessary here, especially since they assess only the TIC load.

**Response: We agree and will therefore remove this comparison in the revised manuscript.**

Subsection 3.1 Runoff, precipitation and temperature patterns. Hydrology model performance is not the scope of the paper; this section seems to water down once more the message/goal of the study. The scaling of the MPI-ESM runoff should be detailed earlier, in the Methods section. How is the factor 1.59 chosen? Why is the runoff not scaled to the OMIP one, to be fully consistent with the freshwater inputs?

**Response: In order to shorten the manuscript, we will merge sections 3 with section 4, while shortening both with only the most central information for the next parts of the manuscript remaining. Factor 1.59 was chosen in order to scale the average of Fekete et al. (2002) and Dai and Trenberth (2002) (l24.p15). In the revised manuscript, this will be more clearly stated.**

Subsection 4.1 Global loads in the context of published estimates L2p19. Does the Fe-P load given in Table 3 only corresponds to the fraction that is desorbed when entering the estuary?

**Response: The Fe-P loads refer to desorbed P loads both for the given modelled estimates (Model. Global load), as well as for the Compton et al. (2000) values. This will be stated in the Table 3 Caption in the revised manuscript.**

Table 3p19. Precise for the POP comparison that the 5.9 value (for 1970) is for PP, and not only POP.
**Response: This will be added in the revised manuscript.**

L20-23p20. Doesn't the potential increase in Si retention at the end of the 20th century mainly concern particulate forms?

**Response: This is not correct. Global reservoirs have also been suggested to retain a significant amount of DSi (Lauerwald et al, 2013; Maavara et al., 2014). This is due to in-reservoir formation of biogenic silica, which is then in turn also increasingly retained.**

L20-26p21. Please better organize text.

**Response: We will improve the organization of the text in the revised manuscript.**

L4p22. Could the discrepancies also be due to the fact that weathering formalisms are less adapted to Arctic regions?

**Response: This is indeed very likely, with weathering mechanisms in the Arctic being likely more complex than is modelled here (for instance due to permafrost, Hartmann et al., 2014).**

5.1 Ocean state – An increased biogeochemical coastal sink L14-15p25. "There is a higher carbon load originating from organic matter, since tDOM C:P ratio is higher than the oceanic DOM C:P ratio" -> This explanation is not straight-forward, since the estimation of DOC is not related to P in the model.

**Response: This is correct but the DOP is coupled to the DOC estimation. In (long-term) model equilibrium, for every mol P exported to the sediment, 122 mol C is also exported. For the river**

inputs however, for every mol P 3583 mol C are added. Therefore there is an accumulation of DIC when tDOM is mineralized and POM/DOM is subsequently produced by the oceanic biology with the released DIP, which leads to higher $pCO_2$ and outgassing. This is explained from l20p42, but we will attempt add clarify to the statement on L14-15p25.

L15p25-L2p26. Comparison with other studies was already discussed earlier.

**Response: The central point here is that the carbon inputs are increased in RIV comparison to REF., in agreement with previous estimates. This is why carbon outgassing takes place and therefore we believe this is central information here, even if it has been stated before.**

L17-19p26. Isn't the effect of large oxygen minimum zones on DIN concentrations also visible in WOA dataset?

**Response: This is indeed the case and we can mention this in the revised manuscript.**

What explains the major differences in DIP, DIN and DSi concentrations in the Southern Ocean?

**Response: This is likely due to the relatively poor representation of the complex ocean circulation of the Southern Ocean in global ocean circulation models. An improved representation would require high enough resolution to better represent vertical mixing taking place in the Southern Ocean and a better representation of the effects from storms.**

Table 5.p27. Do N inputs include N2 uptake from the atmosphere in the two simulations?

**Response**: **Both simulations consider dynamical $N_2$ fixation by cyanobacteria as well as (natural) nitrogen deposition. This is stated in the methods p12.l11: "The changes were made to incorporate dynamical nitrogen fixation through cyanobacteria (Paulsen et al., 2017), .."**

L12-14p28. Comparison of Arctic concentrations to the WOA database could be presented earlier in text, with the rest of the comparisons.

**Response: We agree and will correct this in the revised manuscript.**

7.1 Rivers in an Earth System Model setting. Paragraph 2 (L12-17p36). These improvements could further avoid strong assumptions, such as globally constant N:P ratios for river inputs, and on the spatial distribution of non-weathering sources.

**Response: These are both very good points and we will add these to the subsection in the revised manuscript.**

L11-14p37. Is the need for a runoff scaling factor a major conclusion of this work?

**Response: Although this is a more technical conclusion, this is quite significant: in order to have dynamically changing riverine loads in a fully coupled (land-ocean-atmosphere) model, the runoff would first need to be addressed.**

L17-21p37. "Even for present-day..." -> comparison with literature is already discussed earlier, this could be removed.

**Response: We agree and the sentence will be removed in the revised manuscript.**

L26-27p37. "which have been identified to be more strongly controlled by extreme hydrological events than other C species" -> Was this considered in the study? What does it imply? It seems that this belongs to some discussion (that should be more developed if added).

**Response: The NEWS2 model results, which we derive POC exports from, calculate the year-means of these exports. Therefore, these extreme events are not at all grasped in the model. This will be shortly discussed additionally in the revised manuscript.**

L17p38. "despite previously being sinks" -> be more explicit by explaining that this is without accounting for river inputs, and not "previously" in a temporal way.

**Response: We will correct this statement in the revised manuscript in order to improve clarity.**

L27-29p38. Interhemispheric C transfers are already mentioned earlier in this section (L1-3p38). Please focus in this last paragraph only on the major points that this study shows have to be included in ESMs to better assess land-ocean-atmosphere C transfers.

**We will remove this second discussion interhemispheric C transfer and add an outlook in the conclusions.**

**References (excluding cited references in the manuscript):**

Beusen, A. H. W. Et al.: Coupling global models for hydrology and nutrient loading to simulate nitrogen and phosphorus retention in surface water – description of IMAGE–GNM and analysis of performance, Geosci. Model Dev., 8, 4045-4067, https://doi.org/10.5194/gmd-8-4045-2015, 2015.

Lauerwald et al.: Retention of dissolved silica within the fluvial system of the conterminous. Biogeochemistry, 112,637-659. 2013.

---

## Author Comment (AC2) · 19 Jul 2019

**Response to Review#2**

**General Comments**

The manuscript addresses an important question about material transport across the land-ocean aquatic continuum, and is of particular interest given its global application. I appreciate the substantial amount of work presented here, which includes a synthesis of existing methods to derive a global data set of riverine sources of nutrients and carbonate species to the ocean, long-term simulations of a global ocean-biogeochemical model, and analyses of CO2 outgassing hotspots and the origins and fates of riverine carbon from land/atmosphere to the ocean/atmosphere. I am particularly impressed that the authors were able to run the global model simulations for several thousand years long (even though the model is relatively coarse). The scope of this study is certainly appropriate for publication in Biogeosciences, however the manuscript in this current format requires clarifications in some places while in others the text needs to be shortened and/or streamlined in order to avoid distraction and help readers better capture the key points of this study.

**Response: We would like to thank the reviewer for their helpful comments and criticisms, and are also pleased with the acknowledgement of effort put into synthesizing the state of knowledge of land-ocean biogeochemical exports, as well as our computational expenses to simulate such long time-periods.**

**We agree the manuscript should be streamlined and shortened to a certain extent to improve the reader's understanding of the main points.**

**Major Comments**

**Authors note: The comments found in the next paragraph of the reviewer's comment will be addressed individually.**

My major concern about the study is the comparison of carbon (and other) budgets between the standard simulation (RIV, what does it stand for?) and reference simulation (REF), which led to most key conclusions made in the manuscript.

**Response: In short, the reference simulation REF represents a previous model version of the ocean biogeochemistry model, which is lacking in terms of its constraints and geographical locations of "riverine" inputs. In the simulation RIV, we consider the magnitudes of the pre-industrial riverine inputs as well as their geographic input locations to the best of our knowledge. The focus is to analyze how adding riverine fluxes plausibly in terms of magnitudes, as well as at their correct geographical location (RIV) affected the ocean's biogeochemical state. This is done in comparison to REF, where these inputs are added to the open ocean. As a note, the overarching goal of having biogeochemical inputs in REF is to compensate losses to the sediment; without these inputs, all biogeochemical variables would strive to an equilibrium state equal to zero. This however does not mean that these inputs must be added at the sediment interface (nor is this a more "realistic" solution to the problem), we will revisit this as a response to a later comment.**

**We see the need to improve the explanations of the differences between the simulations REF and RIV in section 2.2.3., as well as in the explanations in Section 5.1 from p25.l10-p26.l11 on how the differences in REF and RIV should affect global ocean biogeochemistry variables (->what is our analysis strategy), since these points were not fully clear for both reviewers. This should be done in the revised manuscript. Cutting unneeded information, as pointed out by the reviewer, will also contribute to a better understanding of these points.**

The authors described REF as a configuration where no river inputs are added to the ocean but "the burial loss of biogeochemical tracers was compensated by a global homogenous flux to the surface ocean", such that REF is "fully constrained by the loss of the sediment layer" (P13L26-29). This REF configuration seems a bit odd or at least not so clear to me.

**Response: We would like to note that the REF simulation is the state-of-the-art previous model version, which is lacking in terms of its representation of terrestrial nutrient and carbon inputs to the ocean, since it considers these inputs solely at magnitudes to compensate for sediment losses, and they are added to the open ocean. We improve the model in RIV in order to represent the riverine loads magnitudes and geographical locations. We did not design the REF model setup for this study in particular, but it is adequate in order to assess the impacts of riverine loads added at their geographical river mouths in comparison to biogeochemical inputs added to the open ocean.**

Is this addition of homogenous flux occurs during the same or next time step?

**Response: The loss of biogeochemical compounds to the sediment is not computed dynamically within the model, but we approximated the sediment loss fluxes for a 100 year mean around every 1000 years of the REF simulation. The resulting fluxes are added to the model at every timestep.**

**We will shortly mention this in the revised manuscript.**

Why choosing to use a homogenous flux instead of a spatially varying flux that directly compensates for the bottom loss at the same location? Wouldn't this framework to some degree arbitrarily homogenize the resource distribution across the ocean?

**Response: In theory, this could be the case. We however show in the paper that the global open ocean distributions of for instance nutrients (DIN, DIP, DSi) and NPP are to a greater extent dictated by ocean circulation. Firstly, the distributions of these variables (for instance DIN oceanic concentrations) are not homogeneous in REF, which does not reflect the distributions of the inputs. Therefore the locations of the inputs clearly do not strongly dictate the distribution. Secondly, the difference between RIV-REF is relatively small compared to the "background" signal (For instance in the case of the NPP: Figure 8b). The mechanisms explaining the stronger importance of ocean circulation are that the fluxes added to the ocean to compensate for the sediment losses are small, especially per area, in comparison to the oceanic inventories of the compounds. This could be shortly shown quantitatively in the Supplementary Information of the revised manuscript. Secondly, the nutrients are transported away from their points of "addition" relatively rapidly, since we do not observe accumulation in the open ocean at the points of inputs (-> the distribution of the nutrients and NPP in the ocean is not at all homogeneous).**

**The reviewer is however to a certain degree correct, since by adding nutrient fluxes homogeneously to the open ocean, nutrient concentrations and NPP are slightly artificially increased due to the addition of nutrients to the open ocean. In reality there are no homogeneous surface fluxes to the open ocean. This is what we improve with the simulation RIV by eliminating biogeochemical compound fluxes to the open ocean and adding them to river mouths.**

Why adding the flux at the surface instead of evenly distributing it throughout the water column?

**Response: Our approach to add nutrient fluxes to the surface (REF) is slightly more realistic than adding them through the water column, since riverine fluxes also enter the ocean at the (near-) surface. Furthermore, adding biogeochemical compounds within all of the water column would also create an artificial signal: in areas where upwelling (rising of water masses from deeper layers to the surface) takes place, there would be an unwanted increase of compound supply. For instance, nutrients added to the deeper water column layers would artificially increase the NPP at the surface of upwelling areas, which is also not at all realistic. That being said, the RIV scenario is the most realistic of all mentioned scenarios, since it also considers the correct geographical location of river inputs.**

How about distributing this flux only along the coastline (acting as a riverine source)?

**Response: This is indeed a more realistic solution than was chosen in REF, but it would not allow us to investigate the effects caused by adding rivers to their correct geographical location in contrast to adding them in the open ocean, which is part of the focus in this manuscript. The difference between RIV and the suggested simulation would not be as large, since adding the flux to the coastline resolves to a certain degree the geographic distribution of the riverine fluxes. We are also not aware of any model approaches that tackle riverine inputs as a homogeneous coastline source, and therefore the found differences would not be of great use.**

It seems that in this framework carbon (and other materials) is being relocated from the bottom to the surface and from some places to others without any explicit transport processes involved. And this would potentially make a HUGE impact on NPP and CO2 outgassing patterns regardless of riverine inputs.

**Response: We show in Figure 8 that the impacts on the NPP between RIV-REF are relatively small, despite this large difference in distributions of the biogeochemical inputs. Therefore, the changes suggested by the editor are unlikely to be huge. We also would like to point out that this mentioned re-location is what happens in reality: riverine inputs at the surface are thought to approximately compensate sediment losses in the ocean sediment (the bottom). Thus, a "reflecting" sediment is not a more realistic scenario.**

**We already state in the manuscript the dominance of the physical circulation in dictating open ocean biogeochemical distributions and would like to avoid extending the manuscript with further discussions on other scenarios that we do not perform.**

Also, what are other inputs to the REF beside this surface flux? Does REF also include N2 fixation? How is carbon synthesis associated with N2 fixation being handled in the model?

**Response: Both model simulations include N₂ fixation, nitrogen deposition and dust deposition. These are all unchanged between in the simulations RIV and REF. At low DIN concentrations and favorable conditions, cyanobacteria fix N₂ in order to produce organic matter. The detailed description of cyanobacteria activity and their effects in the model can be found in Paulsen et al. (2017).**

**While we state that N₂ is fixed dynamically by cyanobacteria in Section 2.2.1, we will add revised manuscript the inputs of nitrogen deposition and dust deposition, and from which literature sources the values for the inputs were derived.**

Depending on these details and whether the comparison between RIV and REF is justifiable, I recommend either a major (which would require a re-configuration run of the 5000-year REF experiment) or a moderate revision (which would be focused on streamlining and shortening the text plus clarification on some details as suggested below) of the paper before considering it for publication in Biogeosciences.

**Response: Leaning on our previous responses, we do not see the need for an additional reference simulation. Using the current simulations enables us to assess the differences between biogeochemical riverine inputs as derived in the manuscript and adding the biogeochemical inputs to the open ocean, as was done previously in simulations in HAMOCC.**

**We agree that some additional clarification, as well as streamlining of the paper might help facilitate the understanding of the main points.**

**Minor Comments**

P2L13: and also released to the atmosphere

**Response: This is correct and will be corrected in the revised manuscript.**

P2L19-P2L6: introductory information in these a few paragraphs needs to be streamlined. I suggest shortening it to 5-8 lines and expand Fig.1 to include more details on the processes to be considered or discussed.

**Response: We assume the reviewer is addressing the whole introduction. The introduction will be streamlined and shortened in the revised manuscript.**

P3L9: without reading the referenced literature, it is not so clear to me why riverine DIC would cause CO2 outgassing. Is it due to solubility change? Would riverine DOC/POC be also, if not more, likely to cause CO2 outgassing as a result of microbial respiration?

**Response: The exact mechanisms which cause pre-industrial carbon outgassing, are often unclearly explained in literature. In this manuscript, we explain this in detail in section C2. Riverine DIC inputs consist majorly of $HCO_3^-$ inputs (or DIC:Alk = 1:1). In the process of carbonate production, $HCO_3^-$ is consumed while increasing the $pCO_2$ during its lifetime in the ocean:**

$$Ca^{2+} + 2HCO_3^- => CaCO_3 + CO_2 + H_2O =>$$

The remineralization of DOM and POM also cause an increase in pCO₂, the released nutrients through the remineralization process however enhance the biological productivity, which counterbalances this effect and decreases the pCO₂. Therefore, it is the C:nutrients ratio within the organic matter which determines the extent of the pCO₂ and of the outgassing. As an example, for tDOM remineralization (first reaction) and the subsequent uptake of the released nutrients (second reaction):

$$C_{2584}PN^*_{16} => 2584DIC + DIP + 16DIN+^* => C_{122}PN^*_{16} + 2462CO_2$$

The extent of the long-term net outgassing is therefore strongly dependent on the carbon to nutrients ratio within POM and DOM.

P3L20: where does photodegradation most likely occur? In the rivers? Coastal margins? Is this process not reflected in the extremely high C:P ratio (2584:1) considered here?

Response: The degradation of the tDOM during its transit time within rivers is likely mostly biotic, thus explaining the low C:nutrients ratios found in tDOM at river mouths. Aarnos et al. (2018) derive from field measurements at 8 major river plumes globally that a substantial amount (around 30%) of tDOM is photodegraded within these plumes. In Fichot et al. (2014), which focuses on the Louisiana shelf, it is suggested that 40-50% of tDOM is mineralized on the shelf, with combined photodegradation and subsequent biotic breakdown being the major pathway for the degradation.

We believe this unfortunately exceeds the scope of our manuscript, since it is already very information-dense.

P3L24: why does POM control the availability of nutrients? By being remineralized?

Response: Indeed, POM contains nutrients that are released during mineralization of the POM. The strength of this control depends on various factors (other nutrient sources, POM composition, POM reactivity etc.).

This can be very shortly mentioned in the revised manuscript.

P4L13: which 10 years? Contemporary?
Response: The mentioned study performs a 10-year simulation only for the present day and for present-day riverine inputs. We can clarify this in the revised manuscript.

P5L8: Is there a particular reason to use 250m rather than the more commonly defined 200m?

Response: Using a depth threshold of 250m yields a better areal representation of the shelves in our relatively coarse resolution. This is especially the case for the narrower shelves.

P7L18: C:P=1000:1 here but 2584:1 in P9L23.

**Response: C:P = 1000 is weight ratio and 2584:1 is the mole ratio. This is mentioned in both statements; we will however avoid switching from weight to mole ratio in the revised manuscript.**

P10 section 2.1.3: why not deriving N:P ratios for different rivers from Global NEWS data set? Or at least to make a comparison with?

**The Global NEWS data set prescribes very high N:P ratios, especially for dissolved inorganic species (approximately N:P = 30). This is plausible since this dataset represents inputs before their processing in estuaries. In estuaries, substantial denitrification and N outgassing takes place (i.e. Seitzinger et al., 2006), but these systems are not representable in the current state of global models due to resolution. We therefore do not have a choice but to simplify the ratios. The global P:N ratio of inputs to the ocean from estuaries should however approximately be 1:16, since the oceanic N source from N fixation and N deposition is thought to be approximately compensated by the sink from denitrification, leaving the elimination through organic matter formation and elimination, which takes place approximately at a P:N ratio of 1:16.**

**We mention our inability to take into account denitrification in p10.l10-l11, and that this escapes the scope of our study. We will add a brief discussion of the limitations of using fixed N:P ratios in section 7.1 in the revised manuscript.**

P12L7: "river freshwater model" -> is this the MPI-ESM model mentioned in P7L3?

**The inputs of the Ocean-Model-Intercomparison-Project do indeed originate from the Hydrological Discharge (HD) which is a component of the global Max-Planck-Institute Earth System Model MPI-ESM.**

P12L9-18: need some clarification on the biogeochemical model configuration and references for different versions of the model. The original model is described in Ilyina et al. 2013, and the version being used in this study is the same with Mauritsen et al., 2018? The major changes include cyanobacterial N2 fixation and incorporation of DOM? Was the DOM improvement also made in Mauritsen et al., 2018? What about Paulsen et al., 2017 and Six and Maier-Reimer, 1996?

**Response: The core of the model, which is essentially still the same, is described in Ilyina et al. (2013). The model developments that were incorporated since the Ilyina et al. (2013) manuscript were published are described in Mauritsen et al., 2018. As stated in our manuscript, the major changes were "to incorporate dynamical nitrogen fixation through cyanobacteria (Paulsen et al., 2017), to follow recommendations from the OMIP protocol (Orr et al., 2017) and to correct errors in the model. " p12.l11-l13**

**The references discussed for these already in the manuscript, cited after the individual model development (for instance Paulsen et al. (2017) for the implementation of cyanobacteria, p12.l12 and p12.l14). The model was already extended from a "classic" NPZD model in the Six and Maier-Reimer (1996) manuscript to consider a DOM pool. This model version did not strongly change from the Six and Maier-Reimer (1996) manuscript to the Ilyina et al. (2013) manuscript.**

**We will attempt to improve the last part of the paragraph in order to improve clarity in the revised manuscript. We will also add a general scheme of the main processes represented in HAMOCC.**

Are these studies relevant to the HAMOCC model development?

**Response: These model developments are relevant to HAMOCC and affect results of the model (See the individual studies).**

Also, is the model (e.g. photosynthesis) N or P or C based?

**Response: We do not fully understand this question. The standard model is an extended NPZD type model (represents nutrients, phytoplankton, zooplankton, detritus), which was extended with DOM and cyanobacteria. The nutrients inorganic pools are DIP, DIN, DSi and DFe and can all be limiting. The phytoplankton and cyanobacteria produce organic matter at a fixed C:N:P ratio (122:16:1) taking up dissolved inorganic nutrients and DIC. The remineralization processes (grazing, or bacterial remineralization) then release dissolved inorganic nutrients and DIC back to the water column.**

**This should be clearer with the process scheme in the revised manuscript.**

Are there different pools for D/POC, D/PON, D/POP, etc?

**Response: For the composition of the organic matter (phytoplankton, cyanobacteria, zooplankton, detritus, DOM), the model solely uses the globally fixed C:N:P ratios. This is mostly due to the computational expenses associated with having to compute 3x5 additional tracers to represent D/POC, D/PON, D/POP.**

**We think that this should be clear with the addition of the HAMOCC process scheme in the revised manuscript, and we can further state all biogeochemical compounds in the model.**

Beside river inputs, are there other inputs from e.g. atmosphere deposition? Is N2 fixation also a carbon input or just N? These important details, particularly on the sinks and sources of N,P,C, need to be provided here.

**Response: This is a good point that was forgotten in the submitted manuscript; the model also represents atmospheric dust and nitrogen deposition. These are estimated datasets for the pre-industrial time frame. Regarding the second question, at low enough DIN concentrations and favorable growth conditions for cyanobacteria (see Paulsen et al. 2017 for details), cyanobacteria produce organic matter (thus take up DIC, DIP, DFe from the water column), while fixing (atmospheric) N2.**

**We will add the description of the dust and nitrogen inputs from atmospheric and their literature sources in the revised manuscript.**

Abstract: focus on the riverine impact on CO2 outgassing and NPP hotspots in the ocean and leave the details of the river export (e.g numbers in P1L8) to the result section. P1L15-19 can be removed or shortened to one sentence.

**Response: This is a good point, but our results are strongly dependent on our estimation of the inputs, which is a substantial part of the manuscript, and we therefore believe it is important to give the reader a grasp of the magnitudes of the exports.**

**References:**

Seitzinger et al. : Denitrification across landscapes and waterscapes: a synthesis. *Ecological Applications*, 16(6), 2064-2090, 2006.

---

## Author Response (AR1)

**Response to Referee #1 (updated)**

**General Comments**

This paper assesses the effect of riverine nutrient and C inputs on C cycling in global oceans, and notably on CO2 outgassing in a pre-industrial state. The work includes quantification of riverine inputs of different elements under inorganic and organic forms, regional coastal shelves analysis, and global ocean biogeochemical modelling. Finally, the authors derive a global land-ocean-atmosphere pre-industrial budget, and discuss the effect of riverine inputs and gaps that need to be further addressed in the future (e.g., including volcanic emissions and the effect of shale organic oxidation). I believe this is a substantial piece of work, improving the current understanding of global C cycling, and totally fitting Biogeosciences' scope.

**Response: We would like to thank the reviewer for the comments and very helpful contributions to improving the manuscript. In this response, we first address the individual comments and then refer to the changes made for the revised manuscript.**

**Specific Comments**

1) In general, the paper is extremely dense and would benefit from being shortened and making it more to the point. Major conclusions should be better highlighted, and repetitions avoided. Also, consider deleting sections with discussions out of the main scope of the study and already addressed in previous work (e.g., on weathering). A few suggestions to improve this point are also listed hereafter, in the specific comments.

**Response: We think that the suggestion to streamline and shorten the manuscript is a good point that has been mentioned by both reviewers, and we therefore considered this for the revised manuscript.**

**We restructured the manuscript with a shortening of the introduction (section 1), the elimination of repetitions in the methods section (section 2), and the merging and shortening of pre-industrial weathering and riverine loads sections (-> section 3), and the merging of information from previous sections into a section 5 (Coastal region analysis).**

2) Further details should be provided on the construction of 2 ocean simulations, RIV and REF. The understanding of the differences between these 2 simulations is key to understand the paper's major conclusions.

**Response: While this was partly done in paragraphs in the methods and in the results and discussion, we also see the need to further highlight the main differences between REF and RIV, and discuss the analysis strategy. These are two things: Firstly, the geographical differences of inputs to the ocean (riverine inputs vs inputs to the open ocean), and secondly magnitudes of the increased carbon inputs to the ocean due to the transfer of carbon from land to ocean.**

**We more clearly stated these points in the revised manuscript (section ). We also believe that the improvements from comment 1) help improve the overview of important points.**

**Minor comments**

L7-8p1. "Thirdly, we quantify the terrestrial origins. . . in the framework, . . ." -> It is not clear from this sentence if this is purely a modelling exercise of if you are assessing global oceanic C budgets using the coupled land-atmosphere-ocean model.

**Response: Here, we solely budget for the carbon fluxes from the "stand-alone" models, which are forced with the same pre-industrial atmospheric state. The models do not however give a feedback, which alters the atmospheric state, since the atmospheric CO2 concentration would decline due to the carbon imbalance of the atmosphere (Figure 11). On the terrestrial side, the carbon fluxes are derived from the weathering and the organic carbon export models. On the ocean side, they are derived from the the ocean biogeochemistry model HAMOCC. We therefore do not take into account for feedbacks in the system, as would be done in a fully coupled land-atmosphere-ocean model, but if so, one would need to consider further carbon outgassing sources (long-term volcanic emissions, shale oxidation), in order to have a balanced budget for the pre-industrial atmosphere.**

**This is clarified in the revised manuscript, although the discussion exceeds the scope of the abstract.**

L11p1. "leads to a global oceanic source of CO2" -> "leads a net global CO2 emission to the atmosphere of". To be consistent with the following sentence, the "source" would be 183+128 TgC/yr (Fig. 11).

**Response: We implemented this correction in the revised manuscript.**

L14p1. It is not clear what a sink due to a model drift is without reading the whole paper in depth. . . **Response: This is a good point, and we think that the word "model disequilibrium" is more understandable for the reader.**

Introduction. The introduction provides a lot of information, some of it not totally relevant to the focus of the study. It should be shortened and re-organized to better highlight current state of knowledge, gaps and how they are addressed in this work.

**Response: We will improve the structure and points of focus of the introduction in the revised manuscript. We will shorten the "background information" paragraphs to only include the most relevant information for the study and to increase the focus on what has been done/not done.**

L15p2. "these knowledge gaps" -> all knowledge gaps are presented in paragraph 7.

**Response: We agree that this part of the sentence was redundant and therefore removed it.**

Paragraphes 2-6. These present in detail processing in watersheds, gaps in knowledge on tDOM and POM degradation and transfers, etc. These could be substantially shortened, since these are not points tackled in the paper, which waters down the main message/focus of the paper. Parts of it (e.g., uncertainties on degradation of different OM forms, desorption of P as it enters saline waters etc.) could however be used in the Discussion (e.g., subsection 7.2).

**Response: We understand the point of view of the reviewer, and also therefore shortened the paragraphs. However, some of the information within these paragraphs is also vital in order to understand the assumptions chosen later in the methods section. For instance, due to the previous degradation of tDOM within rivers, we assume it is less reactive once it reaches the ocean than oceanic DOM, but not completely inert. Therefore it`s remineralization rate was chosen for it to have a longer lifetime than oceanic DOM.**

**In the revised manuscript, we will shorten these paragraphs and increase their focus.**

L2p5. "Pre-industrial" could be defined here (state in 1850).

**Response: We defined pre-industrial here in the revised manuscript.**

L5p5. "spatially explicit quantification of global riverine loads" -> "spatially explicit quantification of riverine exports to global coasts". It is not clear otherwise if the loads within watersheds are quantified as well.

**Response: The loads are indeed quantified for every watershed. We agree this might be somewhat unclear here and corrected this in the revised manuscript.**

L10p5. "We first briefly..." -> Description of the Method's content is already described in the previous paragraph. These introductory sentences could be most of the time removed to make the paper shorter and more to the point.

**Response: We removed this sentence, along with other needless repetitions in the revised manuscript.**

L12p5. Define what you consider like alkalinity here.

**Response: Alkalinity (which is carbonate alkalinity in this case of our derived riverine loads) is more clearly defined in the revised manuscript. This is however more suited in section for the riverine alkalinity loads, and section for alkalinity in the ocean.**

L20-32p5. This is described in detail in the subsections; it could be deleted to avoid repetitions.

**Response: We believe it is important to give a brief overview of all derived loads before addressing the individual elements. We think that the reader will be missing the "big picture" if we start with individual details of every compound load. We however removed large parts of the paragraph, which might be unnecessarily repeating information from later in the revised manuscript.**

Fig.1. Precise that the C from weathering sources is DIC (OC assumed to originate only from the uptake of atmospheric CO2).

**Response: This has been added to the figure in the revised manuscript.**

L12-14p7. Doesn't river damming affect POM loads between pre-industrial and the 1970s? This point could be considered later in the discussion on river loads.

**Response: We assume in the whole study that the organic matter loads have remained constant globally since the pre-industrial time period. While increased organic matter supplies to catchments have been suggested in literature (for instance in Regnier et al. 2013), increased retention and remineralization (for instance due to damming, Maavara et al., 2017) has also been reported. There is however yet a study to spatially quantify these combined effects, while**

comparing the pre-industrial time-frame to the present day. This is briefly already discussed in 4.1. and we believe a more comprehensive discussion of these combined anthropogenic influences escape the scope of our study.

L13p8.  "Pre-industrial runoff..." -> this is already explained L1-5p7,  not necessary. This repetition occurs several times throughout the paper.

**Response: We removed these repetitions in the revised manuscript.**

L24p8.  Are the soil types listed here those with typically low erosion rates?  Is the 0.1 factor also used for wetlands and areas with a high groundwater table?  If yes, what definition did you use for "high water table"?

**Response: The soil types listed here are indeed the ones assumed to have very low weathering rates.  The wetland areas are the Gleysols. These (along with the other soil types) are defined in the Harmonized World Soil Database (Fao et al., 2009). The description found in the database is that gleysols are soils with permanent or temporary wetness near the surface. The derivation of the soil shielding factor is extensively described in Hartmann et al. (2014). In our opinion, further details escape the scope of our study.**

L4p9.  What do the 1.6 TgP/yr used here correspond to?  Fertilizer P in surface runoff reaching rivers? Hart et al. (2004) report an annual fertilizer consumption of 873 TgP in 1913.

**Response: This number was wrongly cited. The 1.6 Tg P to catchments was derived from Figure 3 in Beusen et al. (2016), which on its side derived from inputs described in Hart et al. (2004), as is cited in the corresponding technical manuscript (Beusen et al., 2015). Assuming 873 Tg of P inputs of fertilizer to agricultural land for the year 1913 (Hart et al., 2004), this relatively high value seems plausible.**

L7-9p9.  Why is it reasonable to assume P equilibrium in soils at pre-industrial state (besides that state-of-the-art models usually use this initialization)?

**Response: We agree this assumption is a limitation of our study. For instance, Filipelli et al. (2008) suggest an increase of P inputs to catchments due to increased soil erosion caused by deforestation and land use change. This likely already had an impact on land-ocean fluxes prior to industrialization, although it is completely unknown how large these fluxes could be. This point goes hand-in-hand with our assumption that the organic matter fluxes remain near constant, which is at least consistent within our study.**

L9-11p9.  The assumption that spatial distribution of P river loads is the same in pre-industrial and 1970 is quite strong. For example, agriculture was probably much more developed  in  North America and  Western  Europe  than  in  Asia  at  the  beginning  of  the  20th  century.  Implications of  this  assumption  should  be  discussed  later  in  the  discussion  on  river  loads.  Why  not  use  load distribution from models describing earlier states (e.g., Beusen et al., 2016)?

**Response: This is a good point, we therefore performed an analysis comparing the assumed anthropogenic input distribution in this study (directly derived from NEWS2 year 1970), with the distribution of 1900 and 2000 inputs from the Beusen et al. (2016) study. The global distribution of all three cases are very similar, although there is an increase in some parts of Southeast Asia between 1900 and 2000, which can be observed from the results of the Beusen et al. (2016) study. At the global scale, we deem our assumption to be plausible enough. The analysis can be found in Supplementary information S.1.1.**

L23-24p9. Molar C:P ratios are already provided earlier. You can just say L19 that P is incorporated in organic matter accordingly to C:P molar ratios for tDOM and POM.

**Response: We corrected this in the revised manuscript to avoid repetition.**

L6-12p10. Precise here that DIN export was calculated by subtracting the part contained in organic matter. Was DFe export equal to Fe inputs to catchments?

**Response: The Fe composition of the organic matter is also subtracted from the Fe inputs to the catchments, which results in DFe. It is clarified in the revised manuscript.**

L11-16p11. This is not totally clear. Do you assume that only HCO3- affects alkalinity and that DIC is only HCO3- as well (Alk:DIC = 1:1)?

**Response: We asumed that only HCO$_3^-$ release from weathering affect the alkalinity and that the Alk:DIC ratio is 1:1. This was already stated and justified in the paragraph, but we have made some modifications to the paragraph to make it more understandable.**

Sub-subsection 2.1.5 Silica. Why not use the same weathering model type as for P (Hartmann et al., 2011)?

**Response: In theory, this would be possible, but the model would first need to be calibrated for Si. Since the Beusen et al. (2009) model is already calibrated for Si, we used this one out of simplicity.**

Sub-subsection 2.2.1 Ocean biogeochemistry. A scheme with model state variables and processes would be helpful here (as well as a description of biogeochemical processing equations and parameters as Supplementary Information).

**Response: We added an overview of biogeochemical processes in Figure 1, a scheme which illustrates the main processes represented in HAMOCC (Appendix A) in the revised manuscript. Regarding the equations, this is too extensive in the case of this manuscript. We refer in our manuscript to the model description study of Ilyina et al. (2013) for equations and parameters, with more recent model developments described Paulsen et al. (2017) for cyanobacteria, and in Mauritsen et al. (2019). This has been already published, and in case of further interest, the model code can be requested. We however nevertheless provide a list of parameters in comparison to the ones given in Ilyina et al. (2013) study in a Supplementary Information S.2.**

L22p13. How did you choose the 0.003 d-1 value for tDOM degradation? 0.008 d-1 for the oceanic DOM is also in the literature range for tDOM degradation.

**Response: The assumption chosen here was that the tDOM entering the ocean at the river mouths is less reactive than the "freshly produced" oceanic DOM, since tDOM is reported to have been strongly degraded previously during its transport in rivers. The tDOM is however also not inert in the ocean. This is stated in paragraph p11.l25-p11.l30.**

2.2.3 Pre-industrial ocean biogeochemistry model simulations. Please provide more details here, since the distinction between the REF and RIV simulations is key to understanding the paper's results. In the REF simulation, are input fluxes (per surface area?) globally homogeneous? Or do they compensate sediment losses to reach equilibrium locally (at the cell scale)? In the RIV simulation, are there also open ocean surface inputs in addition to river inputs?

**Response: In short, the REF inputs are added globally homogeneously (per surface area) to the ocean surface. This represents oceanic inputs by (to a major part) passing the coastal ocean and does not at all represent geographic riverine inputs. Their magnitudes are dictated by losses to the**

sediment: every few hundreds of years, the averaged global burial loss of biogeochemical compounds was computed, and these were the fluxes added to the surface ocean. They are indeed homogeneous per surface area. In the RIV simulation, there is no open ocean surface inputs, since these were replaced by riverine inputs, derived as shown in the previous sections of the manuscript.

**We added clarification to these points extensively in the revised manuscript section 2.2.3 and also in the results section 4.1.**

L1-6p14. What do you call quasi-equilibrium? How were the lengths of the different simulation chosen? Why did you perform a succession of 3 runs for the RIV simulation? Does "standard simulation" (L4) refer to REF?

**Response: The lengths of the simulations were chosen accordingly to the model state, meaning the simulation were performed until no strong drifts of model variables was remaining. Since biogeochemical processes in the ocean, and especially in the sediment, likely need larger simulation time-periods to perfectly equilibrate than is feasibly possible with current computing resources, there will however always remain a small drift in model variables in such simulations. Therefore, this is a state of quasi-equilibrium and not a perfect equilibrium state.**

**The 3 sequential simulations for the RIV simulation were done in order to achieve a more stable state in the ocean sediment. The motivation here is that it is much less expensive in terms of computing power to simulate the sediment separately from the ocean, and since the time-scales of processes taking place in the sediment are much longer than in the water-column, it makes sense to perform simulations for the sediment alone once the oceanic water column has reached a state close to equilibrium. We therefore firstly performed a simulation of 4'000 years including both the water column and sediment biogeochemistry, until the global particulate fluxes from the water column to the sediment were approximately stable. Then simulations of the sediment component of the model, while being given the stable global particulate fluxes from the previous simulation. Finally, the sediment state was re-coupled to the ocean water-column for the final 2000 year simulation.**

**We have added some clarification in the revised manuscript, but the discussion of the computational expenses of performing of the model spin-up is too technical for the scope of this journal, and would substantially unnecessarily lengthen the manuscript.**

Are the 100-year means (output results) calculated on the last 100 simulated years for each simulation? What are the simulation timesteps?

**Response: The last 100 year mean of both REF and RIV were used to the analysis. Using a 100 year mean prior to this would not affect the results much. The simulation timestep is one hour. This was added to the revised manuscript.**

It is mentioned that water loads vary inter-annually based on runoff inputs from OMIP. Are the ocean physics' inter-annual variations modelled with the same patterns for every simulated year?

**Response: The parameters within the physical circulation model MPI-OM remain the same. We would like to point out that the model simulates the response of the ocean physics dynamically to the atmospheric state, from which the information is given to the model for every time-step. There is therefore inter-annual variation in the physical features of the ocean, and even variation at every model time-step.**

**It escapes the scope of our study to completely explain mechanisms of such an ocean circulation model in the scope of our study; this has been done many times before and we would like to refer to the cited references for our specific model (Junclaus et al., 2013 and Mauritsen et al., 2019).**

L20-24p16. For which periods were these Si loads from the literature estimated?

**Response: These estimates are for the present day. We will mention this in the revised manuscript.**

L26-32p16. Comparison with Mackenzie et al. (1998) does not seem necessary here, especially since they assess only the TIC load.

**Response: We agree and therefore removed this comparison in the revised manuscript.**

Subsection 3.1 Runoff, precipitation and temperature patterns. Hydrology model performance is not the scope of the paper; this section seems to water down once more the message/goal of the study. The scaling of the MPI-ESM runoff should be detailed earlier, in the Methods section. How is the factor 1.59 chosen? Why is the runoff not scaled to the OMIP one, to be fully consistent with the freshwater inputs?

**Response: Factor 1.59 was chosen in order to scale the average of Fekete et al. (2002) and Dai and Trenberth (2002) In the revised manuscript, this is more clearly stated. In order to shorten the manuscript, we will merge sections 3 with section 4, while shortening both with only the most central information for the next parts of the manuscript remaining. The discussion of the weathering model input parameters runoff, temperature and precipitation was moved to Appendix B.**

Subsection 4.1 Global loads in the context of published estimates L2p19. Does the Fe-P load given in Table 3 only corresponds to the fraction that is desorbed when entering the estuary?

**Response: The Fe-P loads refer to desorbed P loads both for the given modelled estimates (Model. Global load), as well as for the Compton et al. (2000) values. This was stated in the Table 3 caption in the revised manuscript.**

Table 3p19. Precise for the POP comparison that the 5.9 value (for 1970) is for PP, and not only POP.
**Response: This was added to the Table 3 caption in the revised manuscript.**

L20-23p20. Doesn't the potential increase in Si retention at the end of the 20th century mainly concern particulate forms?

**Response: This is not correct. Global reservoirs have also been suggested to retain a significant amount of DSi (Lauerwald et al, 2013; Maavara et al., 2014). This is due to in-reservoir formation of biogenic silica, which is then in turn also increasingly retained.**

L20-26p21. Please better organize text.

**Response: We restructured part of the paragraph and removed unimportant information in the revised manuscript.**

L4p22. Could the discrepancies also be due to the fact that weathering formalisms are less adapted to Arctic regions?

**Response: This is indeed very likely, with weathering mechanisms in the Arctic being likely more complex than is modelled here (for instance due to permafrost, Hartmann et al., 2014). We briefly mentionen this in the revised manuscript.**

5.1 Ocean state – An increased biogeochemical coastal sink L14-15p25. "There is a higher carbon load originating from organic matter, since tDOM C:P ratio is higher than the oceanic DOM C:P ratio" -> This explanation is not straight-forward, since the estimation of DOC is not related to P in the model.

**Response: This is correct, the outgassing is caused by the higher higher carbon loads contained in tDOM, which were not accounted for in REF. However, if the C:P ratio in tDOM were the same as for oceanic organic matter, the DIP and DIN released would drive the biological production to take up DIC, which would compensate the additional DIC originating from remineralization of tDOM in the long term. For oceanic organic matter exported to the sediment, 122 mol C is also exported for every mol P. For the river inputs however, for every mol P 2583 mol C are added. Therefore there is an accumulation of DIC when tDOM is mineralized, which leads to higher $pCO_2$ and outgassing. This is shown in equation C2 in Appendix C1. We have also clarified the sentence in the revised manuscript.**

L15p25-L2p26. Comparison with other studies was already discussed earlier.

**Response: The central point here is that the carbon inputs are increased in RIV comparison to REF (the model version used up to now), which is an agreement with previous estimates. This is why carbon outgassing takes place and therefore we believe this is central information here, even if it has been stated before since we improve the model with regards to its carbon inputs.**

L17-19p26. Isn't the effect of large oxygen minimum zones on DIN concentrations also visible in WOA dataset?

**Response: This is indeed the case and a sign that a substantial magnitude of denitrification takes place in the region.**

What explains the major differences in DIP, DIN and DSi concentrations in the Southern Ocean?

**Response: This is likely due to the relatively poor representation of the complex ocean circulation of the Southern Ocean in global ocean circulation models. An improved representation would require high enough resolution to better represent vertical mixing taking place in the Southern Ocean and a better representation of the effects from storms.**

Table 5.p27. Do N inputs include N2 uptake from the atmosphere in the two simulations?

**Response: Both simulations consider dynamical $N_2$ fixation by cyanobacteria as well as (natural) nitrogen deposition. This is already stated in the methods (section 2.2.1, p10.l19 in revised manuscript): "The changes were made to incorporate dynamical nitrogen fixation through cyanobacteria (Paulsen et al., 2017), .."**

L12-14p28. Comparison of Arctic concentrations to the WOA database could be presented earlier in text, with the rest of the comparisons.

**Response: We agree and implemented this change in the revised manuscript, along with eliminating unnecessary information in the section.**

7.1 Rivers in an Earth System Model setting. Paragraph 2 (L12-17p36). These improvements could further avoid strong assumptions, such as globally constant N:P ratios for river inputs, and on the spatial distribution of non-weathering sources.

**Response: These are both very good points and we added a brief discussion of these to the subsection in the revised manuscript, as well as an analysis in Supplementary Information S.1.1 (anthropogenic input distribution), S.1.2 (allochthonous input distribution) and S.1.3 (N:P ratios).**

L11-14p37. Is the need for a runoff scaling factor a major conclusion of this work?

**Response: This is a more technical conclusion, and therefore we have eliminated it.**

L17-21p37. "Even for present-day…" -> comparison with literature is already discussed earlier, this could be removed.

**Response: We agree and the sentence will be removed in the revised manuscript.**

L26-27p37. "which have been identified to be more strongly controlled by extreme hydrological events than other C species" -> Was this considered in the study? What does it imply? It seems that this belongs to some discussion (that should be more developed if added).

**Response: The NEWS2 model results, which we derive POC exports from, calculate the year-means of these exports. Therefore, these extreme events are not at all grasped in the model. We see this more as a conclusion than a discussion point: One might need to better resolve the intra-annual variability of riverine loads for regions with high annual POC exports.**

L17p38. "despite previously being sinks" -> be more explicit by explaining that this is without accounting for river inputs, and not "previously" in a temporal way.

**Response: We corrected this statement in the revised manuscript.**

L27-29p38. Interhemispheric C transfers are already mentioned earlier in this section (L1-3p38). Please focus in this last paragraph only on the major points that this study shows have to be included in ESMs to better assess land-ocean-atmosphere C transfers.

**We removed this second discussion interhemispheric C transfer and improved the final paragraph.**

Appendix C L4p42. "Table 3" -> "Table 2". How do you get to the 280 Tg/yr of CO2 drawdown? Some calculations could be shorter explained (e.g., CO2 emissions explained L16- 20p42 and L9-12p43).

**The drawdown of CO2 is calculated from directly from the weathering model (See equations (9) and (10) in section 2.1.4 and explanations in the text p14l5-p14l7 in the revised manuscript.**

**Minor Comments**

L5p1. "in a regional shelf analysis" -> "in regional shelf analysis"; there are more than one.

**This is not correct. It is one analysis of many shelf regions. Therefore "in a regional shelf analysis" is correct.**

L10-11p1. "total C" -> "global C"

**Corrected.**

L13p2. "exported to the sediment" -> "stored in the sediment"

**Corrected.**

L18p2. Define NPP L25p5.

**This sentence was removed, but we now define NPP later in the revised manuscript.**

"their its ratios" -> "their ratios"

**Corrected.**

L30p5. "a fraction P" -> "a fraction of P"

**Corrected.**

L4p11. "and that 2 HCO3-" -> "and of 2 HCO3-"

**Corrected.**

L11p12. "dynamical nitrogen fixation through cyanobacteria" -> "dynamic N fixation by cyanobacteria"

**Corrected.**

L14p13. "its consistence" -> "its composition"

**This sentence was removed.**

L11p17. "much stronger gradients of variation" -> "much higher spatial variability"

**This sentence was removed.**

L21p17. End of sentence is missing or "which" should be deleted.

**This sentence was modified.**

L17p20. Start new sentence at "Dürr et al. . ."

**Corrected.**

L2p21. "and to a framework" -> "and could be fed/incorporated into a framework"

**It is not the river loads that would be incorporated into a framework, but the framework that we present here could be incorporated into an Earth System Model.**

L2p30. "compensate" -> "counterbalance"

**Corrected.**

L15p30. "due higher" -> "due to higher"

**This sentence has been removed.**

L18 p36 "consistence" -> "composition"

**Corrected.**

L19- 25p43. Check parentheses.

**Corrected.**

**References (excluding cited references in the manuscript):**

Beusen, A. H. W. Et al.: Coupling global models for hydrology and nutrient loading to simulate nitrogen and phosphorus retention in surface water – description of IMAGE–GNM and analysis of performance, Geosci. Model Dev., 8, 4045-4067, https://doi.org/10.5194/gmd-8-4045-2015, 2015.

Lauerwald et al.: Retention of dissolved silica within the fluvial system of the conterminous. Biogeochemistry, 112,637-659. 2013.

**Response to Review#2 (updated)**

**General Comments**

The manuscript addresses an important question about material transport across the land-ocean aquatic continuum, and is of particular interest given its global application. I appreciate the substantial amount of work presented here, which includes a synthesis of existing methods to derive a global data set of riverine sources of nutrients and carbonate species to the ocean, long-term simulations of a global ocean-biogeochemical model, and analyses of CO2 outgassing hotspots and the origins and fates of riverine carbon from land/atmosphere to the ocean/atmosphere. I am particularly impressed that the authors were able to run the global model simulations for several thousand years long (even though the model is relatively coarse). The scope of this study is certainly appropriate for publication in Biogeosciences, however the manuscript in this current format requires clarifications in some places while in others the text needs to be shortened and/or streamlined in order to avoid distraction and help readers better capture the key points of this study.

**Response: We would like to thank the reviewer for their helpful comments and criticisms, and are also pleased with the acknowledgement of effort put into synthesizing the state of knowledge of land-ocean biogeochemical exports, as well as our computational expenses to simulate such long time-periods.**

**We agree the manuscript should be streamlined and shortened to a certain extent to improve the reader's understanding of the main points.**

**In this response, we first address the individual comments and then refer to the changes made for the revised manuscript.**

**Major Comments**

**Authors note: The comments found in the next paragraph of the reviewer's comment will be addressed individually.**

My major concern about the study is the comparison of carbon (and other) budgets between the standard simulation (RIV, what does it stand for?) and reference simulation (REF), which led to most key conclusions made in the manuscript.

**Response: In short, the reference simulation REF represents a previous model version of the ocean biogeochemistry model, which is lacking in terms of its constraints and geographical locations of "riverine" inputs. In the simulation RIV, we consider the magnitudes of the pre-industrial riverine inputs as well as their geographic input locations to the best of our knowledge. The focus is to analyze how adding riverine fluxes plausibly in terms of magnitudes, as well as at their correct geographical location (RIV) affected the ocean's biogeochemical state. This is done in comparison to REF, where these inputs are added to the open ocean. As a note, the overarching goal of having biogeochemical inputs**

in REF is to compensate losses to the sediment; without these inputs, all biogeochemical variables would strive to an equilibrium state equal to zero. This however does not mean that these inputs must be added at the sediment interface (nor is this a more "realistic" solution to the problem), we will revisit this as a response to a later comment.

We improved the explanations of REF and RIV in the methods section 2.2.3., as well as the main differences in the results & discussion section 4.1.

The authors described REF as a configuration where no river inputs are added to the ocean but "the burial loss of biogeochemical tracers was compensated by a global homogenous flux to the surface ocean", such that REF is "fully constrained by the loss of the sediment layer" (P13L26-29). This REF configuration seems a bit odd or at least not so clear to me.

Response: We would like to note that the REF simulation is the state-of-the-art previous model version, which is lacking in terms of its representation of terrestrial nutrient and carbon inputs to the ocean, since it considers these inputs solely at magnitudes to compensate for sediment losses, and they are added to the open ocean. We improve the model in RIV in order to represent the riverine loads magnitudes and geographical locations. We did not design the REF model setup for this study in particular, but it is adequate in order to assess the impacts of riverine loads added at their geographical river mouths (RIV) in comparison to biogeochemical inputs added to the open ocean (REF).

Is this addition of homogenous flux occurs during the same or next time step?

Response: The loss of biogeochemical compounds to the sediment is not computed dynamically within the model, but we approximated the sediment loss fluxes for a 100 year mean around every 1000 years of the REF simulation. The resulting fluxes are added to the model at every timestep.

We added this information to the revised manuscript.

Why choosing to use a homogenous flux instead of a spatially varying flux that directly compensates for the bottom loss at the same location? Wouldn't this framework to some degree arbitrarily homogenize the resource distribution across the ocean?

Response: In theory, this could be the case. We however show in the paper that the global open ocean distributions of for instance nutrients (DIN, DIP, DSi) and NPP are to a greater extent dictated by ocean circulation. Firstly, the distributions of these variables (for instance DIN oceanic concentrations) are not homogeneous in REF, which does not reflect the distributions of the inputs. Therefore, the locations of the inputs clearly do not strongly dictate the distribution. Secondly, the difference between RIV-REF is relatively small compared to the "background" signal (For instance in the case of the NPP: Figure 8b). The mechanisms explaining the stronger importance of ocean circulation are that the fluxes added to the ocean to compensate for the sediment losses are small, especially per area, in comparison to the oceanic inventories of the compounds. We show this in an analysis in the Supplementary Information S.5; the P inputs are for instance only a fraction of only the surface layer P inventories for various regions. This is because the P inputs are

added over such a large area (in fact, the entire ocean). Furthermore, the nutrients are transported away from their points of "addition" relatively rapidly, since we do not observe accumulation in the open ocean at the points of inputs (-> the distribution of the nutrients and NPP in the ocean is not at all homogeneous).

The reviewer is however to a certain degree correct, since adding nutrient fluxes homogeneously to the open ocean artificially increases nutrient concentrations and NPP to a certain degree due to the addition of nutrients to the open ocean. In reality, there are no homogeneous surface fluxes to the open ocean. This is what we improve with the simulation RIV by eliminating biogeochemical compound fluxes to the open ocean and adding them to river mouths.

Why adding the flux at the surface instead of evenly distributing it throughout the water column?

Response: Our approach to add nutrient fluxes to the surface (REF) is slightly more realistic than adding them through the water column, since riverine fluxes also enter the ocean at the (near-) surface. Furthermore, adding biogeochemical compounds within all of the water column would also create an artificial signal: in areas where upwelling (rising of water masses from deeper layers to the surface) takes place, there would be an unwanted increase of compounds supply. For instance, nutrients added to the deeper water column layers would artificially increase the NPP at the surface of upwelling areas, which is also not at all realistic. That being said, the RIV scenario is the most realistic of all mentioned scenarios, since it also considers the correct geographical location of river inputs.

How about distributing this flux only along the coastline (acting as a riverine source)?

Response: This is indeed a more realistic solution than was chosen in REF, but it would not allow us to investigate the effects caused by adding rivers to their correct geographical location in contrast to adding them in the open ocean, which is part of what we want to investigate in this manuscript. The difference between RIV and the suggested simulation would not be as large as between RIV and REF, since adding the flux to the coastline resolves to a certain degree the geographic distribution of the riverine fluxes. We are also not aware of any model approaches that tackle riverine inputs as a homogeneous coastline source, and therefore the found differences would not be of great use to assess what models are missing by misrepresenting riverine inputs.

It seems that in this framework carbon (and other materials) is being relocated from the bottom to the surface and from some places to others without any explicit transport processes involved. And this would potentially make a HUGE impact on NPP and CO2 outgassing patterns regardless of riverine inputs.

Response: We show in Figure 8 that the impacts on the NPP between RIV-REF are relatively small, despite this large difference in distributions of the biogeochemical inputs. Therefore, the changes suggested by the editor are unlikely to be huge. We also would like to point out that this mentioned re-location is what happens in reality: riverine inputs at

the surface are thought to approximately compensate sediment losses in the ocean sediment (the bottom). Thus, a "reflecting" sediment is not a more realistic scenario.

We already state in the manuscript the dominant role that the physical circulation in dictating open ocean biogeochemical distributions and would like to avoid extending the manuscript with further discussions on other scenarios that we do not perform.

Also, what are other inputs to the REF beside this surface flux? Does REF also include N2 fixation? How is carbon synthesis associated with N2 fixation being handled in the model?

Response: Both model simulations include $N_2$ fixation, nitrogen deposition and dust deposition. These are all unchanged between in the simulations RIV and REF. At low DIN concentrations and favorable conditions, cyanobacteria fix $N_2$ in order to produce organic matter. The detailed description of cyanobacteria activity and their effects on the ocean state can be found in Paulsen et al. (2017).

While we state that $N_2$ is fixed dynamically by cyanobacteria in section 2.2.1, we will add revised manuscript the inputs of nitrogen deposition and dust deposition (in both model simulations), and from which literature sources the values for the inputs were derived. Both these atmospheric deposition fluxes are however identical for REF and RIV, so they do not impact the differences between the simulations.

Depending on these details and whether the comparison between RIV and REF is justifiable, I recommend either a major (which would require a re-configuration run of the 5000-year REF experiment) or a moderate revision (which would be focused on streamlining and shortening the text plus clarification on some details as suggested below) of the paper before considering it for publication in Biogeosciences.

Response: Leaning on our previous responses, we do not see the need for an additional reference simulation. Using the current simulations enables us to assess the differences between biogeochemical riverine inputs as derived in the manuscript and adding the biogeochemical inputs to the open ocean, as was done previously in simulations in HAMOCC.

**Minor Comments**

P2L13: and also released to the atmosphere

Response: This is corrected in the revised manuscript.

P2L19-P2L6: introductory information in these a few paragraphs needs to be streamlined. I suggest shortening it to 5-8 lines and expand Fig.1 to include more details on the processes to be considered or discussed.

Response: We assume the reviewer is addressing the whole introduction. The introduction was streamlined and shortened in the revised manuscript.

P3L9: without reading the referenced literature, it is not so clear to me why riverine DIC would cause CO2 outgassing. Is it due to solubility change? Would riverine DOC/POC be also, if not more, likely to cause CO2 outgassing as a result of microbial respiration?

**Response: The exact mechanisms which cause pre-industrial carbon outgassing, are often unclearly explained in literature. In this manuscript, we explain this in detail in section C2. Riverine DIC inputs consist majorly of $HCO_3^-$ inputs (or DIC:Alk = 1:1). In the process of carbonate production, $HCO_3^-$ is consumed while increasing the $pCO_2$ during its lifetime in the ocean:**

$$Ca^{2+} + 2HCO_3^- => CaCO_3 + CO_2 + H_2O$$

**The remineralization of DOM and POM also cause an increase in $pCO_2$, the released nutrients through the remineralization process however enhance the biological productivity, which counterbalances this effect and decreases the $pCO_2$. Therefore, it is the C:nutrients ratio within the organic matter which determines the extent of the $pCO_2$ and of the outgassing. As an example, for tDOM remineralization (first reaction) and the subsequent uptake of the released nutrients (second reaction):**

$$C_{2583}N_{16}P^* \rightarrow 2583DIC + DIP + 16DIN +..^* \rightarrow C_{122}N_{16}P + 2461CO_2$$

**The extent of the long-term net outgassing is therefore strongly dependent on the carbon to nutrients ratio within POM and DOM. This is all found with more detailed explanation in Appendix C.2.**

P3L20: where does photodegradation most likely occur? In the rivers? Coastal margins? Is this process not reflected in the extremely high C:P ratio (2584:1) considered here?

**We would like to note that we found that the C:P ratio of tDOM was often wrongly stated (2584:1 instead of 2583:1). This was corrected in the revised manuscript.**

**Response: The degradation of the tDOM during its transit time within rivers is likely mostly biotic, thus explaining the low C:nutrients ratios found in tDOM at river mouths. Since the tDOM is already strongly degraded and has a very high C:P ratio, it has been strongly debated how reactive tDOM is within the ocean, since these characteristics hint that biological breakdown might be unfavorable. However, tDOM is also not found in substantial in the open ocean and in the sediment pore water. Therefore, photooxidation might play a strong role in breaking down tDOM in the ocean. Aarnos et al. (2018) derive from field measurements at 8 major river plumes globally that a substantial amount (around 30%) of tDOM is photodegraded within these plumes. In Fichot et al. (2014), which focuses on the Louisiana shelf, it is suggested that 40-50% of tDOM is mineralized on the shelf, with combined photodegradation and subsequent biotic breakdown being the major pathway for the degradation.**

**We believe this unfortunately exceeds the scope of our manuscript, since it is already very information-dense.**

P3L24: why does POM control the availability of nutrients? By being remineralized?

**Response: Indeed, POM contains nutrients that are released during mineralization of the POM. The strength of this control depends on various factors (other nutrient sources, POM composition, POM reactivity etc.).**

**This is very shortly mentioned in the revised manuscript.**

P4L13: which 10 years? Contemporary?

**Response: The mentioned study performs a 10-year simulation only for the present day and for present-day riverine inputs. We clarify this in the revised manuscript.**

P5L8: Is there a particular reason to use 250m rather than the more commonly defined 200m?

**Response: Using a depth threshold of 250m yields a better areal representation of the shelves in our relatively coarse resolution. This is especially the case for the narrower shelves.**

P7L18: C:P=1000:1 here but 2584:1 in P9L23.

**Response: C:P = 1000 is weight ratio and 2583:1 (corrected in revised manuscript) is the mole ratio. We however consistently only state mole ratios in the revised manuscript.**

P10 section 2.1.3: why not deriving N:P ratios for different rivers from Global NEWS data set? Or at least to make a comparison with?

**The Global NEWS data set prescribes very high N:P ratios, especially for dissolved inorganic species (approximately N:P = 29). This is plausible since this dataset represents inputs before their processing in estuaries. In estuaries, substantial denitrification and N outgassing takes place (i.e. Seitzinger et al., 2006), but these systems are not representable in the current state of global models due to resolution. We therefore do not have a choice but to simplify the ratios.**
**We added a brief discussion of these limitations in section 7.1. and an analysis in Supplementary Information 1.3. There, we show that the magnitudes of the global N sink through denitrification in estuaries (Seitzinger et al., 2006) could counterweight the additional N induced by a higher N:P ratio (for instance, from Beusen et al., 2016).**

P12L7: "river freshwater model" -> is this the MPI-ESM model mentioned in P7L3?

**The inputs of the Ocean-Model-Intercomparison-Project do indeed originate from the Hydrological Discharge (HD) which is a component of the global Max-Planck-Institute Earth System Model MPI-ESM.**

P12L9-18: need some clarification on the biogeochemical model configuration and references for different versions of the model. The original model is described in Ilyina et al. 2013, and the version being used in this study is the same with Mauritsen et al., 2018? The major changes include cyanobacterial N2 fixation and incorporation of DOM? Was the DOM improvement also made in Mauritsen et al., 2018? What about Paulsen et al., 2017 and Six and Maier-Reimer, 1996? Are these studies relevant to the HAMOCC model development?

**Response: The core of the model, which is essentially still the same, is described in Ilyina et al. (2013). The model developments that were incorporated since the Ilyina et al. (2013) manuscript were published are described in Mauritsen et al., 2018. As stated in our manuscript, the major changes were "to incorporate dynamical nitrogen fixation through cyanobacteria (Paulsen et al., 2017), to follow recommendations from the OMIP protocol (Orr et al., 2017) and to correct errors in the model. "**

**The references are cited more clearly in the revised manuscript, and we added a list of all model pools. We improved the understandability the last part of the paragraph in order to improve clarity in the revised manuscript. We will also add a general scheme of the main processes represented in HAMOCC in Appendix A. Furthermore, an updated list of the biological parameters can be found in the Supplementary Information S.2.**

Also, is the model (e.g. photosynthesis) N or P or C based?

**Response: We do not fully understand this question. The standard model is an extended NPZD type model (represents nutrients, phytoplankton, zooplankton, detritus), which was extended with DOM and cyanobacteria. The nutrients inorganic pools are DIP, DIN, DSi and DFe and can all be limiting. The phytoplankton and cyanobacteria produce organic matter at a fixed C:N:P ratio (122:16:1) taking up dissolved inorganic nutrients and DIC. The remineralization processes (grazing, or bacterial remineralization) then release dissolved inorganic nutrients and DIC back to the water column. For more information, we refer to the technical manuscript of the model (Ilyina et al., 2013). This should be clearer with the list of pools and the process scheme in the revised manuscript.**

Are there different pools for D/POC, D/PON, D/POP, etc?

**Response: For the composition of the organic matter (phytoplankton, cyanobacteria, zooplankton, detritus, DOM), the model solely uses the globally fixed C:N:P ratios. This is mostly due to the computational expenses associated with having to compute 3x5 additional tracers to represent D/POC, D/PON, D/POP.**

Beside river inputs, are there other inputs from e.g. atmosphere deposition? Is N2 fixation also a carbon input or just N? These important details, particularly on the sinks and sources of N,P,C, need to be provided here.

**Response: This is a good point that was forgotten in the submitted manuscript; the model also represents atmospheric dust and nitrogen deposition. These are estimated datasets**

for the pre-industrial time frame. Regarding the second question, at low enough DIN concentrations and favorable growth conditions for cyanobacteria (see Paulsen et al. 2017 for details), cyanobacteria produce organic matter (thus take up DIC, DIP, DFe from the water column), while fixing (atmospheric) $N_2$.

We added the description of the dust and nitrogen inputs from atmospheric and their literature sources in the revised manuscript.

Abstract: focus on the riverine impact on CO2 outgassing and NPP hotspots in the ocean and leave the details of the river export (e.g numbers in P1L8) to the result section. P1L15-19 can be removed or shortened to one sentence.

Response: This is a good point, but our results are strongly dependent on our estimation of the inputs, which is a substantial part of the manuscript, and we therefore believe it is important to give the reader a grasp of the magnitudes of the exports.

**References:**

Seitzinger et al. : Denitrification across landscapes and waterscapes: a synthesis. *Ecological Applications*, 16(6), 2064-2090, 2006.

[revised manuscript text omitted]

---

## Author Response (AR2)

**Response to Editor**

Dear Prof. Middelburg,

Thank you for the effort you have put in to improve this study. We have carefully addressed the review on a point-by-point basis.

Please also note that we have added a figure in the panel Figure 8 (c), to better illustrate the differences between riverine carbon inputs and riverine-induced carbon outgassing of the northern and southern hemispheres, which were already described in the text. The new sub-figure is referenced at L12p24. There were no other changes undertaken other than this and addressing the comments from the reviewer.

Sincerely,

On behalf of all authors,

Fabrice Lacroix

**Response to Report #1**

**General Comments:**

*Reviewer Comment: In this new version, the authors made the text noticeably more concise and to the point. The discussion is better organized. This makes the paper much clearer and less cumbersome to read. The novelties from this study, and how the latter can be compared to previous work is now easy to grasp.*

*In my previous review, I already acknowledged that this is a thorough piece of work improving our understanding of C cycling at the global scale along the land-ocean-atmosphere continuum. I think its quality now reaches Biogeosciences' standards, but still have some minor comments that could be addressed.*

*Author Comment:* **We would like to thank the reviewer very much for both reviews, which were done in great detail despite the length of the submitted paper. We believe they helped make major improvements to the paper.**

Minor Comments:

*L9p1. "riverine exports leads to" -> "riverine exports lead to"* → **Corrected.**

*L10p1. "results from a source from inorganic carbon loads" -> "results from inorganic carbon loads"* → **Corrected.**

*L11-12p1. "… and a slight model desiquilibrium" ->* This is still not clear if you haven't read the whole paper, and also not the most important conclusion. Maybe just indicating that the sink is mainly caused by uptake and resulting alkalinity production is sufficient. → **Corrected.**

*L19-23p1. "… as well as to consider the long-term…" -> The sentence could be split in 2 here, since it is very long, and presents 2 distinct conclusions of the study, which are not directly related.*

**We agree and split the sentence in two parts, see L19-24p1 from in the revised manuscript.**

*L6p4. The 2 last paragraphs of the Introduction could be more smoothly connected, e.g. by saying that the present study aims at filling the listed knowledge gaps above, or part of them…*

**We have connected the two paragraphs in the revised manuscript as following (L6p4):**

**"To address the knowledge gaps listed above, .."**

*L2p6. It is still not totally clear why the authors decided to scale the runoff. Why would MPI-ESM perform less well than the model from Fekete et al. (2002)? It would seem more logical to scale the runoff to the freshwater inputs used in the ocean model (from OMIP). However, this is probably a detail to the whole study in comparison to other assumptions.*

**This is a good point, which might also illustrate uncertainties in the estimation of the global runoff, for which the Fekete et al. (2002) estimate is still used as the benchmark study despite being relatively dated. In Goll et al. (2014) however, where the performances of 4 ESMs to reproduce the global runoff were assessed, the MPI-ESM was the lowest, whereas other ESMs were closer to the Fekete et al. (2002) estimate. Using the OMIP runoff to scale to would be another potential solution, but the estimates for $HCO_3^-$ weathering release (and exports to the ocean) are already on the low side of literature estimates using the higher scaling factor (to Fekete et al. , 2002). Therefore this higher scaling factor was used. We added a short sentence regarding this in the revised manuscript (Appendix B, last sentence).**

*L10p6. "The Beusen et al. (2005) describes" -> "Beusen et al. (2005) describe"* → **corrected.** *L12p6*

*L23p7. "Pfert,catch was the input from agricultural application of fertilizers, manure and inorganic matter"
-> precise that this is the amount of P reaching surface freshwaters due to surface runoff on agricultural lands within the catchment. It is not totally clear when reading that it is not the amount of fertilizer applied to the agricultural lands.*

**Author Comment: We agree and this was corrected with: "Pfert,catch was the input to rivers from agricultural application of fertilizers, manure and organic matter"(l30p7 of revised manuscript).**

*"P exports due to soil perturbations were not considered" -> This doesn't seem totally accurate, since Pfert,catch from Beusen et al. (2016) does account for disequilibrium in soils.*

**This is true and this was corrected with: "P exports due to changes in organic matter erosion in soils were not considered.." (L6p37)**

*L18-20p8. "We did not consider particulate inorganic phosphorus exports…" -> Even though there are no PIP inputs to the ocean model, this sentence is a bit confusing, since Fe-P exports are estimated. This could be more detailed by stating that for the ocean model inputs, only the desorbed fraction is considered, and the remaining is assumed to be non-bioavailable in the ocean.*

**This is correct and was rephrased in the revised manuscript:**

**L13.p8 "The IP was then fractionated into DIP and Fe-Pw, which was released P that was adsorbed to iron minerals, "**
**L18.p8 "We did not consider particulate inorganic phosphorus exports other than Fe-P, .."**

*L20p10. "and correcting errors in the model" -> this is very vague and maybe not necessary to add here. →* **This was removed in the revised manuscript.**

*L26p11. The origin of some of the parameters is still not totally clear even though the range is discussed (0.003 d-1 degradation rate for tDOM). Is this somehow calibrated? Discussion on this deserves to be refined in 7.2, since due to high C:P ratio, the degradation of tDOM contributes significantly to the estimated CO2 "riverine-induced" outgassing.*

**In a paper being prepared for submission, for which we performed model simulations at a higher resolution than here, we will show how different tDOM degradation rates might impact the fate of tDOM in the ocean and CO2 outgassing on the global continental shelf. The rate chosen here is realistic with regards to the little observational data for validation that we have. Regional validation is relatively difficult in this resolution, but we show in the revised manuscript that the amounts of tDOM degraded at this rate is plausible in comparison to what regional budgets have suggested (Supplementary Information S.2.2).**

*L1p12. Even though the physical modelling part is not the scope of this paper, I think it would be useful to have more information on the hydrology/hydrodynamics inputs and characteristics. E.g., is the meteorology from one specific year used for all simulated years?*

**We have added additional information on in the revised manuscript l12-17p10:**

**"The atmospheric surface boundary data, as well as river freshwater model inputs used to drive the model originate from the Ocean-Model-Intercomparison-Project (OMIP, Röske (2006)) data set, a mean annual cycle for atmospheric parameters at a daily time step from the ECMWF reanalysis**

**project ERA40 data. To produce the OMIP freshwater input climatology, the HD model was driven with the ERA40 atmospheric data (Hagemann and Gates, 2001)."**

**The analysis of the atmospheric and freshwater OMIP data is done relatively extensively in Röske (2006) and Hagemann and Gates, 2001.**

*L9p12. "REF was simulated" -> "REF was run"* → **We changed "was simulated" to "performed", L12p12.**

*L17p12. "was simulated" -> redundant in the sentence.* → **Corrected .**

*L6p13. "0.8 – 4 TgP yr-1" -> To my understanding, the upper limit of the range does not only account for weathering. It is confusing to compare it to the weathering estimates in text, while it was not included in Table 2.*

**The upper range does indeed only consider weathering P release. The iron-bound PIP given in the Compton et al. (2000) is P that was first released during weathering and subsequently adsorbed to iron-manganese oxide and oxyhydroxide particle surfaces. The estimate for this Fe-P is given in Table 1A in Compton et al. (2000). The "detrital " PIP Is given in the Compton et al. (2000) study is on the other hand covalently bound and not available for uptake by the biosphere. We have added the upper limit to Table 2, since it is one of the rare weathering estimates independent from the Hartmann et al. (2014) model.**

*L21p17. "river specie" -> "river species"*

**We changed "species" to "compound" in the revised manuscript L26p17.**

*L23p17. "and to a framework" -> "and constitutes a framework"* → **Corrected L28p17.**

*L3p19. "REF and R" -> "REF and RIV"* → **Corrected L3p21.**

*L1-3p22. Precise that the processes described here are what happens in the model.* → **Corrected l29p21.**

*L11p31. "fixed a N:P ratio for" -> "a fixed N:P ratio of"* → **Corrected.**

*L11p31. "While N:P ratios…" -> "N:P ratios…"* → **Corrected**

*+ precise that "we are incapable" of assessing spatially-variable global N:P ratios for river exports within the present framework (other existing global frameworks do assess variable N:P ratios).*

**We are unaware of any spatially-explicit global models that take into account estuarine transformations that deplete DIN (and potentially other changes to further compound loads). Nevertheless, we added "..which is impossible of representing in our model due to resolution constraints" in the revised manuscript L14p31.**

*L20-22p33. I would end the conclusion by the local importance of riverine inputs, since this is more "impressing" than these 10%, especially that these are fraught with large uncertainties.*

**We think it is better to end the study with a global perspective" despite the uncertainties, since regional models are still better suited to assess these local changes, especially in areas strongly influenced by riverine inputs, whereas the largest advantage of our model is enabling an analysis of the global impacts. 10% is also not negligible at the global scale, especially considering we derive a conservative estimate for the global riverine C flux.**

[revised manuscript text omitted]

---

## Author Response (AR3)

**Response to the Editor**

Dear Prof. Middelburg,

Thank you for the additional points and your large efforts to improve ourstudy. We have carefully addressed the review on a point-by-point basis and revised many parts of the manuscript where the writing was unclear/imprecise, as you suggested.

Sincerely,

On behalf of all authors,

Fabrice Lacroix

**Response to the Editor's comments**

*Editor Comments:* All through:
- be precise if you refer to inorganic or organic carbon

*Author response:* **We have went through the manuscript and fixed some sentences where it was not clear whether inorganic and organic carbon was addressed. At times, we do refer to carbon only, C, this is done voluntarily.**

- use 'on the other hand' only if there is one the hand as well

**This has been corrected throughout the revised manuscript.**

*p. 1., l.6: … long-term fate of …. carbon in the ocean. We…* ->**Corrected**
*p. 2, l. 4: revise: it is not logical to write in the ocean…. are exported offshore…* → **Corrected**
*p.2 l. 10-12: merge this floating sentence with paragraph below* → **Corrected**
*p.2, l. 15 inorganic carbon?* → **Corrected to dissolved inorganic carbon (DIC)**
*p.2, l. 33 reformulate: it previous degradation in rivers to something like degradation during transit in rivers or alike* → **The sentence was reformulated.**
*p. 3, l. 22: Although prior degradation* → **Corrected.**
*p.3 l. 12: delete even* → **Done.**
*p.3, l. 24: the impact of riverine nutrients…* → **Corrected.**
*p.7, l. 25: allochthonous* → **Corrected.**
*p. 10, l.13: diffusion of…* → **Corrected.**
*p. 10, l. 28, 29: total Alk includes carbonate…* → **Corrected.**
*p. 11, l. 26-32: be clear when you have to write rate constant and rate. Now it is unclear.* **This was corrected for the whole manuscript.**
*p.12, l. 20: delete first* → **Corrected.**
*p.12, l. 23: delete ocean* → **Corrected.**
*p. 14, Table entry: the Alk release given is too low. Given that DIC=Alk in rivers (as you state yourself in the text), it should be about 36 and not 18.8 Tmol y-1. Also are you presenting Tmol Alk or Tmol C in this table?* **This is indeed correct and should read 36.8 Teq yr-1. We corrected this both in the table and in the text.**

*p. 19, Figure 3. Why do you present DIC as well as Alk in this figure if they are basically the same but for the unit (again, Tmol equivalents or Tmol C?).* **We removed the Alk in the figure and in the caption, since its addition might have been misleading.**

*p. 21, line 7: increased with respect to what?* → **With respect to the reference simulation (REF), this is corrected in the revised manuscript.**

*p. 21, line 11-12: rephrase, is unclear as written* **The sentence was rephrased as:** **"The global coastal POM deposition range given in a review by (Krumins2013) shows a large degree of uncertainty (0.19-2.20 Gt C yr$^{-1}$), but nevertheless hints that the coastal POM deposition flux is possibly improved in RIV."**

*p. 26, line 9: on the one hand…* → **Corrected.**
*p. 36, lines 16-19: do rewrite these. There is no eq. (16) in the text, uptake of nitrate, but not of ammonium, increases alkalinity, there is no eq. (14).* **The equation references have been corrected in the revised manuscript. HAMOCC calculates alkalinity changes through DIN assuming nitrate and not ammonium.**

*p. 36, line 21: there is no equation D1.* **This refers to D1 in Figure 9 (now Dr1). We changed D1 to Dr1 since it was confusing. The Appendix section is referred to in the Figure 9 caption and in the the text of**

**Section 6.**

*p. 37, line 16: should D2 not be replaced with 11?* **This also refers to Figure 9 and was changed to Dr2 to avoid confusion.**

[revised manuscript text omitted]